
# Logarithmic operators in $c = 0$ bulk CFTs

**Yifei He**

Laboratoire de Physique de l'École Normale Supérieure, ENS, Université PSL,
CNRS, Sorbonne Université, Université Paris Cité, F-75005 Paris, France

## Abstract

We study Kac operators (e.g. energy operator) in percolation and self-avoiding walk bulk CFTs with central charge $c = 0$. The proper normalizations of these operators can be deduced at generic $c$ by requiring the finiteness and reality of the three-point constants in cluster and loop model CFTs. At $c = 0$, Kac operators become zero-norm states and the bottom fields of logarithmic multiplets, and comparison with $c < 1$ Liouville CFT suggests the potential existence of arbitrarily high rank Jordan blocks. We give a generic construction of logarithmic operators based on Kac operators and focus on the rank-2 pair of the energy operator mixing with the hull operator. By taking the $c \to 0$ limit, we compute some of their conformal data and use this to investigate the operator algebra at $c = 0$. Based on cluster decomposition, we find that, contrary to previous belief, the four-point correlation function of the bulk energy operator does not vanish at $c = 0$, and a crucial role is played by its coupling to the rank-3 Jordan block associated with the second energy operator. This reveals the intriguing way zero-norm operators build long-range higher-point correlations through the intricate logarithmic structures in $c = 0$ bulk CFTs.

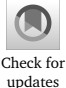

# 1 Introduction

One of the most fascinating remaining challenges in two-dimensional conformal field theories (CFT) is the study of those at central charge $c = 0$. A broad range of interesting critical phenomena are described by $c = 0$ CFTs. This includes the classic examples of geometrical-type phase transitions – percolation and self-avoiding random walk (SAW), Anderson transitions in disordered fermionic systems, and more recently, a newly discovered class of critical phenomena characterized by entanglement [1]. Despite such broad applications, $c = 0$ CFTs are notoriously hard to study. The difficulties first arise from the non-unitarity of the theories, physically inherited from the use of local field theories to describe non-local types of observables or disorder averaging of the physical systems. An even more severe issue is that one encounters the "$c \to 0$ catastrophe" where the operator product expansion (OPE) becomes singular. This is due to the fact that the stress-energy tensor $T$ that controls the two-dimensional conformal algebra becomes a zero-norm state at $c = 0$. In this case, for the theory to be non-trivial, $T$ necessarily becomes part of a logarithmic pair $(t, T)$ and has non-vanishing overlap $\langle T | t \rangle = b$ with its logarithmic partner operator $t$, defining the well-known logarithmic coupling $b$ [2].

While this has to be the case for any non-trivial $c = 0$ CFTs [3, 4], in the particular cases of geometrical phase transitions like percolation and SAW, other logarithmic operators are also known to arise, concerning particularly the energy operator and the hull operators [5, 6]. The logarithmic mixings give rise to two-point correlation functions with logarithmic dependence on distance in addition to the power law behavior, and they characterize geometrical observables at the transitions. Given their ubiquity in the $c = 0$ CFTs, understanding these logarithmic operators – including their operator algebra and correlation functions – appear to be an essential step to tackle these challenging CFTs.

In the early investigations on these logarithmic operators in percolation and SAW, studies have been focused on their two-point functions and logarithmic couplings (analogue of $b$, see e.g. [7–9] for chiral logarithmic CFTs), and little could be said about additional CFT properties like their three-point couplings, OPE and higher-point functions. One new ingredient we have now for making progress is the recent understanding of the cluster and loop model bulk CFTs at generic $c$. It is well-known that percolation and SAW can be defined by taking the limits of random cluster and loop models. In particular, for critical values of the lattice couplings, the Fortuin-Kasteleyn cluster formulation [10] of the $Q$-state Potts model gives rise to percolation clusters in the $Q \to 1$ limit, and the $O(n)$ dilute loop model [11] gives rise to SAW in the $n \to 0$ limit. Moreover, there is another critical phase of the $O(n)$ model known as the dense loop model whose $n \to 0$ limit describes percolation hulls [12]. The continuum limit of the critical cluster and loop models are described by bulk CFTs. Recent numerical and analytical bootstrap techniques [13, 14] have allowed extensive studies of these CFTs at generic $c$. Four-point correlation functions in the cluster and loop models have been computed using the numerical bootstrap approach in a series of work [15–18], where some of the operator algebra and conformal data at generic $c$ have been determined numerically. Furthermore, it has also been realized recently that the cluster and loop model CFTs are in fact logarithmic at generic $c$ [16, 19, 20]. These progresses have allowed a renewed investigation on the percolation and SAW bulk CFTs at $c = 0$ [21]: by taking the $c \to 0$ limit of the spin operator (order parameter) OPE and four-point functions, a rank-3 Jordan block involving the non-chiral field $T\bar{T}$ was uncovered and the operator algebra of the spin operator in the percolation and SAW bulk CFTs at $c = 0$ was determined to leading terms.

In this work, we continue employing this time-honored strategy of taking the $c \to 0$ limit, but focus on the Kac operators, namely operators with integer Kac indices. Kac operators have played a crucial role in the analytic approaches to 2d CFTs: Not only have they allowed solving the minimal models in the early days, the use of Kac operators has also led to analytic solutions to the $c < 1$ Liouville CFT [22, 23], building generic recursion relations of CFT data [14, 24], constructing interchiral conformal blocks in the cluster and loop models [15, 17], and the discovery of logarithmic CFTs at generic $c$ [16, 19, 20]. From the physical point of view, Kac operators correspond to a class of energy-type operators, so their correlation functions encode interesting observables at the geometrical transition. The nature of the Kac operator OPE at $c = 0$ is fundamentally distinct from that of the spin operator and in fact, this involves a special resolution to the "$c \to 0$ catastrophe" where, as pointed out in [25], these operators all acquire vanishing norms and necessarily sit at the bottom of Jordan blocks. Their OPE thus concerns the interesting question of logarithmic operator algebras – an aspect of the $c = 0$ bulk CFTs that has rarely been studied. To carry out the study, the first thing is to understand how to properly normalize these operators. To do this, we will consider the generic $c$ cluster/loop model CFTs, and deduce the non-trivial normalizations of Kac operators (as functions of the central charge $c$) by combining the analytic bootstrap results with the physical principle of real CFTs [26]. The latter says that the cluster/loop model CFTs have real correlation functions as they arise from the long distance descriptions of microscopic models with real and positive measures. Once the operators are properly normalized, we can compute their three-point

couplings by taking the $c \to 0$ limit. For this, we will focus on the simplest Kac operators – the energy operator $\varepsilon$ and the second energy operator $\varepsilon'$ that belong to rank-2 and (the newly uncovered) rank-3 Jordan blocks respectively. These Jordan blocks turn out to have a nontrivial coupling at $c = 0$ and cluster decomposition further allows to examine the fate of the energy operator OPE and four-point function at $c = 0$. Our surprising finding is that the latter does not vanish as previously believed, and the non-trivial long-range four-energy correlation is built through the coupling to the rank-3 Jordan block. The result suggests intriguing ways of building higher-point correlation functions in the $c = 0$ bulk CFTs for zero-norm states through intricate logarithmic structures, and awaits further investigations.

The paper is organized as follows. In the next section, we review two resolutions to the "$c \to 0$ catastrophe" and the logarithmic multiplets involving $T$ and $T\bar{T}$ at $c = 0$. While the latter has been analyzed recently in [21], here we take a slightly different perspective and build the Jordan blocks from properly normalized operators. The construction here is cleaner compared to [21] and provides an example for analyzing the Kac operators next. In the following section, we study the proper normalization of Kac operators $\Phi_{r,s}$ at generic $c$ in the cluster and loop models. We deduce such normalization by analyzing their amplitudes in the four-spin correlation functions at generic $c$ and requiring that the three-point couplings are real. The zeros of the norms at $c = 0$ indicate logarithmic mixing for the Kac operators, and the orders of zeros suggest the ranks of the Jordan blocks. In section 4, we discuss a generic construction of logarithmic operators at $c = 0$ based on Kac operators. The starting point here is the properly normalized Kac operators at the bottom of Jordan blocks. The top field can be written down accordingly and once this is done, conformal Ward identities give the logarithmic couplings solely from the dimensions and the normalizations of the operators. This includes the simplest and most interesting case of the energy density logarithmic pairs $(\tilde{\varepsilon}, \varepsilon)$. In the section after, we compute $c = 0$ conformal data of energy density operator in percolation and SAW CFTs. We focus in particular on the couplings to the rank-2 and rank-3 Jordan blocks of $T, T\bar{T}$ and use these to analyze the OPEs and four-point functions. In the final section, we conclude and discuss some curious aspects of our results that deserve future studies. The appendices provides further references for calculations used in the main text. In appendix A, we briefly summarize some of the conformal data at generic $c$ in cluster and loop model CFTs that are used for analyzing the $c \to 0$ limit. Appendix B and C contain some additional calculations in building the Jordan blocks and computing the $c = 0$ conformal data. In appendix D, we give a brief review of relevant correlation functions and OPEs involving logarithmic operators as derived from conformal Ward identities and cluster decomposition.

**Notations**  Throughout the paper, we will parameterize the central charge as:

$$c = 13 - \frac{6}{\beta^2} - 6\beta^2. \tag{1}$$

To cover the families of cluster and loop models, we will take the range $-2 \le c \le 1$ corresponding to

$$\frac{1}{2} \le \beta^2 \le 1, \tag{2}$$

where $\beta^2 = 2/3$ for $c = 0$. The conformal dimension corresponding to Kac indices $(r,s)$ is given by:

$$h_{r,s} = \frac{1}{4}\left(\left(r\beta^{-1} - s\beta\right)^2 - \left(\beta^{-1} - \beta\right)^2\right). \tag{3}$$

Note that the Kac operators have $r, s \in \mathbb{N}^+$ but operators in the cluster and loop model spectrum can be parameterized generically in this way where $r, s$ can take fractional values. To be

explicit, in this notation, the spin operator is given by the indices:

$$\sigma : (r,s) = \begin{cases} \left(\frac{1}{2}, 0\right), & \text{cluster and dilute loop,} \\ \left(0, \frac{1}{2}\right), & \text{dense loop,} \end{cases} \tag{4}$$

and the Kac operators are given by the indices

$$\Phi_{r,s} : (r,s) = \begin{cases} (1,1), (1,3), (1,5), \dots, & \text{dilute loop,} \\ (1,1), (2,1), (3,1), (4,1), \dots, & \text{cluster,} \\ (1,1), (3,1), (5,1), \dots, & \text{dense loop.} \end{cases} \tag{5}$$

Note that fixing the range (2) means that the indices $(r,s)$ are switched between the dense and dilute loop models. In particular, in our notation, the Kac operator $\Phi_{1,2}$ is degenerate in the dilute loop model and the Kac operator $\Phi_{2,1}$ is degenerate in dense loop/cluster models.[1]

To write the two-point functions, we will use a simplified notation to denote:

$$\mathbb{P}_2^c \equiv \frac{1}{z^{2h(c)} \bar{z}^{2\bar{h}(c)}}, \tag{6}$$

such that for example the two-point functions of non-logarithmic operators at generic $c$ can be written as

$$\langle \mathcal{O}(z,\bar{z}) \mathcal{O}(0,0) \rangle = B_{\mathcal{O}}(c) \mathbb{P}_2^c. \tag{7}$$

The expression depends on the central charge $c$ through $B_{\mathcal{O}}(c)$ and the conformal dimension $(h(c), \bar{h}(c))$ of the operator $\mathcal{O}$. The same notation (6) is also used for two-point functions exactly at $c = 0$ as:

$$\mathbb{P}_2^0 \equiv \frac{1}{z^{2h(c=0)} \bar{z}^{2\bar{h}(c=0)}}, \tag{8}$$

which could be multiplied by a logarithmic factor $\ln(z\bar{z})$. See for example expression (31) below.

For three-point functions, we use a similar simplified notation:

$$\mathbb{P}_3^c \equiv \frac{1}{z_{12}^{h_1(c)+h_2(c)-h_3(c)} \bar{z}_{12}^{\bar{h}_1(c)+\bar{h}_2(c)-\bar{h}_3(c)} z_{13}^{h_1(c)+h_3(c)-h_2(c)} \bar{z}_{13}^{\bar{h}_1(c)+\bar{h}_3(c)-\bar{h}_2(c)} z_{23}^{h_2(c)+h_3(c)-h_1(c)} \bar{z}_{23}^{\bar{h}_2(c)+\bar{h}_3(c)-\bar{h}_1(c)}}, \tag{9}$$

such that for example at generic $c$ one has a standard expression for non-logarithmic three-point functions

$$\langle \mathcal{O}_1(z_1,\bar{z}_1) \mathcal{O}_2(z_2,\bar{z}_2) \mathcal{O}_3(z_3,\bar{z}_3) \rangle = C_{\mathcal{O}_1 \mathcal{O}_2 \mathcal{O}_3}(c) \mathbb{P}_3^c. \tag{10}$$

Additional factors that depend logarithmically on the three operator insertions are denoted as

$$\tau_1 = \ln \frac{z_{23} \bar{z}_{23}}{z_{12} \bar{z}_{12} z_{13} \bar{z}_{13}}, \qquad \tau_2 = \ln \frac{z_{13} \bar{z}_{13}}{z_{12} \bar{z}_{12} z_{23} \bar{z}_{23}}, \qquad \tau_3 = \ln \frac{z_{12} \bar{z}_{12}}{z_{13} \bar{z}_{13} z_{23} \bar{z}_{23}}, \tag{11}$$

and correspondingly $\mathbb{P}_3^0$ denotes the expression (9) at $c = 0$. See e.g. (57) for some of these logarithmic three-point functions.

In this work, we consider the cluster, dilute and dense loop models which at $c = 0$ describe respectively percolation, SAW and percolation hulls. We will use the same notations $\sigma$ to indicate the spin operator and $\varepsilon$ to indicate the energy density operator, but with superscripts e.g. "perco", "SAW", "hull" to distinguish the theory we are referring to.

Finally, we will discuss the proper normalization of an operator $\Phi$ and in this case, we will use a hat ^ to indicate the properly normalized operator $\hat{\Phi}$ and reserve the notation $\Phi$ without the hat for the usual unit normalized operator.

---

[1] As studied in various previous work, the loop model spectrum itself does not contain the Kac operators $\Phi_{1,2}$ or $\Phi_{2,1}$ but their Virasoro degeneracy can be used to establish recursions of operators in the spectrum. See e.g. [17].

# 2 $c \to 0$ catastrophe and Jordan blocks

It was pointed out since the early days of logarithmic CFTs [2, 3, 27] that the OPE becomes singular for CFTs with central charge $c = 0$:

$$\mathcal{O}(z, \bar{z}) \mathcal{O}(0, 0) = (z\bar{z})^{-2h_{\mathcal{O}}(c)} B_{\mathcal{O}}(c) \left( 1 + \frac{h_{\mathcal{O}}(c)}{c/2} (z^2 T + \bar{z}^2 \bar{T}) + \dots \right), \tag{12}$$

where $\mathcal{O}$ stands for some primary operators. This is referred to as the "$c \to 0$ catastrophe" and there are several ways to resolve this issue [25, 28]:

  i There is an additional contribution in ... from an operator $X$ that cancels the singularity.

  ii The scaling dimension $h_{\mathcal{O}}(c)$ vanishes at $c = 0$.

  iii The normalization $B_{\mathcal{O}}(c)$ of the operator $\mathcal{O}$ vanishes at $c = 0$.

In this section, we will study the first two resolutions where resolution i applies for $\mathcal{O}$ being the spin operator (order parameter) in percolation cluster and SAW, and resolution ii applies for $\mathcal{O}$ being the dense loop spin which inserts percolation hulls. Resolution i has been analyzed recently in [21], where the $c \to 0$ limit of the spin operator OPE uncovers a rank-3 Jordan block associated with the field $T\bar{T}$. Here, we start by reviewing this construction, but take a slightly different perspective, focusing on the normalizations of the operators and how vanishing norms naturally lead to logarithmic operator mixings. This will help with the main subject later in this paper to study the resolution iii in the case of Kac operators.

## 2.1 Logarithmic operators at generic $c$

Let us start by recalling the recently discovered logarithmic operators at generic $c$ in cluster and loop models. At generic $c$, the spectrum of Potts cluster and $O(n)$ loop models involve the 4-leg (2-hull) operator $\Phi_{0,2}$ and 2-leg (hull) operator $\Phi_{1,0}$ (illustrated below in Fig. 2) with their conformal dimensions:

$$\begin{aligned} \text{cluster/dense loop:} \quad & \Phi_{0,2}(h_{0,2}(c), h_{0,2}(c)), \\ \text{dilute loop:} \quad & \Phi_{1,0}(h_{1,0}(c), h_{1,0}(c)). \end{aligned} \tag{13}$$

Although the operators in (13) are scalar operators, the CFTs are non-diagonal and the fusions of these hull operators with Kac operator $\Phi_{2,1}$ or $\Phi_{1,3}$ generate the spin-2 4-leg (in cluster model) and 2-leg (in dilute loop model) operator[2]

$$X : (h_{1,-2}(c), h_{1,2}(c)), \qquad \bar{X} : (h_{1,2}(c), h_{1,-2}(c)). \tag{14}$$

Note that the integer Kac indices at generic $c$ indicate that their level 2 descendants

$$\bar{A} X = A \bar{X}, \qquad A = L_{-2} - \frac{1}{\beta^2} L_{-1}^2, \tag{15}$$

have zero norms. However, unlike the case of Kac operators, these zero norm descendants do not decouple from the CFT state space; in fact, they have non-vanishing overlap with another state $\Psi$

$$\langle \bar{A} X | \Psi \rangle = b_{1,2}(c). \tag{16}$$

---

[2]We follow the notation of [21].

The operators $(\Psi, \bar{A}X)$ then form a rank-2 Jordan block. Their two-point functions take the standard form:

$$\langle \Psi(z,\bar{z})\Psi(0,0)\rangle = \left(-2b_{1,2}(c)\ln(z\bar{z}) + \lambda(c)\right)\mathbb{P}_2^c, \tag{17a}$$

$$\langle \Psi(z,\bar{z})\bar{A}X(0,0)\rangle = b_{1,2}(c)\mathbb{P}_2^c, \tag{17b}$$

$$\langle \bar{A}X(z,\bar{z})\bar{A}X(0,0)\rangle = 0, \tag{17c}$$

as a result of conformal Ward identities. Such a rank-2 Jordan block appears in both cluster/dense loop and dilute loop model CFTs (there is a spin-2 hull operator $X$ in both cluster/dense loop and dilute loop models) but the logarithmic couplings $b_{1,2}(c)$ are different: its values are determined by the Virasoro degeneracy of $\Phi_{2,1}$ in the cluster/dense loop model and that of $\Phi_{1,2}$ in the dilute loop model and were computed to be [16, 20]:

$$b_{1,2}^{\text{cluster/dense}}(c) = 2 - \frac{2}{\beta^4} + \frac{4}{\beta^2} - 4\beta^2, \tag{18a}$$

$$b_{1,2}^{\text{dilute}}(c) = 2 + \frac{1}{\beta^6} - \frac{2}{\beta^4} - \frac{1}{\beta^2}. \tag{18b}$$

The two-point functions (17) of course depend on the generic $c$ unit-normalization of the operator $X$:

$$\langle X(z,\bar{z})\, X(0,0)\rangle = \mathbb{P}_2^c, \tag{19}$$

since the Jordan block $(\Psi, \bar{A}X)$ and $X$ are related by Virasoro actions:

$$A^\dagger \Psi = L_2 \Psi = b_{1,2}\bar{X}, \tag{20a}$$

$$\bar{A}^\dagger \Psi = \bar{L}_2 \Psi = b_{1,2}X. \tag{20b}$$

The particular interest in the spin-2 2-hull operator $X$ is due to its coincidental dimension at $c = 0$ with the stress tensor:

$$(h_{1,-2}(c=0),\, h_{1,2}(c=0)) = (2,0), \tag{21}$$

suggesting a possible logarithmic mixing. We will see however that, to properly study this logarithmic mixing, (19) is not the natural normalization at generic $c$ and we will consider a different norm for the operator $X$.

## 2.2 Spin OPE in percolation and SAW

To study the logarithmic mixing of $T$ at $c = 0$, an entry point is the four-point correlation functions of the spin operator $\sigma$ in the cluster and dilute loop models that probe the spectrum of the CFTs. The spin operator in both models carries Kac indices $(1/2, 0)$. In the cluster model, it has the physical interpretation of inserting a random cluster, and in the loop model, it inserts a dilute polymer line. Physically, the four-spin correlators can encode various geometrical configurations of four cluster/line insertions. We focus particularly on the geometric configurations as depicted in Fig. 1. In these cases, the $s$-channel limit $(z_1, \bar{z}_1) \to (z_2, \bar{z}_2), (z_3, \bar{z}_3) \to (z_4, \bar{z}_4)$ of the four-spin correlator contains the Kac operators (including identity operator) in the spectrum [29] and this allows to investigate the stress energy tensor $T$ as $c \to 0$.

In the cluster case, the $s$-channel spectrum of the four-spin correlator $\langle \sigma\sigma\sigma\sigma\rangle^{\text{cluster}}$ includes the following operators:

$$\begin{aligned} &\mathbb{I},\, T,\, \bar{T},\, T\bar{T},\, \Phi_{2,1},\, \Phi_{3,1},\, \dots \\ &\Phi_{0,2},\, X,\, \bar{X},\, \bar{A}X,\, \Psi,\, \dots \\ &\dots \end{aligned} \tag{22}$$

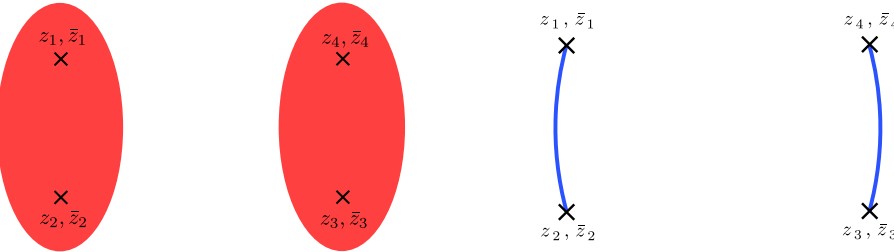

Figure 1: The geometrical configurations described by four-spin correlator in cluster and loop models where the $s$-channel spectrum involves the stress tensor $T$. Left: spin operators at $1, 2$ belong to one cluster and those at $3, 4$ belong to a different cluster. Right: spin operators at $1, 2$ are connected by a dilute polymer line and those at $3, 4$ connected by a different one.

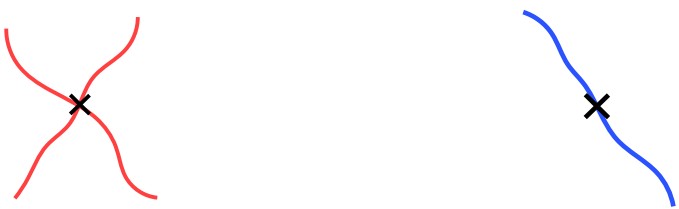

Figure 2: Illustrations of the 2-hull or four-leg operator in cluster model (left) and hull or two-leg operator in dilute loop model (right).

We have listed two specific types of operators: the first line are the Kac operators (and Virasoro descendants – $T, \bar{T}, T\bar{T}$). In particular, $\Phi_{2,1}$ refers to the energy operator $\varepsilon$ and $\Phi_{3,1}$ is the second energy operator $\varepsilon'$. The second line are the hull operators (and their Virasoro family).[3] The operator $\Phi_{0,2}$ has the geometric meaning of joining two clusters (having four legs representing cluster boundaries), the operators $X, \bar{X}$ are similar to $\Phi_{0,2}$ but carry spin 2. See Fig. 2 for an illustration. In the case of dilute loop model, one has analogously:

$$
\begin{aligned}
&\mathbb{I},\, T,\, \bar{T},\, T\bar{T},\, \Phi_{1,3},\, \Phi_{1,5},\, \dots \\
&\Phi_{1,0},\, X,\, \bar{X},\, \bar{A}X,\, \Psi,\, \dots,
\end{aligned}
\tag{23}
$$

where $\Phi_{1,3}, \Phi_{1,5}$ are the energy and second energy operators $\varepsilon, \varepsilon'$ respectively, and $\Phi_{1,0}$ is the hull operator where a polymer line goes through (see Fig. 2). Below we will consider explicitly the case of cluster model since the CFT analysis is identical in these two cases.

To start the analysis, we write down the OPE of cluster spin operator at generic $c$:

$$
\sigma^{\text{cluster}}(z,\bar{z})\sigma^{\text{cluster}}(0,0) = (z\bar{z})^{-2h_\sigma^{\text{cluster}}(c)}
\tag{24}
$$

$$
\begin{aligned}
\times \Bigg\{ &1 + \frac{h_\sigma^{\text{cluster}}(c)}{c/2}\big(z^2 T + \bar{z}^2 \bar{T}\big) + \frac{\big(h_\sigma^{\text{cluster}}(c)\big)^2}{c^2/4}(z\bar{z})^2 T\bar{T} + \dots \\
&+ D_{\sigma\sigma\Phi_{2,1}}^{\text{cluster}}(c)(z\bar{z})^{h_{2,1}(c)}\Phi_{2,1} + \dots + D_{\sigma\sigma\Phi_{0,2}}^{\text{cluster}}(c)(z\bar{z})^{h_{0,2}(c)}\Phi_{0,2} + \dots \\
&+ D_{\sigma\sigma\Phi_{3,1}}^{\text{cluster}}(c)(z\bar{z})^{h_{3,1}(c)}\Phi_{3,1} + \dots \\
&+ D_{\sigma\sigma X}^{\text{cluster}}(c)\bigg((z\bar{z})^{h_{1,2}(c)}\big(\bar{z}^2 \bar{X} + z^2 X + \dots\big) \\
&\qquad\qquad\qquad + g(c)(z\bar{z})^{h_{-1,2}(c)}\big(\Psi + \ln(z\bar{z})\bar{A}X\big) + \dots\bigg) + \dots \Bigg\},
\end{aligned}
$$

---

[3]The field $\Psi$ here belong to the logarithmic Virasoro conformal family of $X$, but is not the Virasoro descendant of $X$.

where operators on the RHS are inserted at 0 and the $D_{\sigma\sigma\mathcal{O}}(c)$'s denote the OPE coefficients. From (24), one sees clearly the singularity as $c \to 0$.

### 2.2.1 Rank-2 Jordan block $(t, T)$

The singularity in the $c \to 0$ limit of the OPE (24) is due to the fact that the two-point function of the stress-energy tensor $T$

$$\langle T(z)T(0)\rangle = \frac{c}{2}\mathbb{P}_2^c, \tag{25}$$

vanishes at $c = 0$. Due to state-operator correspondence, the state $|T\rangle$ becomes a zero norm state in the $c = 0$ CFT state space and is at risk of being removed, rendering the CFT trivial. To avoid this situation, it is necessary that there exists another state $|t\rangle$ whose overlap with $|T\rangle$ does not vanish, defining the $b$-number, or the logarithmic coupling:[4]

$$\langle T|t\rangle = b. \tag{26}$$

To make the OPE (24) well-defined at $c = 0$, the general idea in resolution i is that we need another operator with coincidental conformal dimension to cancel the divergence.[5] The natural choice is the operator $X$ reviewed in section 2.1 whose dimension (21) coincides with that of $T$ at $c = 0$. However, the cancellation suggests that it is more natural to consider an operator $\hat{X}$ with a non-trivial (non-unit) normalization

$$\langle \hat{X}(z,\bar{z})\hat{X}(0,0)\rangle \overset{c\to 0}{\simeq} -\frac{c}{2}\mathbb{P}_2^c, \tag{27}$$

where we have used the hat to indicate the properly normalized operator. Note that in (27), we have used $\simeq$ to indicate the leading behavior of the two-point function near $c = 0$, ignoring sub-leading terms: the operator $\hat{X}$ exists at generic values of central charge and its norm $B_{\hat{X}}(c)$ depends on $c$, but all we are saying here is that the norm has a first order zero at $c = 0$.

It is now straightforward to define the logarithmic partner $t$ of $T$ as[6]

$$t = \frac{b}{c/2}\left(T - \hat{X}\right). \tag{28}$$

The two-point function of $T, t$ gives the logarithmic coupling $b$:

$$\langle t(z,\bar{z})T(0)\rangle = b\,\mathbb{P}_2^0, \tag{29}$$

as we want from (26). We can compute the two-point function of $t$ using the definition (28), eq. (25) and the two-point function of $\hat{X}$ at generic $c$:

$$\langle \hat{X}(z,\bar{z})\hat{X}(0,0)\rangle = B_{\hat{X}}(c)\mathbb{P}_2^c, \qquad B_{\hat{X}}(c) = -\frac{c}{2} + B_{\hat{X}}''c^2 + o(c^2). \tag{30}$$

Taking the $c \to 0$ limit and keeping the $O(1)$ term, we find

$$\langle t(z,\bar{z})t(0,0)\rangle = \left(4b^2 h_{1,2}' \ln(z\bar{z}) + \theta\right)\mathbb{P}_2^0 = \left(-2b\ln(z\bar{z}) + \theta\right)\mathbb{P}_2^0, \tag{31}$$

---

[4]Here, we appeal to the principle that if a state is orthogonal to all other states in the CFT, then this state decouples from the CFT state space. Note that in this case, by operator product expansion, the corresponding operator also does not have non-trivial three-point functions with other operators.

[5]See [30] for an early analysis of the $(t, T)$ pair.

[6]As pointed out in [28], the definition is dimensionally problematic since $T$ and $\hat{X}$ do not have the same conformal dimension at $c \neq 0$. One should in fact include a factor $\mu^{-2h_{1,2}}$ with some scale $\mu$ that ends up in the logarithm $\ln(\mu^2 z\bar{z})$. We suppressed this scale in this work by setting $\mu = 1$.

where $h'_{1,2}$ denotes the derivative of $h_{1,2}(c)$ with respect to $c$ evaluated at $c = 0$, and $\theta = 4b^2 B''_{\hat{X}}$ here. The last equality in (31) is required by conformal Ward identity (eq. (D.9)), and this fixes the logarithmic coupling to be

$$b = -\frac{1}{2h'_{1,2}} = -5 \,, \tag{32}$$

the value that was previously measured on the lattice in [31].

Using the properly normalized operator, we can also compute the three-point functions of $(t, T)$ with the percolation spin operator. At generic $c$, one has:

$$\langle \sigma(z_1, \bar{z}_1) \sigma(z_2, \bar{z}_2) T(z_3) \rangle^{\text{cluster}} = h_\sigma^{\text{cluster}}(c) \mathbb{P}_3^c \,, \tag{33a}$$

$$\langle \sigma(z_1, \bar{z}_1) \sigma(z_2, \bar{z}_2) \hat{X}(z_3, \bar{z}_3) \rangle^{\text{cluster}} = C_{\sigma\sigma\hat{X}}^{\text{cluster}}(c) \mathbb{P}_3^c \,. \tag{33b}$$

The first equation leads to a finite three-point function of the percolation cluster spin operator and the bottom field $T$ of the Jordan block:

$$\langle \sigma(z_1, \bar{z}_1) \, \sigma(z_2, \bar{z}_2) \, T(z_3) \rangle^{\text{perco}} = h_\sigma^{\text{perco}} \mathbb{P}_3^0 \,, \qquad h_\sigma^{\text{perco}} = h_\sigma^{\text{cluster}}(c = 0) = \frac{5}{96} \,, \tag{34}$$

where we have use the superscript "perco" to denote the conformal data exactly at $c = 0$. Moreover, the finiteness of the three-point function $\langle \sigma\sigma t \rangle^{\text{perco}}$ requires

$$C_{\sigma\sigma\hat{X}}^{\text{cluster}}(0) = h_\sigma^{\text{perco}} \,. \tag{35}$$

In previous work [15], we have checked this behavior with results from numerical bootstrap. The resulting three-point function at $c = 0$ is

$$\langle \sigma(z_1, \bar{z}_1) \sigma(z_2, \bar{z}_2) t(z_3, \bar{z}_3) \rangle^{\text{perco}} = \left( C_{\sigma\sigma T}^{\text{perco}} \tau_3 + C_{\sigma\sigma t}^{\text{perco}} \right) \mathbb{P}_3^0 \,, \tag{36}$$

with

$$\tau_3 = \ln \frac{z_{12}\bar{z}_{12}}{z_{13}\bar{z}_{13}z_{23}\bar{z}_{23}} \,, \qquad C_{\sigma\sigma T}^{\text{perco}} = h_\sigma^{\text{perco}} \,. \tag{37}$$

Note the agreement of $C_{\sigma\sigma T}^{\text{perco}}$ from (36) and (34) provides a consistency check.

It is worth commenting that the above construction starts with the well-known two-point function of stress tensor (25), and we naturally define $T$ as the bottom field of the rank-2 Jordan block. However, as already hinted in [21], one can equivalently take the bottom field to be $\hat{X}$ and the rest of the construction, the two- and three-point functions remain the same. See in particular (35) comparing with (34). Namely, the fields $T$ and $\hat{X}$ are at completely equal footing in building the Jordan block. We conveniently denote the bottom field as $T$ but it should really be interpreted as both $T$ and $\hat{X}$ which essentially become degenerate at $c = 0$ and this zero-norm field sits at the bottom of the rank-2 Jordan block. This intuition will be useful for constructing the more complicated higher rank Jordan block which we now turn to.

### 2.2.2 Rank-3 Jordan block of $T\bar{T}$

The appearance of a higher rank Jordan block follows essentially the same idea, namely that due to the vanishing norms of the operators, one needs other operators to cancel the singularities in the OPE, as well as keep the zero norm states in the CFT. This in the meantime gives rise to logarithmic structures in correlation functions. The mixing here is however slightly more complicated as it involves higher order vanishing norms.

Consider the two-point function of the field $T\bar{T}$ at generic $c$

$$\langle T\bar{T}(z, \bar{z}) T\bar{T}(0, 0) \rangle = \frac{c^2}{4} \mathbb{P}_2^c \,, \tag{38}$$

which has a double zero at $c = 0$ and the singularity is supposedly harder to cancel. On the other hand, we have more candidates for logarithmic mixing which can be selected from the spectrum to be the operators with the same dimensions at $c = 0$. In the cluster model this includes (see eq. (22)):

$$\Phi_{3,1}, \Psi, \bar{A}X. \tag{39}$$

Note that the last two fields already form a rank-2 Jordan block (20) at generic $c$, and appear in the logarithmic Virasoro family of the field $\hat{X}$, see (20). In fact, the proper normalization of $\hat{X}$ in (30) also dictates that the properly normalized $(\hat{\Psi}, \bar{A}\hat{X})$ pair has the two-point functions:

$$\langle \hat{\Psi}(z,\bar{z})\hat{\Psi}(0,0)\rangle = \left( -2b_{1,2}(c)B_{\hat{X}}(c)\ln(z\bar{z}) + \lambda(c)B_{\hat{X}}(c)\right)\mathbb{P}_2^c, \tag{40a}$$

$$\langle \hat{\Psi}(z,\bar{z})\bar{A}\hat{X}(0,0)\rangle = b_{1,2}(c)B_{\hat{X}}(c)\mathbb{P}_2^c, \tag{40b}$$

$$\langle \bar{A}\hat{X}(z,\bar{z})\bar{A}\hat{X}(0,0)\rangle = 0, \tag{40c}$$

where we use hat to indicate the properly normalized operators. These two-point functions vanish at $c = 0$, posing the same kind of divergence problem as we have seen above for the spin operator OPE, and one could expect that they are further mixed into higher logarithmic structure, canceling the divergences.

An intuitive guess is that we can mixed two rank-2 Jordan blocks with opposite logarithmic couplings into a rank-3 Jordan block, just as how we have mixed two non-logarithmic fields (i.e., rank-1 Jordan blocks) into a rank-2 Jordan block. Let us try to obtain another rank-2 Jordan block starting with $T\bar{T}$. We first need another field with the opposite normalization near $c = 0$, and the only choice left here is to take the second energy operator $\Phi_{3,1}$ with the two-point function behaving as

$$\langle \hat{\Phi}_{3,1}(z,\bar{z})\hat{\Phi}_{3,1}(0,0)\rangle \overset{c\to 0}{\simeq} -\frac{c^2}{4}\mathbb{P}_2^c, \tag{41}$$

based on the intuition that it should reduce the divergence coming from $T\bar{T}$. Now, to design the top field – call it $\Theta$ – in this first-step rank-2 Jordan block, we use (40) as a guidance. We would like to have the behavior

$$\langle \Theta(z,\bar{z})T\bar{T}(0,0)\rangle \overset{c\to 0}{\simeq} \frac{b_{1,2}c}{2}\mathbb{P}_2^0, \tag{42}$$

where $b_{1,2}$ indicate $b_{1,2}(c = 0)$. This leads to the definition of $\Theta$ as

$$\Theta \equiv \frac{b_{1,2}c/2}{c^2/4}\left(T\bar{T} - \hat{\Phi}_{3,1}\right). \tag{43}$$

Note the definition here is such that the field $T\bar{T}$ and $\hat{\Phi}_{3,1}$ can be equivalently taken as the bottom field (denoted as $T\bar{T}$) with the two-point function (42) – inspired by rank-2 Jordan block construction in the previous subsection. The two-point function of $\Theta$ can be computed (see (C.2) for more):[7]

$$\langle \Theta(z,\bar{z})\Theta(0,0)\rangle = \left(2b_{1,2}^2 h_{3,1}' c \ln(z\bar{z}) + const.\right)\mathbb{P}_2^0$$
$$= \left(-b_{1,2}c \ln(z\bar{z}) + const.\right)\mathbb{P}_2^0, \tag{44}$$

---

[7]here in (44) and above in (42), we have used $\mathbb{P}_2^0$ since the field $\Theta$ defined in (43) becomes logarithmic at $c = 0$. The two-point functions of $\Theta$ however vanish at $c = 0$ and it will further mix into a higher rank Jordan block with finite correlation functions at $c = 0$. See appendix C for more discussions. Later in section 4.3 we will revisit this construction but neglect this intermediate step involving $\Theta$.

where the last line is from conformal Ward identities. This requires

$$b_{1,2} = -\frac{1}{2h'_{3,1}}, \tag{45}$$

which can be easily checked to be true from (18a). This confirms our intuitive argument: we now have two rank-2 Jordan blocks – $(\hat{\Psi}, \bar{A}\hat{X})$ and $(\Theta, \hat{\Phi}_{3,1})$ – at $O(c)$ with opposite logarithmic couplings awaiting further mixing.

We now take the bottom field of a rank-3 Jordan block to be

$$\Psi_0 = T\bar{T}, \qquad \hat{\Phi}_{3,1} \quad \text{or} \quad \bar{A}\hat{X}, \tag{46}$$

whose norm vanish at $c = 0$. The top field is then defined as

$$\begin{aligned}\Psi_2 &= \frac{a}{b_{1,2}c/2}(\Theta - \hat{\Psi}) \\ &= \frac{a}{c^2/4}(T\bar{T} - \hat{\Phi}_{3,1}) - \frac{a}{b_{1,2}c/2}\hat{\Psi},\end{aligned} \tag{47}$$

such that the two-point function with the bottom field is given by

$$\langle \Psi_2(z,\bar{z})\Psi_0(0,0)\rangle = a\,\mathbb{P}_2^0, \tag{48}$$

defining the logarithmic coupling $a$ that characterizes the rank-3 Jordan block. The two-point function of $\Psi_2$ can be computed directly from the definition (47) using the two-point functions at generic $c$. At generic $c$, in addition to (38) and (40), take

$$\langle \hat{\Phi}_{3,1}(z,\bar{z})\hat{\Phi}_{3,1}(0,0)\rangle = B_{\hat{\Phi}_{3,1}}(c)\,\mathbb{P}_2^c, \qquad B_{\hat{\Phi}_{3,1}}(c) = -\frac{c^2}{4} + B_{\hat{\Phi}_{3,1}}^{(3)}c^3 + o(c^3). \tag{49}$$

We find[8]

$$\langle \Psi_2(z,\bar{z})\Psi_2(0,0)\rangle = \left(-\frac{2a^2(b - 2b_{1,2})}{b_{1,2}^2 b}\ln^2(z\bar{z}) - 2a_1\ln(z\bar{z}) + a_2\right)\mathbb{P}_2^0, \tag{50}$$

which allows us to identify the logarithmic coupling $a$ from conformal Ward identities:

$$a = \frac{b_{1,2}^2 b}{2b_{1,2} - b} = \frac{1}{4(2h'_{1,2} - h'_{3,1})h'_{3,1}} = -\frac{25}{48}. \tag{51}$$

Starting from the top field $\Psi_2$, it is a straightforward exercise to reach the field between the top and the bottom of the Jordan block by acting with the $L_0 - 2, \bar{L}_0 - 2$. There are however subtleties which we discuss in appendix C. We find the middle field $\Psi_1$

$$\begin{aligned}\Psi_1 &= \frac{a}{b_{1,2}c/2}\left(\hat{\Phi}_{3,1} - b_{1,2}h''_{3,1}c\hat{\Phi}_{3,1} - \bar{A}\hat{X} + \frac{c}{2b}\hat{\Psi}\right) \\ &= \frac{2a}{b_{1,2}c}\hat{\Phi}_{3,1} - 2ah''_{3,1}\hat{\Phi}_{3,1} - \frac{2a}{b_{1,2}c}\bar{A}\hat{X} + \frac{a}{bb_{1,2}}\hat{\Psi},\end{aligned} \tag{52}$$

and the two-point functions of $(\Psi_2, \Psi_1, \Psi_0)$ satisfy the standard form (D.10).

---

[8]The condition to cancel the $O(c^{-1})$ singularity in this two-point function fixes the $B_{\hat{\Phi}_{3,1}}^{(3)}$. See appendix B.1.

With the rank-3 Jordan block $(\Psi_2, \Psi_1, \Psi_0)$ defined in (46), (47) and (52), we can compute their three-point functions with the spin operator. Take the generic $c$ three-point functions

$$\langle \sigma(z_1,\bar{z}_1)\sigma(z_2,\bar{z}_2)T\bar{T}(z_3,\bar{z}_3)\rangle = C^{\text{cluster}}_{\sigma\sigma T\bar{T}}(c)\mathbb{P}^c_3, \tag{53a}$$

$$\langle \sigma(z_1,\bar{z}_1)\sigma(z_2,\bar{z}_2)\bar{A}\hat{X}(z_3,\bar{z}_3)\rangle = C^{\text{cluster}}_{\sigma\sigma\bar{A}\hat{X}}(c)\mathbb{P}^c_3, \tag{53b}$$

$$\langle \sigma(z_1,\bar{z}_1)\sigma(z_2,\bar{z}_2)\hat{\Psi}(z_3,\bar{z}_3)\rangle = \left(C^{\text{cluster}}_{\sigma\sigma\hat{\Psi}}(c) + C^{\text{cluster}}_{\sigma\sigma\bar{A}\hat{X}}(c)\tau_3\right)\mathbb{P}^c_3, \tag{53c}$$

$$\langle \sigma(z_1,\bar{z}_1)\sigma(z_2,\bar{z}_2)\hat{\Phi}_{3,1}(z_3,\bar{z}_3)\rangle = C^{\text{cluster}}_{\sigma\sigma\hat{\Phi}_{3,1}}(c)\mathbb{P}^c_3, \tag{53d}$$

where

$$C^{\text{cluster}}_{\sigma\sigma T\bar{T}}(c) = \left(h^{\text{cluster}}_\sigma(c)\right)^2, \tag{54}$$

and the other expressions are given in appendix B.1. We then take $c \to 0$ limit and keeping the $O(1)$ terms. We first find the singularity cancellation conditions

$$C^{\text{cluster}}_{\sigma\sigma\bar{A}\hat{X}}(0) = \left(h^{\text{perco}}_\sigma\right)^2, \qquad C^{\text{cluster}}_{\sigma\sigma\hat{\Phi}_{3,1}}(0) = \left(h^{\text{perco}}_\sigma\right)^2, \tag{55}$$

as well as

$$C'_{\sigma\sigma\hat{\Phi}_{3,1}}(0) = C_{\sigma\sigma\hat{\Psi}}(0)\,h'_{3,1} + 2h^{\text{perco}}_\sigma\left(h^{\text{perco}}_\sigma\right)', \tag{56}$$

and these are checked to hold in appendix B.1. The results are given by

$$\langle \sigma(z_1,\bar{z}_1)\sigma(z_2,\bar{z}_2)\Psi_0(z_3,\bar{z}_3)\rangle^{\text{perco}} = C^{\text{perco}}_{\sigma\sigma\Psi_0}\mathbb{P}^0_3, \tag{57a}$$

$$\langle \sigma(z_1,\bar{z}_1)\sigma(z_2,\bar{z}_2)\Psi_1(z_3,\bar{z}_3)\rangle^{\text{perco}} = \left(C^{\text{perco}}_{\sigma\sigma\Psi_1} + C^{\text{perco}}_{\sigma\sigma\Psi_0}\tau_3\right)\mathbb{P}^0_3, \tag{57b}$$

$$\langle \sigma(z_1,\bar{z}_1)\sigma(z_2,\bar{z}_2)\Psi_2(z_3,\bar{z}_3)\rangle^{\text{perco}} = \left(C^{\text{perco}}_{\sigma\sigma\Psi_2} + C^{\text{perco}}_{\sigma\sigma\Psi_1}\tau_3 + \frac{1}{2}C^{\text{perco}}_{\sigma\sigma\Psi_0}\tau_3^2\right)\mathbb{P}^0_3, \tag{57c}$$

with

$$C^{\text{perco}}_{\sigma\sigma\Psi_0} = \left(h^{\text{perco}}_\sigma\right)^2. \tag{58}$$

We do not copy the expressions of $C^{\text{perco}}_{\sigma\sigma\Psi_2}, C^{\text{perco}}_{\sigma\sigma\Psi_1}$ here, since these numbers are not intrinsic to the rank-3 Jordan block (see below). However, it is worth commenting that the agreement of $C^{\text{perco}}_{\sigma\sigma\Psi_0}, C^{\text{perco}}_{\sigma\sigma\Psi_1}$ in the three expressions of (57), that yields the standard form (D.22), serves as a consistency check of the rank-3 Jordan block construction.

As mentioned before, the bottom field $\Psi_0$ in the rank-3 Jordan block can be equivalently taken to be any of the fields $T\bar{T}, \hat{\Phi}_{3,1}, \bar{A}\hat{X}$ which are completely symmetric in this construction. This can be seen in particular from the conditions (55). This means that the second energy operator $\varepsilon' \sim \hat{\Phi}_{3,1}$ becomes degenerate with the fields $T\bar{T}, \bar{A}\hat{X}$ at $c = 0$ so in fact we discovered a rank-3 Jordan block associated with the second energy operator – a Kac operator. In the following sections, we will study Kac operators that generically have vanishing norms at $c = 0$. This also includes the energy operator $\varepsilon \sim \Phi_{2,1}$ (together with the two-hull operator $\Phi_{0,2}$) whose contribution to the generic $c$ OPE (24) has not been analyzed up to this point.

At this point, let us briefly comment on the logarithmic conformal data we have encountered. As is known and briefly reviewed in the appendix D, the logarithmic correlation functions has in fact a manifest scale dependence through the logarithm (set to unity in the above expressions) and the true scale independence of physical observables are recovered by a change of basis in the Jordan blocks. In the $c = 0$ two- and three-point functions we see above, we have explicitly computed the conformal data such as $b, a, C_{\sigma\sigma T}, C_{\sigma\sigma\Psi_0}$ which are "intrinsic" – independent of the basis in Jordan blocks. After fixing a basis (see appendix D.3, eqs. (D.46) and (D.47)), we can now write down the following non-singular OPE (operators

on the RHS are at 0):

$$\sigma^{\text{perco}}(z,\bar{z})\sigma^{\text{perco}}(0,0) = (z\bar{z})^{-2h_\sigma^{\text{perco}}}\left\{1 + \ldots + z^2\frac{h_\sigma^{\text{perco}}}{b}\Big(t + \ln(z\bar{z})T\Big) + c.c.\right. \tag{59}$$

$$+ z\bar{z}^2\frac{h_\sigma^{\text{perco}}}{2b}\partial\bar{t} + z^2\bar{z}\frac{h_\sigma^{\text{perco}}}{2b}\bar{\partial}t + (z\bar{z})^2\frac{h_\sigma^{\text{perco}}}{4b}\big(\partial^2\bar{t} + \bar{\partial}^2 t\big)$$

$$\left. + (z\bar{z})^2\frac{\big(h_\sigma^{\text{perco}}\big)^2}{a}\Big(\Psi_2 + \ln(z\bar{z})\Psi_1 + \frac{1}{2}\ln^2(z\bar{z})\Psi_0\Big) + \ldots\right\},$$

thus resolving the "$c \to 0$ catastrophe" through resolution i. Note that as identified previously in [21], there are five fields with dimensions $(2,2)$: $\partial^2\bar{t}, \bar{\partial}^2 t, \Psi_2, \Psi_1, \Psi_0$. In this OPE, we have neglected the terms involving $\Phi_{2,1}, \Phi_{0,2}$ and we will complete this in section 4.2.2 and write down the leading terms of the $s$−channel expansion the four-point function $\langle\sigma\sigma\sigma\sigma\rangle^{\text{perco}}$.

Finally, let us point out that although we have analyzed explicitly the cluster model (22), the analysis with the dilute loop model (23) is identical. In fact, as studied in [21], we find that curiously the Jordan blocks in these two cases have the same logarithmic couplings $b, a$. In the case of $b$, from the expression (32) it is clear that this should be the same for both percolation and dilute polymer, since we have the same field $X$ in the construction of $(t, T)$ pair in both cases and the logarithmic coupling is determined by the dimension of the field and the normalization. As for the log coupling $a$, in the dilute case it is given by

$$a = \frac{1}{4(2h'_{1,2} - h'_{1,5})h'_{1,5}} = -\frac{25}{48}, \tag{60}$$

which coincides with the value in the cluster model (51). The two number are however of different origins, as evident from the expression. It would be interesting to understand what this coincidence means physically.

## 2.3 Spin OPE in percolation hull

The case of spin operator in the dense loops, which describes cluster boundaries or percolation hulls, provides an example of resolution ii for the "$c \to 0$ catastrophe".

At generic $c$, dense loop spin OPE is given by:

$$\sigma^{\text{dense}}(z,\bar{z})\sigma^{\text{dense}}(0,0) = (z\bar{z})^{-2h_\sigma^{\text{dense}}(c)}$$

$$\times\left\{1 + \frac{h_\sigma^{\text{dense}}(c)}{c/2}\big(z^2 T + \bar{z}^2\bar{T}\big) + \frac{\big(h_\sigma^{\text{dense}}(c)\big)^2}{c^2/4}(z\bar{z})^2 T\bar{T} + \ldots\right.$$

$$+ D_{\sigma\sigma\Phi_{3,1}}^{\text{dense}}(c)(z\bar{z})^{h_{3,1}}\Phi_{3,1} + \ldots \tag{61}$$

$$\left. + D_{\sigma\sigma X}^{\text{dense}}(c)(z\bar{z})^{h_{1,2}(c)}\big(\bar{z}^2\bar{X} + z^2 X + \ldots\big) + \ldots\right\}.$$

Note that the logarithmic pair $(\Psi, \bar{A}X)$ in the conformal family of $X, \bar{X}$ does not couple to the dense spin operator. This can be seen from the vanishing three-point coupling $C_{\sigma\sigma A\bar{X}}^{\text{dense}}$ in terms of a finite $C_{\sigma\sigma\bar{X}}^{\text{dense}}$ as determined from Virasoro symmetry.

At $c = 0$, the conformal dimension of the spin operator vanish:

$$h_\sigma^{\text{hull}} = h_\sigma^{\text{dense}}(c = 0) = 0, \tag{62}$$

and one finds that at $c = 0$:

$$C_{\sigma\sigma T}^{\text{hull}} = 0, \qquad C_{\sigma\sigma T\bar{T}}^{\text{hull}} = 0, \tag{63}$$

where we use the superscript "hull" for the dense loop case at $c = 0$ since the spin operator in this case inserts a percolation hull. It appears that the OPE (61) is not singular at $c = 0$. Despite this, we are still faced with the issue that $T, T\bar{T}$ have vanishing norms, and are at risk of being removed from the CFT, rendering the CFT trivial. Therefore, they still necessarily acquire logarithmic partners. The construction of logarithmic operators from the cluster model apply directly, and in particular, the logarithmic couplings are the same: from the expressions (32) and (51), the two expressions only depend on the derivatives of field dimensions mixed into the Jordan block, which are the same for cluster and dense loop model.

Computing the three-point function of the spin operator with the rank-2 Jordan block, we find:

$$\langle \sigma(z_1, \bar{z}_1) \sigma(z_2, \bar{z}_2) T(z_3) \rangle^{\text{hull}} = 0 \,, \tag{64a}$$

$$\langle \sigma(z_1, \bar{z}_1) \sigma(z_2, \bar{z}_2) t(z_3, \bar{z}_3) \rangle^{\text{hull}} = C^{\text{hull}}_{\sigma\sigma t} \, \mathbb{P}^0_3 \,, \tag{64b}$$

with

$$C^{\text{hull}}_{\sigma\sigma t} = 2b \Big( \big( h^{\text{dense}}_\sigma \big)' + \big( C^{\text{dense}}_{\sigma\sigma\hat{X}} \big)' \Big) \,, \tag{65}$$

as well as the cancellation condition

$$C^{\text{dense}}_{\sigma\sigma\hat{X}}(0) = 0 \,, \tag{66}$$

which is consistent with identifying $T \sim \hat{X}$. Continuing to the rank-3 Jordan block, we find the singularity cancellation condition

$$C^{\text{dense}}_{\sigma\sigma\hat{\Phi}_{3,1}}(0) = 0 \,, \qquad C^{\text{dense}}_{\sigma\sigma\bar{A}\hat{X}}(0) = 0 \,, \tag{67}$$

as expected by the identification of $T\bar{T} \sim \hat{\Phi}_{3,1} \sim \bar{A}\hat{X}$ at $c = 0$. In addition, we need

$$\big( C^{\text{dense}}_{\sigma\sigma\hat{\Phi}_{3,1}} \big)' = 0 \,, \tag{68}$$

which can be checked to be true. The three-point functions are given by:

$$\langle \sigma(z_1, \bar{z}_1) \sigma(z_2, \bar{z}_2) \Psi_0(z_3, \bar{z}_3) \rangle^{\text{hull}} = 0 \,, \tag{69a}$$

$$\langle \sigma(z_1, \bar{z}_1) \sigma(z_2, \bar{z}_2) \Psi_1(z_3, \bar{z}_3) \rangle^{\text{hull}} = 0 \,, \tag{69b}$$

$$\langle \sigma(z_1, \bar{z}_1) \sigma(z_2, \bar{z}_2) \Psi_2(z_3, \bar{z}_3) \rangle^{\text{hull}} = C^{\text{hull}}_{\sigma\sigma\Psi_2} \, \mathbb{P}^0_3 \,, \tag{69c}$$

with (omitting the superscript $^{\text{dense}}$ for simple notation)

$$C^{\text{hull}}_{\sigma\sigma\Psi_2} = 4a \Big( \big( h'_\sigma \big)^2 + C''_{\sigma\sigma\hat{\Phi}_{3,1}} \Big) \,. \tag{70}$$

Note that all the three-point functions with the hull spin operator are non-logarithmic. This results in the following spin operator OPE for percolation hull:

$$\sigma^{\text{hull}}(z, \bar{z}) \sigma^{\text{hull}}(0, 0) = (z\bar{z})^{-2h^{\text{hull}}_\sigma} \left\{ 1 + z^2 \frac{C^{\text{hull}}_{\sigma\sigma t}}{b} T + \bar{z}^2 \frac{C^{\text{hull}}_{\sigma\sigma t}}{b} \bar{T} + (z\bar{z})^2 \frac{C^{\text{hull}}_{\sigma\sigma\Psi_2}}{a} \Psi_0 + \dots \right\}. \tag{71}$$

In this case, despite that the stress tensor $T$ and $\Psi_0 \equiv T\bar{T}$ still belong to rank-2 and rank-3 Jordan blocks (same as in percolation cluster), this OPE only probes the bottom fields and does not see the full structures of the Jordan blocks. We can directly write down the $s$-channel limit of the four-point function

$$\langle \sigma(\infty) \sigma(1) \sigma(z, \bar{z}) \sigma(0) \rangle^{\text{hull}} = 1 + \dots \,, \tag{72}$$

which contains no logarithm.

## 3 Kac operator normalizations

One important point we have stressed in the previous section is that the logarithmic mixing appears naturally as a consequence of the vanishing norms of certain operators. In the previous examples, we have started the analysis with stress tensor $T$ and the non-chiral $T\bar{T}$. Their norms are fixed by the conformal symmetry and allow an unambiguous analysis. We have also seen in the case of the rank-3 Jordan block that, Kac operator $\hat{\Phi}_{3,1}$ with a proper normalization in cluster/dense loop model and $\hat{\Phi}_{1,5}$ in dilute loop model can be equivalently chosen as the bottom fields of Jordan blocks.

In fact, the norms of Kac operators in general vanish at $c = 0$ and this is directly linked to the resolution iii of the "$c \to 0$ catastrophe" (12). The idea is the following: when the $\mathcal{O}$ in (12) is a Kac operator, its OPE at generic $c$ is severely constrained by degenerate fusion rules and the additional operator $X$ that cancels the singularity in resolution i is simply not there in the OPE. The only resolution is that Kac operators in general have vanishing norms at $c = 0$ [25, 28]. To analyze this subtle situation, we take one step back in this section and consider Kac operators in the cluster and loop models at generic $c$ to study their normalizations. We will take advantage of the fact that their contributions in four-point functions are under full analytical control as a result of the analytic bootstrap [14, 15, 22–24].

Consider in the cluster or loop model CFT the four-point function of an operator $\mathcal{O}$ [9] written in terms of the conformal block expansion:

$$\langle \mathcal{O}\mathcal{O}\mathcal{O}\mathcal{O} \rangle = \sum_{\Phi} A^{\mathcal{O}}_{\Phi} \mathcal{F}^{\mathcal{O}}_{\Phi} . \tag{73}$$

The amplitudes $A_{\Phi}$ for Kac operators $\Phi$ can be computed through a recursion relation, as a result of the Virasoro degeneracy of $\Phi_{2,1}$ in cluster/dense loop model and $\Phi_{1,2}$ in dilute loop model. More precisely, the degeneracies of $\Phi_{2,1}$ and $\Phi_{1,2}$ allow to analytically determine the amplitude recursion

$$\text{degenerate } \Phi_{2,1} \to \frac{A^{\mathcal{O}}_{\Phi_{e+1,1}}}{A^{\mathcal{O}}_{\Phi_{e-1,1}}} ,$$

$$\text{degenerate } \Phi_{1,2} \to \frac{A^{\mathcal{O}}_{\Phi_{1,e+1}}}{A^{\mathcal{O}}_{\Phi_{1,e-1}}} , \tag{74}$$

by examining the crossing equations of several four-point functions that involve $\Phi_{2,1}$ or $\Phi_{1,2}$ [14, 22–24]. We briefly summarize some of these results in appendix A. In the special case where the $\mathcal{O}$ is the spin operator in the cluster model $\sigma^{\text{cluster}}$, a stronger amplitude recursion can be obtained [15]:

$$\text{degenerate } \Phi_{2,1} \to \frac{A^{\sigma,\text{cluster}}_{\Phi_{e+1,1}}}{A^{\sigma,\text{cluster}}_{\Phi_{e,1}}} . \tag{75}$$

The grouping of the Virasoro conformal blocks through such recursions is dubbed interchiral conformal blocks in [15] and there is an underlying symmetry – the interchiral symmetry – whose algebra [32] should be thought as the continuum limit of the lattice affine-Temperley-Lieb algebra describing loop patterns in the microscopic models.

For clarity, we focus on the simplest non-trivial four-point correlation function – that of the spin operators in the cluster and loop models. Due to the recursions, the amplitudes of the Kac operators $A^{\sigma,\text{cluster}}_{\Phi}$ can be determined up to an overall constant – the amplitude of the identity operator. To fix this overall constant, we take a unit amplitude for the identity operator:

$$A^{\sigma,\text{cluster}}_{\Phi_{1,1}} = 1 , \tag{76}$$

---

[9] the correlator does not have to be of identical operators, we write identical operators here for simplicity.



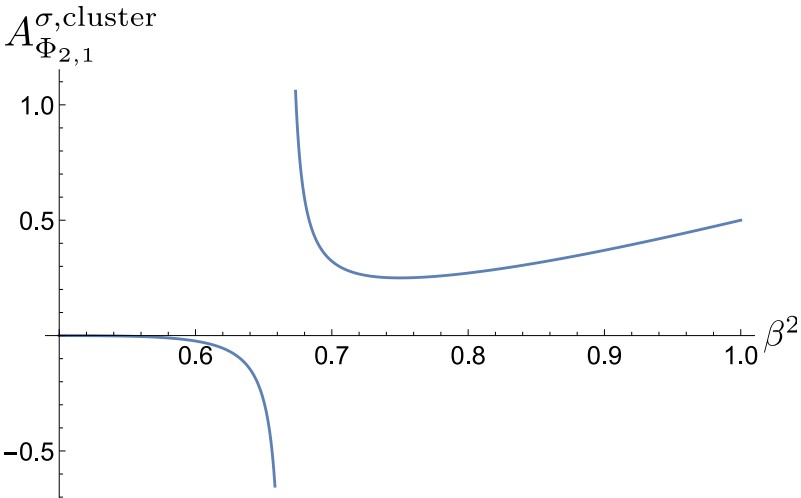

Figure 3: The amplitude $A^{\sigma,\text{cluster}}_{\Phi_{2,1}}$ of the energy operator $\Phi_{2,1}$ in cluster model four-spin correlator $\langle\sigma\sigma\sigma\sigma\rangle^{\text{cluster}}$ as a function of $0 \leq \beta^2 \leq 1$. The amplitude develops a singularity at $\beta^2 = \frac{2}{3}$ and changes sign.

which is essentially choosing the spin operator to have unit normalization. This is justified by the bootstrap results [15, 16] of cluster connectivities since this normalization has led to the consistency with the three-point connectivity [33–35].

One of the main challenges in non-unitary CFTs is that the amplitudes $A$ can be negative, preventing the use of robust positive bootstrap techniques. Moreover, when we consider CFTs that depend on a continuous parameter, as in the case of cluster and loop models, the amplitude as a function of the parameter (e.g. the central charge $c$) can develop singularities. An example of these two aspects is shown in figure 3 obtained from previous bootstrap results [15]. In this section, we attribute such negativity and singularity to the normalization of operators, and use the analytically known amplitudes for Kac operators to deduce some of these normalizations (with subtleties).

## 3.1 Real CFT operators

Recall the amplitudes $A_\Phi$ in a four-point function arise from the combination of conformal data – the three-point constant $C$ and the two-point constant $B$:

$$A^{\sigma,\text{cluster}}_{\Phi}(c) = \frac{C^2_{\sigma\sigma\Phi}(c)}{B_\Phi(c)}. \tag{77}$$

In unitary CFTs, the norms of states are positive, namely $B > 0$. It is conventional to choose $B = 1$ for unitary CFTs which does not change the sign of the norm. As a result, the positivity of $A$ is equivalent to saying that the three-point constant is real and thus

$$C^2_{\mathcal{O}_1\mathcal{O}_2\mathcal{O}_3} \geq 0. \tag{78}$$

This requirement of reality of the three-point constant is actually a particular case of the more general condition on the real QFTs [26]: in real theories, it is possible to define the so-called real operators $\mathcal{O}_i = \mathcal{O}_i^*$ which satisfy the condition:

$$\langle\mathcal{O}_1(x_1)\mathcal{O}_2(x_2)\ldots\mathcal{O}_n(x_n)\rangle^* = \langle\mathcal{O}_1^*(x_1)\mathcal{O}_2^*(x_2)\ldots\mathcal{O}_n^*(x_n)\rangle = \langle\mathcal{O}_1(x_1)\mathcal{O}_2(x_2)\ldots\mathcal{O}_n(x_n)\rangle. \tag{79}$$

For three-point functions of real operators, one has

$$\langle\mathcal{O}_1(x_1)\mathcal{O}_2(x_2)\mathcal{O}_3(x_3)\rangle^* = \langle\mathcal{O}_1(x_1)\mathcal{O}_2(x_2)\mathcal{O}_3(x_3)\rangle, \tag{80}$$

and thus the reality of the structure constant

$$C^*_{\mathcal{O}_1\mathcal{O}_2\mathcal{O}_3} = C_{\mathcal{O}_1\mathcal{O}_2\mathcal{O}_3}, \tag{81}$$

leading to (78). All unitary theories satisfy this reality condition, and in addition, they satisfy the important requirement of reflection positivity whose special case on the two-point function requires that

$$B_{\mathcal{O}} \geq 0, \tag{82}$$

and therefore the positive amplitudes.

As we move onto the vast unexplored area of non-unitary theories, not all is lost. For field theories of clusters and loops that we study here, they arise as the long-distance descriptions of random geometrical objects with real positive measure. As pointed out in [26], despite in this case, reflection positivity is absent [36], rendering the field theories non-unitary, the random cluster or loop descriptions are real in the critical regime of the models[10] and thus should correspond to real CFTs. Moreover, the CFTs are parametrized by a continuous number $c$ (equivalent to $Q$ or $n$) and the correlation functions in general carry the physical meaning of computing probability-type quantities of geometrical configurations in the scaling limit, so intuitively there is no reason to find geometrical observables to be singular at some values of the parameter $c$. This means that in the cluster and loop model CFTs, one should find real operators with real and finite three-point constants. From (77), this suggests that the negativities and divergences in the amplitudes have to be attributed to the negativities and vanishing norms $B_{\mathcal{O}}$ for real operators. Clearly, as a function of $c$, the norm $B_{\mathcal{O}}$ can cross zero. If the zero is of odd order, then the corresponding CFT state $|\mathcal{O}\rangle$ goes from a positive norm state to negative norm one. This gives rise to singularities and sign changes in the amplitudes $A$ in the four-point functions.

It is worth mentioning operators with negative norms have been uncovered in sparse examples such as the free $O(N)$ theories with non-integer $N$ [37] and Wilson-Fischer fixed point in non-integer dimensions [38]. In these cases, at particular values of the continuous parameter, the operators with zero norm decouple from the theories. The situation we have here – as already hinted in the previous section – is quite different: when the norms of the operators vanish, they do not decouple; instead, they mix into logarithmic multiplets and form Jordan blocks.

What are these real operators and how to properly normalize them? Since the reality of the theory has its origin in the microscopic descriptions, a natural thing to do is to identify these real operators from their lattice origin associated with various geometrical configurations. Such a lattice analysis exists for cluster model in a generic dimensional setting using the $S_Q$ symmetry [6, 39, 40]. The situation in 2d is however more subtle where the global symmetry is insufficient to fully characterize the geometrical correlation functions [41]. Alternatively, one can focus on the probabilistic description of the random objects directly in the continuum limit, known as the conformal loop ensembles (CLE) [42, 43]. Such probabilistic approach has made significant progress recently in computing geometrical observables in the CFTs (see e.g. [44] and the references therein) and might also shed light on the real operators and their norms. In any case, here we focus on analyzing the norms of Kac operators using the CFT intuitions. From a CFT point of view, we suspect that a proper definition of the operator norms might be given by requiring the reality and finiteness of *all* three-point constants in a given CFT, but this is out-of-reach at the moment.

---

[10]For $Q$-state Potts model related to random clusters, this corresponds $0 \leq Q \leq 4$ and for $O(n)$ loop models this corresponds to $-2 \leq n \leq 2$.

## 3.2 Kac operators in cluster model

We start by considering Kac operators

$$\Phi_{e,1}, \qquad e = 1, 2, 3, \dots, \tag{83}$$

in the four-spin correlator of the cluster model $\langle\sigma\sigma\sigma\sigma\rangle^{\text{cluster}}$. At generic $c$, the Kac operators appear uniquely in the cluster connectivity $P_{aabb}$ (fig. 1). Their amplitudes in this four-spin correlator can be uniquely obtained by the interchiral bootstrap approach [15] as a recursive product:

$$A_{\Phi_{e,1}}^{\sigma,\text{cluster}} = \prod_{i=1}^{e-1} R_{i,1}^{\text{cluster}}, \qquad e = 2, 3, \dots, \tag{84}$$

with the recursion given by:

$$R_{i,1}^{\text{cluster}} = \frac{A_{\Phi_{i+1,1}}^{\sigma,\text{cluster}}}{A_{\Phi_{i,1}}^{\sigma,\text{cluster}}} = \frac{2^{4-\frac{4i+2}{\beta^2}}\Gamma\left(\frac{1}{2}-\frac{i}{2\beta^2}\right)\Gamma\left(\frac{3}{2}-\frac{i+1}{2\beta^2}\right)\Gamma\left(\frac{i}{2\beta^2}\right)\Gamma\left(\frac{i+1}{2\beta^2}\right)}{\Gamma\left(1-\frac{i}{2\beta^2}\right)\Gamma\left(\frac{i}{2\beta^2}+\frac{1}{2}\right)\Gamma\left(1-\frac{i+1}{2\beta^2}\right)\Gamma\left(\frac{i+1}{2\beta^2}-\frac{1}{2}\right)}, \qquad i = 1, 2, \dots \tag{85}$$

As mentioned, we fix the amplitude of the identity operator $\Phi_{1,1}$ to be $A_{\Phi_{1,1}}^{\sigma,\text{cluster}} = 1$, which is consistent with the three-point connectivity [33]. The amplitudes of all the Kac operators in the four-spin correlator is thus fixed; for example, fig. 3 showed the analytic amplitude for the energy density operator $\Phi_{2,1}$ in the four-spin correlator.

Denote the properly normalized Kac operator as $\hat{\Phi}_{e,1}$, and recall that the amplitudes arise from the following conformal data

$$A_{\hat{\Phi}_{e,1}}^{\sigma,\text{cluster}} = \frac{C_{\sigma\sigma\hat{\Phi}_{e,1}}^2(c)}{B_{\hat{\Phi}_{e,1}}(c)}, \tag{86}$$

where both the numerator and denominator are continuous functions of the central charge $c$. Now, requiring $C_{\sigma\sigma\hat{\Phi}_{e,1}}^2(c)$ to be non-negative as a result of the real CFT determines the two-point structure constant $B$, i.e., the operator normalization, up to a positive function of $c$.

Inspecting the expression of the recursion relation (85), we see that the amplitude changes sign as one crosses the zeros or poles of the factor:

$$\frac{\Gamma\left(\frac{1}{2}-\frac{i}{2\beta^2}\right)\Gamma\left(\frac{3}{2}-\frac{i+1}{2\beta^2}\right)}{\Gamma\left(1-\frac{i}{2\beta^2}\right)\Gamma\left(1-\frac{i+1}{2\beta^2}\right)}, \qquad i = 1, \dots, e-1. \tag{87}$$

Let us label the zeros and poles as $\beta_{\text{zeros},i}^2$ and $\beta_{\text{poles},i}^2$. It is straightforward to see that one gets zeros and poles at

$$\begin{aligned}
\beta_{\text{zero},i}^2 &= \frac{i}{2n}, \frac{i+1}{2n}, & n \in \mathbb{N}^+, \quad \beta^2 \in \left(\frac{1}{2}, 1\right), \\
\beta_{\text{poles},i}^2 &= \frac{i}{2n-1}, \frac{i+1}{2n+1}, & n \in \mathbb{N}^+, \quad \beta^2 \in \left(\frac{1}{2}, 1\right),
\end{aligned} \tag{88}$$

and keep in mind that due to our convention (2), we only select the ones with $1/2 < \beta^2 < 1$.[11] To make the $C_{\sigma\sigma\hat{\Phi}}^2$ always positive, we have to choose the normalization of the field $\hat{\Phi}_{e,1}$ to be

$$B_{\hat{\Phi}_{e,1}} \sim \prod_{i=1}^{e-1}\left(\beta^2 - \beta_{\text{poles},i}^2\right)\left(\beta^2 - \beta_{\text{zeros},i}^2\right). \tag{89}$$

---

[11]At $\beta^2 = 1/2$, we get a double zero for each recursion factor which does not affect the positivity of $C^2$ and there are no zeros/poles of the expression at $\beta^2 = 1$.

For example, it is straightforward to find the normalization of the first two Kac operators:

$$B_{\hat{\Phi}_{2,1}} \sim \left(\beta^2 - \frac{2}{3}\right), \tag{90a}$$

$$B_{\hat{\Phi}_{3,1}} \sim \left(\beta^2 - \frac{2}{3}\right)^2 \left(\beta^2 - \frac{3}{5}\right)\left(\beta^2 - \frac{3}{4}\right). \tag{90b}$$

One can continue this to higher Kac operators, where things become more subtle: since the amplitude is given recursively in (84), a pair of zero and pole from $R_{i,1}$ and $R_{i+1,1}$ could cancel each other, so a pair of zero and pole in (89) could be removed altogether while keeping the $C^2_{\sigma\sigma\hat{\Phi}_{e,1}} \geq 0$; on the other hand, keeping both does not modify the sign of the operator norm but introduces higher order zeros in the norm, as well as adding a double zero to $C^2$ at that central charge. This happens for the field $\hat{\Phi}_{4,1}$:

$$B_{\hat{\Phi}_{4,1}} \sim \left(\textcolor{red}{\beta^2 - \frac{2}{3}}\right)^2 \left(\beta^2 - \frac{3}{5}\right)^2 \left(\beta^2 - \frac{4}{5}\right)\left(\beta^2 - \frac{4}{7}\right)\left(\beta^2 - \frac{3}{4}\right)^2\left(\textcolor{blue}{\beta^2 - \frac{2}{3}}\right), \tag{91}$$

where we have used red and blue to label respectively the vanishing factors at $c = 0$ ($\beta^2 = 2/3$) in the norm (89) deduced from poles and zeros. Here, a pole and a zero could "cancel" each other, giving rise to a simple zero at $\beta^2 = \frac{2}{3}$, i.e. $c = 0$, and still keeping $C^2_{\sigma\sigma\hat{\Phi}_{4,1}} \geq 0$. On the other hand, without this cancellation, $C^2_{\sigma\sigma\hat{\Phi}_{4,1}}$ would instead have a double zero at $c = 0$. Continuing to the next Kac operator, we get

$$\begin{aligned} B_{\hat{\Phi}_{5,1}} \sim &\left(\textcolor{red}{\beta^2 - \frac{2}{3}}\right)^2 \left(\beta^2 - \frac{3}{5}\right)^2 \left(\beta^2 - \frac{4}{5}\right)^2 \left(\beta^2 - \frac{4}{7}\right)^2 \left(\beta^2 - \frac{5}{7}\right)\left(\beta^2 - \frac{5}{9}\right) \\ &\times \left(\textcolor{blue}{\beta^2 - \frac{2}{3}}\right)^2 \left(\beta^2 - \frac{3}{4}\right)^2 \left(\beta^2 - \frac{5}{8}\right)\left(\beta^2 - \frac{5}{6}\right). \end{aligned} \tag{92}$$

Here again the two pairs of zeros and poles at $\beta^2 = \frac{2}{3}$ could annihilate each other while keeping $C^2_{\sigma\sigma\hat{\Phi}_{5,1}} \geq 0$ leading to a non-vanishing norm at $\beta^2 = \frac{2}{3}$. This however cannot be the case. It is known that Kac operators must have vanishing norms at $c = 0$ as a resolution to the $c \to 0$ catastrophe [25] (we will study this point later in section 5). This pattern of zeros/poles at $\beta^2 = \frac{2}{3}$ continues to higher Kac operators.

At the moment, by looking at the spin operator four-point function alone, we are not able to resolve this subtleties of the order of zeros in the norm of the higher Kac operators. We will comment more on this later in this section.

### 3.2.1 Vanishing norms

As we have seen in section 2, the vanishing norms of $T, T\bar{T}$ at $c = 0$ suggest logarithmic mixing. For this logarithmic mixing to happen, there has to be additional operators in the spectrum with coincidental dimensions to cancel the singularities in the OPEs. In particular, in the case of $T\bar{T}$ where the zero in the norm is second order, we actually found four fields with the same dimension to mix into a rank-3 Jordan block. In the previous subsection, we see that the norms of Kac operators in general acquire zeros at $\beta^2 = \frac{2}{3}$, i.e., $c = 0$. These vanishing norms then suggest potential logarithmic mixings into Jordan blocks in the $c = 0$ CFTs. We now consider these possibilities.

$\hat{\Phi}_{2,1}$    The simplest Kac operator beyond identity is the energy density operator $\hat{\Phi}_{2,1}$ with vanishing norm at $\beta^2 = \frac{2}{3}$, namely $c = 0$ which results in the positive $C^2_{\sigma\sigma\hat{\Phi}_{2,1}}$. In figure 4, we

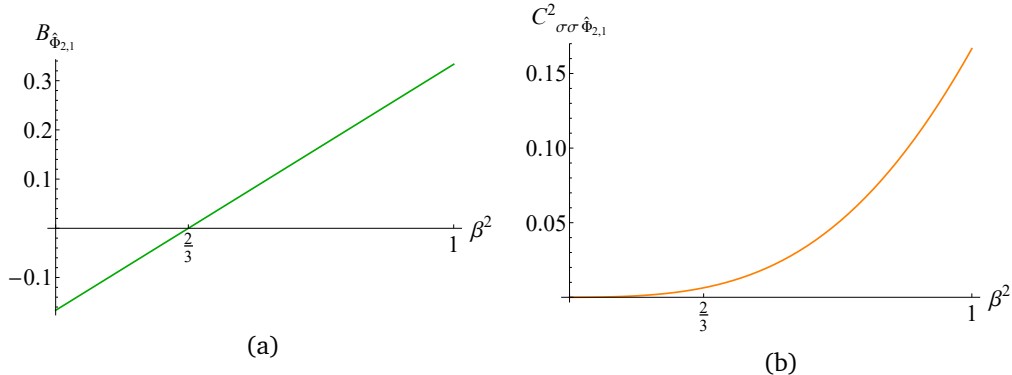

Figure 4: The deduced norm (eq. (90a)) of the operator $\hat{\Phi}_{2,1}$ (left) and the resulting positive $C^2_{\sigma\sigma\hat{\Phi}_{2,1}}$ (right). The norm has a first order zero at $\beta^2 = 2/3$, i.e. $c = 0$ where the $C^2_{\sigma\sigma\hat{\Phi}_{2,1}}$ is finite.

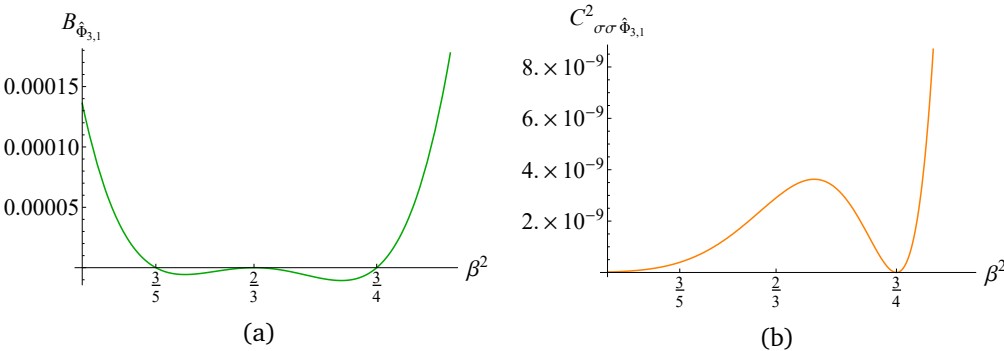

Figure 5: The deduced norm (eq. (90b)) of the operator $\hat{\Phi}_{3,1}$ (left) and the resulting positive $C^2_{\sigma\sigma\hat{\Phi}_{3,1}}$ (right). The zero at $\beta^2 = \frac{2}{3}$ is second order and suggests a logarithmic mixing into a rank-3 Jordan block as we have seen previously.

plot the expression (90a) and the resulting $C^2_{\sigma\sigma\hat{\Phi}_{2,1}}$ as a function of $\beta^2$. Keep in mind that both quantities are subject to multiplying positive functions that are not fixed from the above argument.

The zero of the norm at the percolation point hints at a logarithmic mixing. One quickly identifies the 2-hull operator $\hat{\Phi}_{0,2}$ whose conformal dimension coincides with that of $\hat{\Phi}_{2,1}$. Since there are no other operators in the spectrum with the same dimension, the Jordan block has rank 2, as suggested by the first order zero in the norm of $\hat{\Phi}_{2,1}$. Note that the norms of $\hat{\Phi}_{2,1}$ and $\hat{\Phi}_{0,2}$ should have opposite signs around $\beta^2 = \frac{2}{3}$ (or $Q = 1$), which is necessary for the logarithmic mixing.

$\hat{\Phi}_{3,1}$   From (90b), we first see that the norm of $\hat{\Phi}_{3,1}$ has a double zero at $\beta^2 = \frac{2}{3}$, i.e. $c = 0$, with a negative coefficient. This agrees with the norm we have deduced previously in (41). As we have seen there, this allows $\hat{\Phi}_{3,1}$, at $c = 0$, to mix with $T\bar{T}$ as well as a rank-2 Jordan block to form a rank-3 Jordan block. The rank of the Jordan block is related to the double zero in the norm of $\hat{\Phi}_{3,1}$. See fig. 5.

**Higher $\hat{\Phi}_{e,1}$**   As we have mentioned previously, for Kac operators $\hat{\Phi}_{e,1}$ with $e \geq 4$, subtleties arise regarding the order of zeros of their norm at $\beta^2 = \frac{2}{3}$. From the expression (91), if the pair of zero and pole annihilates, then $\hat{\Phi}_{4,1}$ would have a simple zero in its norm at $c = 0$

and according to our previous analysis, this suggests a possible rank-2 Jordan block mixing. Alternatively, we can try to keep all the zeros in the expression (91) and the norm of $\hat{\Phi}_{4,1}$ has a triple zero, suggesting a rank-4 Jordan block. At $\beta^2 = \frac{2}{3}$, we can find the coincident dimension with level 4 descendants of $\Phi_{2,2}$, which is itself a rank-2 Jordan block at generic $c$ [16, 20]. In addition, the dimension of $\hat{\Phi}_{4,1}$ also coincide with $\Phi_{0,5}$ and the level 4 descendants of $\Phi_{0,1}$ which only appear in the dense loop model spectrum.

Note that in the case where the norm of $\hat{\Phi}_{4,1}$ indeed acquires a triple zero at $c = 0$, which would suggest that $\hat{\Phi}_{4,1}$ actually sits at the bottom of a rank-4 Jordan block, the spin operator three-point structure constants $C_{\sigma\sigma\hat{\Phi}_{4,1}}$ would vanish. This means that even if $\hat{\Phi}_{4,1}$ belongs to a higher rank Jordan block, the high logarithmic structure would be invisible in the cluster spin OPE. One would have to analyze other type of correlation functions to see such potential full structure. Below in section 3.4, we will briefly come back to the issue of higher Kac operator normalizations after the comparison with $c < 1$ Liouville CFTs.

### 3.2.2 Decoupling at Ising point

It is interesting to slightly divert from the $c = 0$ case for a moment to consider the $\beta^2 = \frac{3}{4}$ Ising point which is the $Q = 2$ Potts model. In previous work [45] it was understood that the usual Ising spin four-point function can be written as a sum of the four cluster connectivities of the Potts model:

$$\langle \sigma\sigma\sigma\sigma \rangle^{\text{cluster}} \propto P_{aaaa} + P_{aabb} + P_{abba} + P_{abab}, \qquad Q = 2, \tag{93}$$

and despite that each of the connectivities involves a complicated infinite spectrum, the states that do not appear in the unitary Ising model must have their amplitudes in each connectivity intricately canceled, recovering the simple Ising CFT spectrum. While this is complicated to analyze for generic operators, the case is particularly simple for Kac operators $\hat{\Phi}_{e,1}$ since in the Potts connectivities, Kac operators appear uniquely in the connectivity $P_{aabb}$. We see first from Fig. 4 that the energy operator $\varepsilon = \hat{\Phi}_{2,1}$ indeed has a positive norm at the Ising point, with non-vanishing three-point coupling $C_{\sigma\sigma\hat{\Phi}_{2,1}}$, consistent with the unitarity of the theory. For the Kac operator $\hat{\Phi}_{3,1}$, the zero norm in this case actually indicates the decoupling of the operator,[12] and in the meantime, the three-point constant $C_{\sigma\sigma\hat{\Phi}_{3,1}}$ vanishes. We recover the well-known fusion rule in the 2d Ising model:

$$\sigma \times \sigma \sim \mathbb{I} + \varepsilon. \tag{94}$$

Similar analysis can be done for degenerate fusion rule

$$\hat{\Phi}_{2,1} \times \hat{\Phi}_{2,1} \sim \mathbb{I} + \hat{\Phi}_{3,1}, \tag{95}$$

where (see eq. (A.22))

$$\frac{C^2_{\hat{\Phi}_{2,1}\hat{\Phi}_{2,1}\hat{\Phi}_{3,1}}}{B^2_{\hat{\Phi}_{2,1}} B_{\hat{\Phi}_{3,1}}} = -\frac{\Gamma\left(2 - \frac{2}{\beta^2}\right)^2 \Gamma\left(\frac{3}{\beta^2} - 1\right) \Gamma\left(\frac{1}{\beta^2}\right)}{\Gamma\left(2 - \frac{3}{\beta^2}\right) \Gamma\left(\frac{2}{\beta^2}\right)^2 \Gamma\left(\frac{\beta^2-1}{\beta^2}\right)} \overset{\beta^2 \to \frac{3}{4}}{\simeq} \mathcal{O}\left(\beta^2 - \frac{3}{4}\right), \tag{96}$$

recovering the fusion rule

$$\varepsilon \times \varepsilon \sim \mathbb{I}, \tag{97}$$

at the Ising point. In this case, the second energy operator $\varepsilon' = \hat{\Phi}_{3,1}$ decouples from the unitary sector of Ising CFT.

---

[12]We do not see other operators with the same dimension so the zero norm would indicate a true decoupling instead of logarithmic mixing.

## 3.3 Kac operators in loop models

Similar to the cluster model, we can consider the amplitudes of Kac operators in the four-spin correlators in loop models, which corresponds to the polymer line configuration in fig. 1. Keep in mind that the spin operator is given by Kac indices $(0, \frac{1}{2})$ in the dense loop model and $(\frac{1}{2}, 0)$ in dilute and cluster models. In the loop models, the spin operator inserts and absorbs a polymer line, as opposed to the case of the cluster model where it inserts and absorbs a cluster. In the dense loop case, such lines can be thought of as the cluster boundaries (that end on the dense loop spin operator). Therefore, even if we consider the same Kac operator (say $\Phi_{3,1}$), its amplitudes would not be the same in dense loop four-spin correlator and cluster four-spin correlator, since the geometrical objects are different. By normalizing the identity operator to have amplitude 1 (this is equivalent to normalizing the loop model spin operator to have unit normalization), we can obtain the amplitudes for all degenerate operators:

$$
\begin{aligned}
\text{dense loop:} \quad & \Phi_{1,1}, \Phi_{3,1}, \Phi_{5,1}, \dots, \\
\text{dilute loop:} \quad & \Phi_{1,1}, \Phi_{1,3}, \Phi_{1,5}, \dots
\end{aligned}
\tag{98}
$$

Their amplitudes can be obtained by the following recursion where in the dense case:

$$
R_i^{\text{dense}} = \frac{A_{\Phi_{i+1,1}}^{\sigma,\text{dense}}}{A_{\Phi_{i-1,1}}^{\sigma,\text{dense}}} = -\frac{2^{8\left(1-\frac{i}{\beta^2}\right)}\Gamma\left(\frac{1-i}{\beta^2}+1\right)\Gamma\left(\frac{i}{\beta^2}-1\right)^2\Gamma\left(2-\frac{i+1}{\beta^2}\right)}{\Gamma\left(1-\frac{i}{\beta^2}\right)^2\Gamma\left(\frac{i+1}{\beta^2}-1\right)\Gamma\left(\frac{i-1}{\beta^2}\right)}, \quad i = 2,4,6,\dots, \tag{99}
$$

and in the dilute case:

$$
R_i^{\text{dilute}} = \frac{A_{\Phi_{1,i+1}}^{\sigma,\text{dilute}}}{A_{\Phi_{1,i-1}}^{\sigma,\text{dilute}}} = -\frac{2^{8-8\beta^2 i}\Gamma\left((1-i)\beta^2+1\right)\Gamma\left(i\beta^2-1\right)^2\Gamma\left(2-(i+1)\beta^2\right)}{\Gamma\left((i-1)\beta^2\right)\Gamma\left(1-i\beta^2\right)^2\Gamma\left((i+1)\beta^2-1\right)}, \quad i = 2,4,6,\dots \tag{100}
$$

Keep in mind that in the dilute case the recursion arises from the degeneracy of $\Phi_{1,2}$ and in the dense case the recursion arises from the degeneracy of $\Phi_{2,1}$ – the same as that of the cluster model. Let us now follow the same idea as the previous section and see what this tells us about the normalization of Kac operators.

For dense loop model, the expression (99) can be analyzed in the same way as before. One finds poles at:

$$
\beta^2_{\text{poles},i} = \frac{i-1}{n}, \frac{i+1}{n+1}, \qquad n \in \mathbb{N}^+, \quad \beta^2 \in \left(\frac{1}{2}, 1\right), \tag{101}
$$

where keep in mind we have $i = 2,4,\dots$ here. These poles completely take care of the sign and one can simply take the norms for the Kac operator $\hat{\Phi}$ to be $\prod_i(\beta^2 - \beta^2_{\text{poles},i})$ to make the $C^2$ non-negative. Note that the expression also has double zeros at

$$
\beta^2_{\text{zeros},i} = \frac{i}{n}, \qquad n \in \mathbb{N}^+, \quad \beta^2 \in \left(\frac{1}{2}, 1\right), \tag{102}
$$

indicating the vanishing three-point constants with the spin operator.

Consider for example $\hat{\Phi}_{3,1}$, i.e. the energy operator in the dense loop model. We see that to guarantee $C^2_{\sigma\sigma\hat{\Phi}_{3,1}} \geq 0$, it suffices to have the norm

$$
B_{\hat{\Phi}_{3,1}} \sim \left(\beta^2 - \frac{3}{4}\right)\left(\beta^2 - \frac{3}{5}\right), \tag{103}
$$

coming from the poles (101). On the other hand, it is also possible to include a double zero at $\beta^2 = \frac{2}{3}$ from (102) into the norm while keeping $C^2 \geq 0$. The simplest possibility (103)

clearly cannot be true. As pointed out in [25] and will be explained in section 5, Kac operators must have vanishing norms at $c = 0 (\beta^2 = \frac{2}{3})$ as resolution iii of the "$c \to 0$ catastrophe" (12). Therefore, the norm of $\hat{\Phi}_{3,1}$ must also include at least a double zero at $\beta^2 = \frac{2}{3}$:

$$B_{\hat{\Phi}_{3,1}} \sim \left(\beta^2 - \frac{3}{4}\right)\left(\beta^2 - \frac{3}{5}\right)\left(\beta^2 - \frac{2}{3}\right)^2. \tag{104}$$

The normalization (104) is consistent with our analysis in section 2 where $\hat{\Phi}_{3,1}$ is identified with $T\bar{T}$ as the bottom field of the rank-3 Jordan block. The three-point constant with the spin operator $C^{\text{dense}}_{\sigma\sigma\hat{\Phi}_{3,1}}$, in this case, has a double zero at $c = 0$, consistent with eqs. (67), (68). It is interesting to note that the normalization of the operator $\hat{\Phi}_{3,1}$ now have identical analytic structure in the cluster CFT (eq. (90b)) that describes percolation clusters and in the dense loop one (eq. (104)) that describes the percolation hulls.

Let us summarize the case of dilute loop model. From the expression (100), we get poles at

$$\beta^2_{\text{poles},i} = \frac{n}{i-1}, \frac{n+1}{i+1}, \qquad n \in \mathbb{N}^+, \quad \beta^2 \in \left(\frac{1}{2}, 1\right), \tag{105}$$

and in addition, the amplitude recursion has double zeros at

$$\beta^2_{\text{zeros},i} = \frac{n}{i}, \qquad n \in \mathbb{N}^+, \quad \beta^2 \in \left(\frac{1}{2}, 1\right). \tag{106}$$

This allows us to write down the norm for the energy operator $\varepsilon \sim \hat{\Phi}_{1,3}$ of dilute loop model:

$$B_{\hat{\Phi}_{1,3}} \sim \left(\beta^2 - \frac{2}{3}\right). \tag{107}$$

This first order zero for the energy operator $\hat{\Phi}_{1,3}$ at $\beta^2 = \frac{2}{3}$ suggests a mixing into rank-2 Jordan block as we will see in section 4.2 for details. For the second energy operator $\varepsilon' \sim \hat{\Phi}_{1,5}$, to make $C^2 \geq 0$, it suffices to have

$$B_{\hat{\Phi}_{1,5}} \sim \left(\beta^2 - \frac{2}{3}\right)^2\left(\beta^2 - \frac{3}{5}\right)\left(\beta^2 - \frac{4}{5}\right), \tag{108}$$

where the factors come from (105). Similar to the dense loop case above, one could also include a double zero from (106) at $\beta^2 = 3/4$ in the norm which would mean:

$$B_{\hat{\Phi}_{1,5}} \sim \left(\beta^2 - \frac{2}{3}\right)^2\left(\beta^2 - \frac{3}{5}\right)\left(\beta^2 - \frac{3}{4}\right)^2\left(\beta^2 - \frac{4}{5}\right). \tag{109}$$

This ambiguity can be resolved in the following way. Recall that at $c = 1/2 (\beta^2 = 3/4)$, the dilute $O(n = 1)$ loop model becomes the Ising model. The spin operator algebra we are looking at must reduce to the unitary Ising model OPE, similar to the analysis in section 3.2.2 for cluster model. The second energy operator $\varepsilon' \sim B_{\hat{\Phi}_{1,5}}$ should then become a zero norm state and decouple from the unitary sector, which suggests the normalization (109) is the correct one. Note also that the norm of $\hat{\Phi}_{1,5}$ has a double zero at $c = 0$ ($\beta^2 = 2/3$) and indeed as we have commented at the end of section 2.2.2, the operator mixes into the rank-3 Jordan block for SAW. We plot in Fig. 6 the norms (107) and (109). Finally, it is curious to notice that the analytic structure of the norms of the energy operator for cluster and dilute loop model are the same, and this suggests that there might be a first principles approach to understand these normalizations.

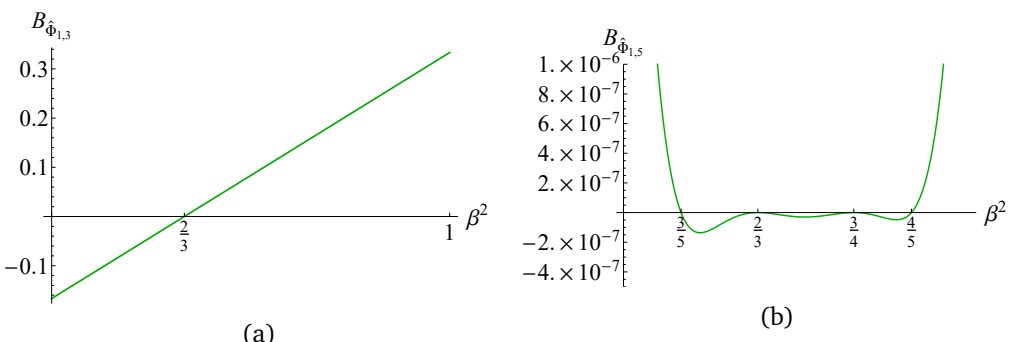

Figure 6: The deduced norms (eqs. (107) and (109)) of the energy operator $\varepsilon \sim \hat{\Phi}_{1,3}$ (left) and second energy operator $\varepsilon' \sim \hat{\Phi}_{1,5}$ (right) in dilute loop model.

### 3.4  Comparison with $c < 1$ Liouville CFT

So far we have been analyzing the diagonal Kac operators completely within cluster and loop model CFTs. This in particular means that we have used the analytic amplitudes that arise from either the degeneracy of $\Phi_{2,1}$ in the cluster/dense loop model or that of $\Phi_{1,2}$ in the dilute loop model, but not both. It was however observed in recent work [46] that the analytic conformal data of diagonal fields in loop models formally coincided with that of the $c < 1$ Liouville CFT. Note that the $c < 1$ Liouville CFT contains degeneracies of both $\Phi_{2,1}$ and $\Phi_{1,2}$ which led to a full analytic solution [22, 23]. Despite the physics behind this coincidence is unclear (e.g. the $c < 1$ Liouville CFT has a continuum spectrum and the loop models have discrete ones), it is instructive to compare the Kac operator normalizations we deduced in the previous subsections with the analytic known expressions in $c < 1$ Liouville CFTs.

The two-point functions for diagonal operators with momentum $P$ in $c < 1$ Liouville CFTs are given through the Barnes' double Gamma function $\Gamma_\beta$ as

$$B_P^L = \prod_{\pm\pm} \frac{1}{\Gamma_\beta(\beta^{\pm 1} \pm 2P)}, \tag{110}$$

where we recall that the Liouville momentum associated with Kac indices $P_{r,s}$ is given by

$$P_{r,s} = \frac{1}{2}\left(\frac{r}{\beta} - s\beta\right). \tag{111}$$

We can then check the Liouville norm of $P_{r,1}$ for the Kac operators in cluster/dense loop models and $P_{1,s}$ for dilute loop models and compare with our analysis.

In Fig. 7, we plot the Liouville norm for $P_{2,1}, P_{3,1}, P_{1,3}, P_{1,5}$. These are to be compared with the normalizations of $\hat{\Phi}_{2,1}, \hat{\Phi}_{3,1}, \hat{\Phi}_{1,3}, \hat{\Phi}_{1,5}$ in the cluster and loop models we deduced above – see eqs. (90a), (90b), (107) and (109) as well as Figs. 4a, 5a, 6a and 6b. We see that the analytic structure of the norms for $1/2 < \beta^2 < 1$ deduced from cluster and loop models fully agree with that of the $c < 1$ Liouville CFTs with the same momenta. The Liouville norms appear to have additional zeros at $\beta^2 = 1/2, 1$ which we do not consider.

This agreement is suggestive and it is interesting to further check the Liouville norm for momenta corresponding to higher Kac operators, in particular $P_{4,1}$ and $P_{5,1}$ that we have discussed at the end of subsection 3.2. In Figs. 8 and 9, we show the Liouville normalizations for momenta $P_{4,1}, P_{5,1}$. It is fascinating to notice that at $\beta^2 = \frac{2}{3}$, $B^L(P_{4,1})$ has a third order zero and $B^L(P_{5,1})$ has a fourth order zero. If the Kac operator norms in cluster model indeed coincide with that of $c < 1$ Liouville CFTs, as we have seen above, this would imply that the percolation CFT contains rank-4, rank-5, and even arbitrarily higher rank Jordan blocks whose bottom

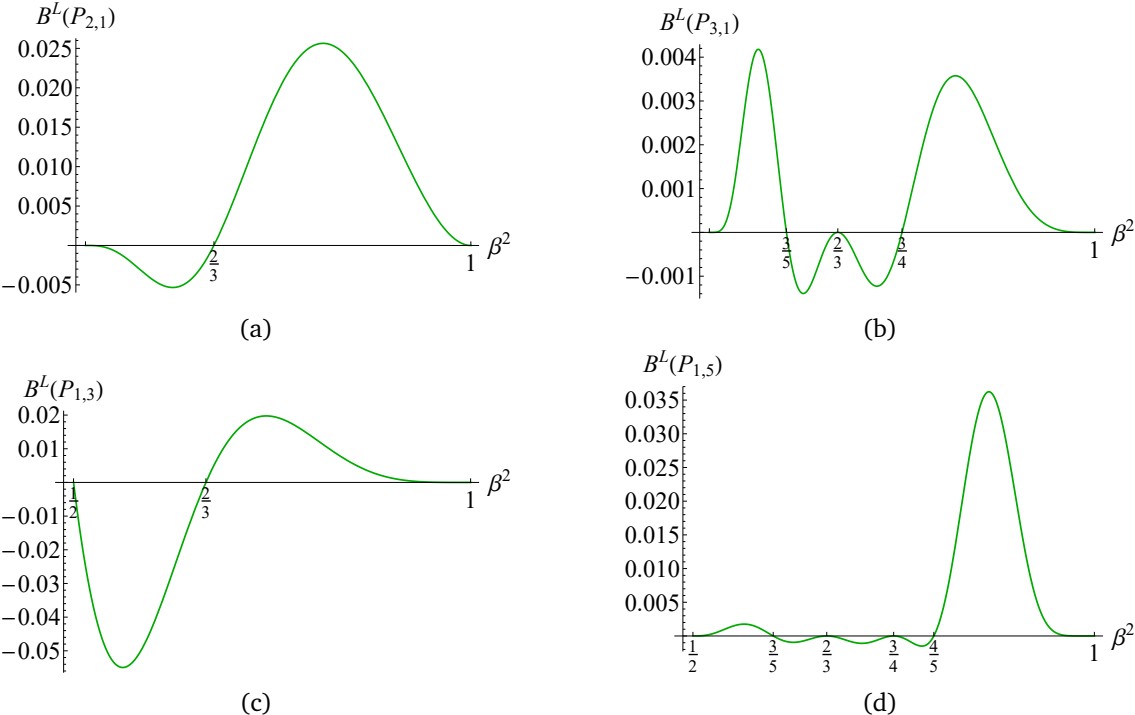

Figure 7: Normalizations $B^L(P)$ of diagonal operators in $c < 1$ Liouville CFTs with momenta $P_{2,1}, P_{3,1}, P_{1,3}, P_{1,5}$ – to be compared with the normalizations of Kac operators $\hat{\Phi}_{2,1}, \hat{\Phi}_{3,1}, \hat{\Phi}_{1,3}, \hat{\Phi}_{1,5}$ in cluster and loop models.

fields are identified with these Kac operators. The existence of arbitrary higher rank Jordan block was suggested in an earlier work [47] from a lattice algebraic analysis, but has yet to be seen in the CFT context. Moreover, as we have mentioned at the end of section 3.2.1, a mixing for higher Kac operators into higher rank Jordan blocks would require invoking operators that only appear in dense loop CFTs, implying that there might be a bigger CFT including both the percolation and percolation hulls. In any case, as we commented before, such additional zeros in the norms of Kac operators $\hat{\Phi}_{r,1}, r \geq 4$ would render the three-point couplings $C_{\sigma\sigma\hat{\Phi}_{r,1}}$ vanish so the potential higher logarithmic structure cannot be seen from the spin operator algebra in percolation, and requires studying more complicated operators and correlation functions. We will comment on such possibilities in the conclusions.

## 4 Constructing logarithmic operators

We have seen above that the norms of Kac operators acquire zeros at rational central charges, $c = 0$ in particular. As pointed out in [25], at $c = 0$, Kac operators appear generically at the bottom of Jordan blocks and the orders of zero in the norm suggest the ranks of Jordan block. This allows a general scheme for constructing logarithmic operators at $c = 0$ using Kac operators.

In this section, we study this generic construction. It is worth pointing out that some similar constructions have been done before in the chiral case for rank-2 Jordan blocks, e.g. [8], although no Jordan blocks with rank higher than two was found in the chiral CFTs. While the construction here is a bit formal, we will see that the somewhat mysterious logarithmic couplings – especially in the case of higher rank Jordan blocks that only appears in bulk theories – can be obtained directly from the norms and dimensions of the operators.

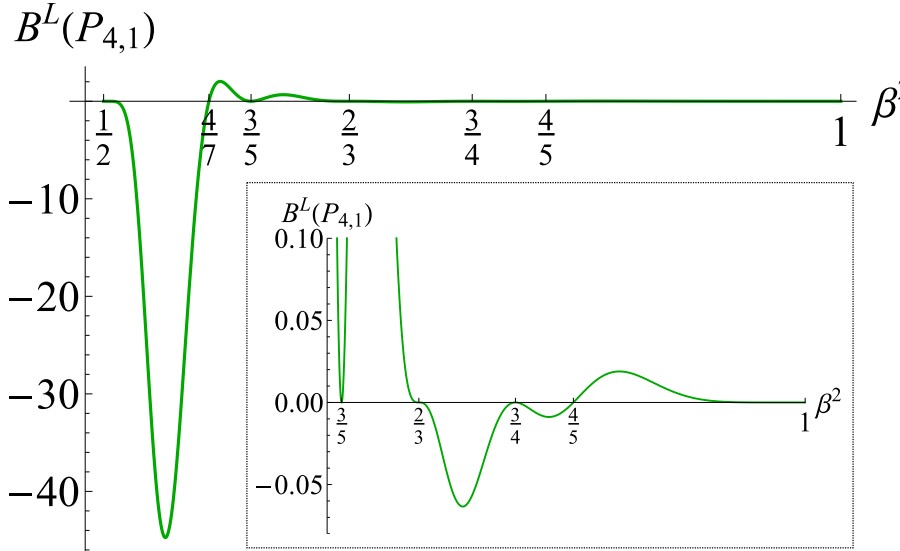

Figure 8: Normalization of operator with momentum $P_{4,1}$ in $c < 1$ Liouville CFT. The norm has a third order zero at $\beta^2 = \frac{2}{3}$, i.e. $c = 0$.

## 4.1 Rank-2 Jordan block

We start with the case of rank-2 Jordan block. Consider two properly normalized operators $\phi$ and $\psi$ and one of them, say $\phi$, could be the Kac operator whose norm acquires a zero at rational central charge $c_*$. In the meantime the other operator $\psi$ also has a zero in its norm with derivative of the opposite sign:

$$\langle \phi(z,\bar{z})\phi(0,0)\rangle = B_\phi(c)\mathbb{P}_2^c \simeq B_\phi'(c-c_*)\mathbb{P}_2^* + o(c-c_*), \tag{112a}$$

$$\langle \psi(z,\bar{z})\psi(0,0)\rangle = B_\psi(c)\mathbb{P}_2^c \simeq B_\psi'(c-c_*)\mathbb{P}_2^* + o(c-c_*), \tag{112b}$$

with

$$B_\phi' = -B_\psi', \tag{113}$$

where $B'$ indicates the derivative of the norm $B(c)$ evaluated at $c = c_*$. Here we have used $\mathbb{P}_2^*$ to denote the expression of (6) at $c = c_*$ and when $c = 0$ this is $\mathbb{P}_2^0$. We can now define the top field of the rank-2 Jordan block as[13]

$$\mathcal{O}^{(2;2)} = \gamma\left(\frac{\phi}{B_\phi(c)} + \frac{\psi}{B_\psi(c)}\right), \tag{114}$$

and the bottom field as

$$\mathcal{O}^{(2;1)} = \phi \text{ or } \psi, \tag{115}$$

where we use $(2;1),(2;2)$ to denote the first and second operators in a rank-2 Jordan block (from bottom up). By definition we find

$$\langle \mathcal{O}^{(2;2)}(z,\bar{z})\mathcal{O}^{(2;1)}(0,0)\rangle = \gamma\mathbb{P}_2^*, \tag{116}$$

and clearly we have

$$\langle \mathcal{O}^{(2;1)}(z,\bar{z})\mathcal{O}^{(2;1)}(0,0)\rangle = 0. \tag{117}$$

---

[13]As commented in footnote 6, the definition of the operator here implicitly includes a scale $\mu$ which enters the logarithm in the correlation functions and we have taken $\mu = 1$ throughout.

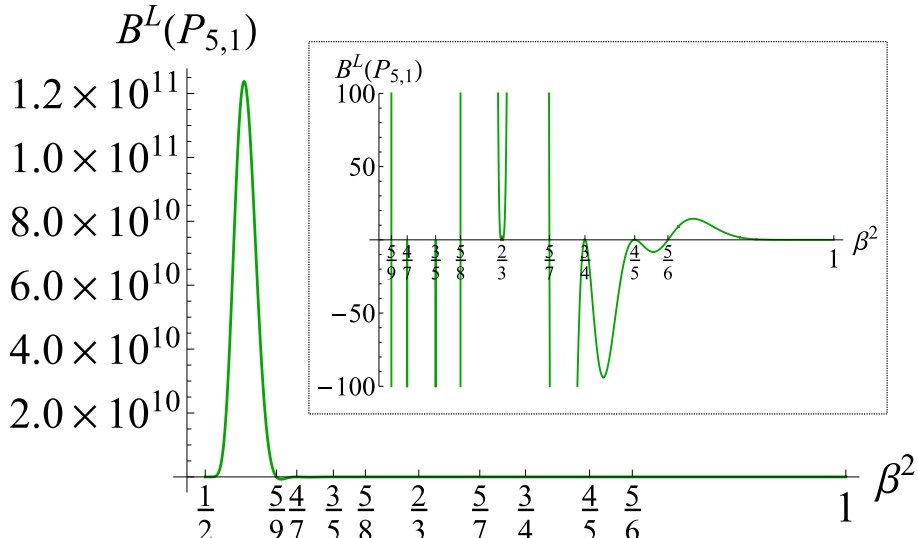

Figure 9: Normalization of operator with momentum $P_{5,1}$ in $c < 1$ Liouville CFT. The norm has a fourth order zero at $\beta^2 = \frac{2}{3}$, i.e. $c = 0$.

Eq. (116) defines the logarithmic coupling $\gamma$. Using the definition (114), we can compute the two-point function of $\mathcal{O}^{(2;2)}$ using the generic $c$ two-point functions of $\phi, \psi$ and taking the limit $c \to c^*$. Keeping the finite term, we find

$$\langle \mathcal{O}^{(2;2)}(z,\bar{z}) O^{(2;2)}(0,0) \rangle = \left( -2\gamma^2 \frac{h'_\phi - h'_\psi}{B'_\phi} \ln(z\bar{z}) + const. \right) \mathbb{P}_2^* = \left( -2\gamma \ln(z\bar{z}) + const. \right) \mathbb{P}_2,$$
(118)

where the last equality comes from conformal Ward identities (D.9). In arriving (118), the condition (113) is important for canceling a leading singularity at $O((c - c_*)^{-1})$. Eq. (118) fixes the logarithmic coupling $\gamma$ to be

$$\gamma = \frac{B'_\phi}{h'_\phi - h'_\psi} .$$
(119)

Note that the logarithmic coupling is completely determined by the normalizations and the conformal dimensions as functions of the parameter $c$. Note also that the expression of $\gamma$ is symmetric with respect to the field $\phi$ and $\psi$ considering (113).

This generic construction can be used immediately on the logarithmic pair of the stress-energy tensor, although this does not involve Kac operators. In that case, one can take for example:

$$\phi : T, \qquad \psi : \hat{X},$$
(120)

with the normalization given in (25), namely $B'_\phi = \frac{1}{2}$ where $c_* = 0$. Since the dimension of stress-energy tensor $T$ is fixed to be $h_T = 2$, (119) gives us

$$b = \frac{1}{-2h'_X} ,$$
(121)

as in (32).

## 4.2 Energy density logarithmic pair

An important, and also the simplest example of rank-2 Jordan block built out of Kac operator at $c = 0$ is the case of the energy operator which in cluster and dilute loop models are given

by

$$\varepsilon = \begin{cases} \hat{\Phi}_{2,1}, & \text{cluster}, \\ \hat{\Phi}_{1,3}, & \text{dilute loop}. \end{cases} \tag{122}$$

At generic $c$, they appear in the spin operator OPE (24), together with the hull operators $\Phi_{0,2}$ and $\Phi_{1,0}$ in cluster and dilute loop models respectively. In both cases, we have seen from the previous section that these Kac operators acquire a first order zero in their norms at $c_* = 0$ for percolation and SAW. In the meantime, one finds the coincidental dimensions of the energy operator with the hull operators:

$$\begin{aligned} \text{percolation:} \quad (h_{2,1}, h_{2,1}) &= (h_{0,2}, h_{0,2}) = \left(\frac{5}{8}, \frac{5}{8}\right), \\ \text{SAW:} \quad (h_{1,3}, h_{1,3}) &= (h_{1,0}, h_{1,0}) = \left(\frac{1}{3}, \frac{1}{3}\right). \end{aligned} \tag{123}$$

We expect logarithmic mixings into rank-2 Jordan blocks in both cases. Early works [5,6] have analyzed these Jordan blocks based on global symmetry considerations. Here we see that they arise from the previous normalizations of Kac operators. Take the following normalization for the energy operator:[14]

$$\langle \varepsilon(z,\bar{z})\,\varepsilon(0,0)\rangle \overset{c\to 0}{\simeq} \frac{c}{2}\mathbb{P}_2^c, \tag{124}$$

for both cluster and dilute loop models, namely $B'_* = \frac{1}{2}$. This suggests the following behavior of the two-point functions of the hull operators:

$$\begin{aligned} \langle \hat{\Phi}_{0,2}(z,\bar{z})\,\hat{\Phi}_{0,2}(0,0)\rangle &\overset{c\to 0}{\simeq} -\frac{c}{2}\mathbb{P}_2^c, \\ \langle \hat{\Phi}_{1,0}(z,\bar{z})\,\hat{\Phi}_{1,0}(0,0)\rangle &\overset{c\to 0}{\simeq} -\frac{c}{2}\mathbb{P}_2^c. \end{aligned} \tag{125}$$

According to (114), we can construct the following top fields for the Jordan blocks

$$\tilde{\varepsilon} = \begin{cases} \gamma\left(\dfrac{\hat{\Phi}_{2,1}}{B_{\hat{\Phi}_{2,1}}(c)} + \dfrac{\hat{\Phi}_{0,2}}{B_{\hat{\Phi}_{0,2}}(c)}\right), & \text{percolation}, \\[3mm] \gamma\left(\dfrac{\hat{\Phi}_{1,3}}{B_{\hat{\Phi}_{1,3}}(c)} + \dfrac{\hat{\Phi}_{1,0}}{B_{\hat{\Phi}_{1,0}}(c)}\right), & \text{SAW}. \end{cases} \tag{126}$$

Using (119), we find the following logarithmic couplings for the energy operators:

$$\gamma^{\text{perco}} = \frac{1}{2(h'_{2,1} - h'_{0,2})} = -\frac{5}{4}, \tag{127a}$$

$$\gamma^{\text{SAW}} = \frac{1}{2(h'_{1,3} - h'_{1,0})} = \frac{5}{3}. \tag{127b}$$

Curiously, these are twice the logarthmic coupling – the $b$-number – in the boundary/chiral case [2,8].

### 4.2.1 Comparison with [6] for percolation

As a check, let us compare the construction in the case of percolation with earlier work [6] based on $S_Q$ symmetry considerations. The energy operator in [6] has the two-point function at generic $c$: (neglecting the tensor indices)

$$\langle \varepsilon(z,\bar{z})\,\varepsilon(0,0)\rangle = \frac{\tilde{A}(Q)(Q-1)}{(z\bar{z})^{2h_\varepsilon(c)}}, \tag{128}$$

---

[14]We choose a constant factor $1/2$ here, motivated by the fact that in the chiral case, the energy operator coincide with the stress tensor which has such a constant factor.

which vanishes at $Q = 1$, i.e., $c = 0$. Using

$$\sqrt{Q} = 2\cos(\pi\beta^2), \tag{129}$$

and comparing our definition (124) with (128) we can fix the normalization of (128) to be

$$\tilde{A}(1) = \frac{5\sqrt{3}}{8\pi}. \tag{130}$$

The logarithmic operator in [6] (i.e., top field in the Jordan block) was defined as (ignoring the $S_Q$ tensor indices)

$$\tilde{\psi} = \hat{\psi} + \frac{2}{Q(Q-1)}\varepsilon, \tag{131}$$

with $\hat{\psi}$ playing the similar role as $\hat{\Phi}_{0,2}$ and the logarithmic two-point function of $\tilde{\psi}$ is:

$$\langle \tilde{\psi}(z,\bar{z})\tilde{\psi}(0,0)\rangle = 2\tilde{A}(1)\frac{\frac{2\sqrt{3}}{\pi}\ln(z\bar{z}) + const.}{(z\bar{z})^{\frac{5}{4}}}. \tag{132}$$

Now, to make a proper comparison of the logarithmic couplings, we need to specify the difference in the normalizations between $\tilde{\psi}$ and our logarithmic operator $\tilde{\varepsilon}$ in (126), and this can be done by comparing the coefficient in front of $\varepsilon$ in the definitions (126) and (131). We find

$$\tilde{\psi} \sim \frac{5\sqrt{3}}{4\pi\gamma}\tilde{\varepsilon}. \tag{133}$$

The means we should have the logarithmic couplings in (132) and (127a) to match

$$\frac{4\sqrt{3}}{\pi}\tilde{A}(1) = \left(\frac{5\sqrt{3}}{4\pi\gamma}\right)^2(-2\gamma), \tag{134}$$

which can be easily checked to hold using (130).

### 4.2.2 In the spin operator OPE

At this point, we can go back to the spin operator OPE (24) and consider the contributions from the energy operator $\hat{\Phi}_{2,1}$ and 2-hull operator $\hat{\Phi}_{0,2}$ in cluster model (and similarly in the dilute loop model case) and analyze their $c \to 0$ limit. Taking percolation for example, the top field is defined in (126) as

$$\tilde{\varepsilon}^{\text{perco}} = \gamma^{\text{perco}}\left(\frac{\hat{\Phi}_{2,1}}{B_{\hat{\Phi}_{2,1}}(c)} + \frac{\hat{\Phi}_{0,2}}{B_{\hat{\Phi}_{0,2}}(c)}\right), \tag{135}$$

and the normalization $B_{\hat{\Phi}_{2,1}}$ of the bottom field – the energy operator $\varepsilon \sim \hat{\Phi}_{2,1}$ – takes the form (124):

$$B_{\hat{\Phi}_{2,1}}(c) \simeq \frac{c}{2} + o(c). \tag{136}$$

Using the analytic amplitude (84) at generic $c$, we find

$$\langle \sigma(z_1,\bar{z}_1)\sigma(z_2,\bar{z}_2)\varepsilon(z_3,\bar{z}_3)\rangle^{\text{perco}} = C_{\sigma\sigma\varepsilon}^{\text{perco}}\mathbb{P}_3^0, \tag{137}$$

where the three-point constant is given by:

$$C_{\sigma\sigma\varepsilon}^{\text{perco}} = \frac{1}{2}\sqrt{\frac{5}{6}\frac{\Gamma(3/4)}{\Gamma(1/4)}}. \tag{138}$$

Furthermore, taking the three-point function at generic $c$:

$$\langle \sigma(z_1,\bar{z}_1)\sigma(z_2,\bar{z}_2)\hat{\Phi}_{0,2}(z_3,\bar{z}_3)\rangle^{\text{cluster}} = C^{\text{cluster}}_{\sigma\sigma\hat{\Phi}_{0,2}}(c)\mathbb{P}_3^0, \tag{139}$$

one finds

$$\langle \sigma(z_1,\bar{z}_1)\sigma(z_2,\bar{z}_2)\tilde{\varepsilon}(z_3,\bar{z}_3)\rangle^{\text{perco}} = \left(C^{\text{perco}}_{\sigma\sigma\tilde{\varepsilon}} + C^{\text{perco}}_{\sigma\sigma\varepsilon}\tau_3\right)\mathbb{P}_3^0, \tag{140}$$

where singularity cancellation requires

$$C^{\text{cluster}}_{\sigma\sigma\hat{\Phi}_{0,2}}(c=0) = C^{\text{perco}}_{\sigma\sigma\varepsilon}. \tag{141}$$

This can be checked using the bootstrap results from [15] which we show in Fig. 11. Note once again that the logarithmic mixing (e.g. from the cancellation condition (141)) suggests that the energy operator $\varepsilon$ and the 2-hull operator $\hat{\Phi}_{0,2}$ essentially become degenerate at $c=0$ and can be equivalently taken as the bottom field in the rank-2 Jordan block. Using (137) and (140), we can now write down the contributions of the energy operator logarithmic pair $(\tilde{\varepsilon},\varepsilon)$ to the percolation cluster spin OPE (59):

$$\sigma^{\text{perco}}(z,\bar{z})\sigma^{\text{perco}}(0,0) = \ldots + (z\bar{z})^{-2h_\sigma+h_\varepsilon}\frac{C^{\text{perco}}_{\sigma\sigma\varepsilon}}{\gamma}\left(\tilde{\varepsilon}^{\text{perco}} + \ln(z\bar{z})\varepsilon^{\text{perco}} + \frac{C^{\text{perco}}_{\sigma\sigma\tilde{\varepsilon}}}{C^{\text{perco}}_{\sigma\sigma\varepsilon}}\varepsilon^{\text{perco}}\right) + \ldots, \tag{142}$$

and the $s$-channel limit of the percolation cluster four-spin correlator:[15]

$$\langle \sigma(\infty)\sigma(1)\sigma(z,\bar{z})\sigma(0)\rangle^{\text{perco}} = (z\bar{z})^{-2h_\sigma}\Bigg(1 + (z\bar{z})^{h_\varepsilon}\frac{C^2_{\sigma\sigma\varepsilon}}{\gamma^2}\left(\theta_1 + \gamma\ln(z\bar{z})\right) \tag{143}$$

$$+ (z^2+\bar{z}^2)\frac{C^2_{\sigma\sigma T}}{b^2}\left(\theta + b\ln(z\bar{z})\right) + \ldots$$

$$+ (z\bar{z}^2+z^2\bar{z})\frac{C^2_{\sigma\sigma T}}{2b} + (z\bar{z})^2\frac{C^2_{\sigma\sigma T}}{2b}$$

$$+ (z\bar{z})^2\frac{C^2_{\sigma\sigma\Psi_0}}{a^2}\left(a_2 + a_1\ln(z\bar{z}) + \frac{a}{2}\ln^2(z\bar{z})\right) + \ldots\Bigg).$$

Let us remark that the logarithmic dependence $\ln(z\bar{z})$ in (143) appears in the percolation limit of geometric configurations of the type Fig. 1 (left) where points 1,2 (placed at $\infty, 1$ here) and points 3,4 (placed at $(z,\bar{z}), 0$ here) belong to two distinct clusters. A geometrical interpretation of such logarithmic dependence (from the energy logarithmic pair contributions) was recently given in a very interesting probabilistic construction [48, 49]: in this context, the logarithm essentially appears from summing probabilities of events of the same order at different scales. It would be very interesting to see if other logarithmic contributions we study here using the CFT analysis – in particular those arising from higher rank Jordan blocks – can be similarly analyzed using the probabilistic approach.

### 4.2.3  Remarks on the 2-hull operator $\hat{\Phi}_{0,2}$

In section 3.2, we have analyzed the normalization of Kac operators in the cluster model as a function of the central charge. This analysis was possible due to the complete analytic control of the Kac operator amplitudes in the four-spin correlator. Although we believe that similar analysis could apply to other operators, the studies would be much more involved, due to the lack of full control of analytic data at generic $c$. In subsection 4.2 above, we saw that the 2-hull

---

[15] From (142) to (143), we have fixed a basis for the logarithmic pair $(\tilde{\varepsilon},\varepsilon)$. See appendix D.3. $\theta_1$ here is the constant in the two-point function of $\tilde{\varepsilon}$: $\langle\tilde{\varepsilon}\tilde{\varepsilon}\rangle = (-2\gamma\ln(z\bar{z}) + \theta_1)\mathbb{P}_2^0$.

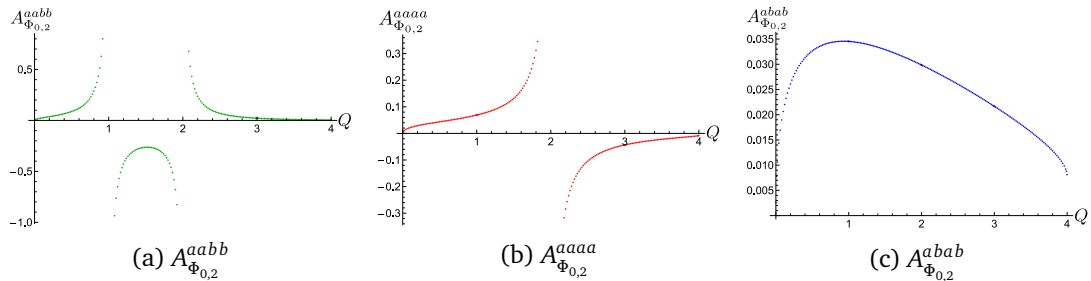

Figure 10: The amplitudes of the 2-hull operator $\Phi_{0,2}$ in the connectivities $P_{aabb}, P_{aaaa}, P_{abab}$. Plots are extracted from [15].

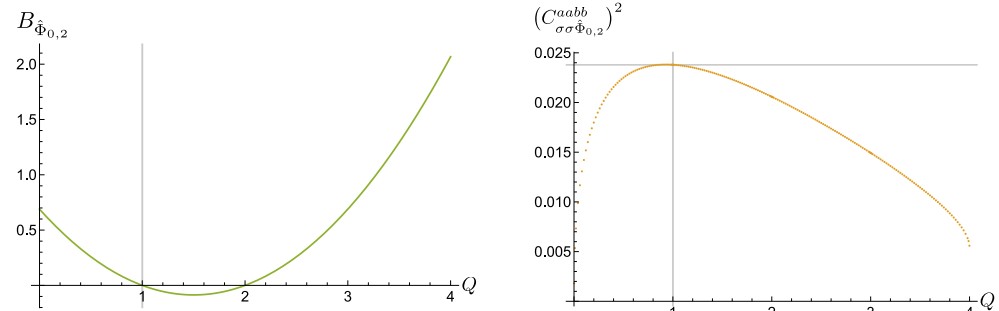

Figure 11: The deduced norm (145) and the resulting three-point constant squared for the 2-hull operator $\hat{\Phi}_{0,2}$ by considering the four-point connectivity $P_{aabb}$. The horizontal grid line in the right plot indicates the value from (138) and thus provide a check on the cancellation condition (141).

operator $\hat{\Phi}_{0,2}$ in cluster model should have its norm behaving as in eq. (125), as expected from mixing with energy operator $\hat{\Phi}_{2,1}$ at $c = 0$. Here we make some remarks on the normalization for the 2-hull operator.

In the cluster connectivities, the 2-hull operator appears in all four different connectivities

$$P_{aaaa}, \quad P_{abab}, \quad P_{aabb}, \quad P_{abba}, \tag{144}$$

where the subscript indicates the connectivity of the four points of operator insertions (e.q. $P_{aabb}$ depicted in Fig. 1). Its amplitudes in these connectivities were first bootstrapped in [15] and as a function of $Q$ they are given in Fig. 10. See also [18] for a recent bootstrap extraction of the analytic expressions of these amplitudes. If we focus on the amplitudes in $P_{aabb}$ where we have already analyzed the Kac operators above, we are tempted to take the normalization of the 2-hull operator as

$$B_{\hat{\Phi}_{0,2}}(Q) \overset{Q\to 1}{\sim} \frac{5\sqrt{3}}{8\pi}(Q-1)(Q-2), \tag{145}$$

where we have included a constant factor to match the normalization constant factor in (125). See fig. 11. This appears to be a reasonable guess from several aspects. First, this leads to a non-vanishing three-point coupling $C_{\sigma\sigma\hat{\Phi}_{0,2}}^{aabb}$ ($c = 0$) as we expect from (141) (Fig. 11), and as a result, the percolation spin four-point function (143) describing $P_{aabb}$ indeed contains logarithmic contribution from the energy logarithmic pair $(\tilde{\varepsilon}, \varepsilon)$. Note that the norm (145) of $\hat{\Phi}_{0,2}$ has an opposite derivative at $c = 0$ from the norm of (90a) of $\hat{\Phi}_{2,1}$, as needed for the logarithmic mixing. Second, the amplitudes of $\hat{\Phi}_{0,2}$ in different connectivities are related

as [45]:

$$(1-Q)\left(C^{aabb}_{\sigma\sigma\hat{\Phi}_{0,2}}\right)^2 = \left(C^{aaaa}_{\sigma\sigma\hat{\Phi}_{0,2}}\right)^2,$$
$$\frac{(1-Q)(2-Q)}{2}\left(C^{aabb}_{\sigma\sigma\hat{\Phi}_{0,2}}\right)^2 = \left(C^{abab}_{\sigma\sigma\hat{\Phi}_{0,2}}\right)^2. \tag{146}$$

The finite $C^{aabb}_{\sigma\sigma\hat{\Phi}_{0,2}}$ makes the three-point constants

$$C^{aaaa}_{\sigma\sigma\hat{\Phi}_{0,2}} = C^{abab}_{\sigma\sigma\hat{\Phi}_{0,2}} = 0. \tag{147}$$

This is consistent with the degeneracy of the energy operator $\varepsilon \sim \hat{\Phi}_{2,1}$ and the 2-hull operator $\hat{\Phi}_{0,2}$ as the bottom field in the Jordan block at the percolation point: Since the energy operator $\hat{\Phi}_{2,1}$ does not appear in the connectivities $P_{abab}$ and $P_{aaaa}$ at generic $c$, we have

$$C^{aaaa}_{\sigma\sigma\varepsilon} = C^{abab}_{\sigma\sigma\varepsilon} = 0, \tag{148}$$

agreeing with (147). The vanishing three-point structure constant (148) then means that the three-point function between spin operator and the top field $\tilde{\varepsilon}$ (in the cluster decomposition of configurations $P_{aaaa}, P_{abab}$)

$$\langle\sigma(z_1,\bar{z}_1)\sigma(z_2,\bar{z}_2)\tilde{\varepsilon}(z_3,\bar{z}_3)\rangle = \left(C_{\sigma\sigma\tilde{\varepsilon}} + C_{\sigma\sigma\varepsilon}\tau_3\right)\mathbb{P}^0_3 = C_{\sigma\sigma\tilde{\varepsilon}}\mathbb{P}^0_3, \tag{149}$$

does not contain a logarithm. Correspondingly, the four-point functions of the percolation spin that describe the configurations $P_{aaaa}, P_{abab}$ do not get a logarithm from $(\tilde{\varepsilon}, \varepsilon)$ in the $s$-channel limit. This agrees with the recent probabilistic construction [48].

Nevertheless, the tentative normalization (145) leaves puzzles. The most puzzling point is that such normalization only makes one of the three-point structure constant $\left(C^{aabb}_{\hat{\Phi}_{0,2}}\right)^2$ positive for $0 < Q < 4$ and leave the $\left(C^{aaaa}_{\hat{\Phi}_{0,2}}\right)^2$ and $\left(C^{abab}_{\hat{\Phi}_{0,2}}\right)^2$ still negative for some range of $Q$. In fact, due to the universal amplitude ratios (146), it would be impossible to make all the three-point constants real. This might be related to the recent findings that operators such as the hull operators are complicated objects containing several different operators transforming in different representations of the global symmetry group [17, 19, 50]. A generic analysis of the operator normalizations in those cases remains a complicated subject for future studies.

## 4.3 Higher rank Jordan block

Let us now consider the case of rank-3 Jordan block. As we have claimed in section 2.2.2, this happens when the Kac operator (denoted $\phi$ here) acquires a double zero at central charge $c_*$:

$$\langle\phi(z,\bar{z})\phi(0,0)\rangle = B_\phi(c)\mathbb{P}^c_2 \simeq B^{(2)}_\phi(c-c_*)^2\mathbb{P}^*_2 + o\left((c-c_*)^2\right). \tag{150}$$

We now consider building a rank-3 Jordan block with the bottom field $\phi$ which could for example represent Kac operator $\hat{\Phi}_{3,1}$ in cluster model or $\hat{\Phi}_{5,1}$ in dilute loop model. In the meantime, we have another field $\psi$ that could represent $T\bar{T}$ with normalization

$$B_\psi(c) \simeq B^{(2)}_\psi(c-c_*)^2 + o\left((c-c_*)^2\right), \qquad B^{(2)}_\phi = -B^{(2)}_\psi, \tag{151}$$

and a rank-2 Jordan block of $\left(\mathcal{O}^{(2;2)}, \mathcal{O}^{(2;1)}\right)$ at generic $c$ with two-point functions

$$\langle\mathcal{O}^{(2;2)}(z,\bar{z})\mathcal{O}^{(2;2)}(0,0)\rangle = \left(-2\gamma_\mathcal{O}(c)\ln(z\bar{z}) + \theta_\mathcal{O}(c)\right)\mathbb{P}^c_2, \tag{152a}$$
$$\langle\mathcal{O}^{(2;2)}(z,\bar{z})\mathcal{O}^{(2;1)}(0,0)\rangle = \gamma_\mathcal{O}(c)\mathbb{P}^c_2, \tag{152b}$$
$$\langle\mathcal{O}^{(2;1)}(z,\bar{z})\mathcal{O}^{(2;1)}(0,0)\rangle = 0. \tag{152c}$$

The logarithmic coupling of the rank-2 Jordan block (152) has a single zero at $c = c_*$:

$$\gamma_{\mathcal{O}}(c) \simeq \gamma'_{\mathcal{O}}(c - c_*) + o(c - c_*). \tag{153}$$

We can now define the top field of the rank-3 Jordan block symmetrically as:

$$\mathcal{O}^{(3;3)} = a\left(\frac{\phi}{B_\phi(c)} + \frac{\psi}{B_\psi(c)} + \frac{\mathcal{O}^{(2;2)}}{\gamma_{\mathcal{O}}(c)}\right). \tag{154}$$

This definition is based on the simple consideration that we can choose the bottom field $\mathcal{O}^{(3;1)}$ to be:

$$\mathcal{O}^{(3;1)} = \phi, \psi \text{ or } \mathcal{O}^{(2;1)}, \tag{155}$$

and the two-point function with the top field:

$$\langle \mathcal{O}^{(3;3)}(z,\bar{z})\,\mathcal{O}^{(3;1)}(0,0)\rangle = a, \tag{156}$$

defines the logarithmic coupling $a$ for the rank-3 Jordan block. From the last term of (154), we see that in the case of a rank-2 Jordan block, its logarithmic coupling plays the role of the usual two-point constants (normalization) for non-logarithmic operators.

Computing the two-point function of $\mathcal{O}^{(3;3)}$, we first find the singularity cancellation condition

$$\gamma'_{\mathcal{O}} = -\frac{B_\phi^{(2)}}{h'_\phi - h'_\psi}, \tag{157}$$

with the expression evaluated at $c = c_*$ and

$$\theta_{\mathcal{O}}(c) \simeq \theta'_{\mathcal{O}}(c - c_*), \quad \theta'_{\mathcal{O}} = \frac{B_\phi^{(3)} + B_\psi^{(3)}}{(h'_\phi - h'_\psi)^2}. \tag{158}$$

The finite two-point function is finally:

$$\langle \mathcal{O}^{(3;3)}(z,\bar{z})\,\mathcal{O}^{(3;3)}(0,0)\rangle = \left(-\frac{2a^2}{B_\phi^{(2)}}(2h'_{\mathcal{O}} - h'_\phi - h'_\psi)(h'_\phi - h'_\psi)\ln^2(z\bar{z}) + a_1 \ln(z\bar{z}) + a_2\right)\mathbb{P}_2^*. \tag{159}$$

Conformal Ward identities require:

$$-\frac{2a^2}{B_\phi^{(2)}}(2h'_{\mathcal{O}} - h'_\phi - h'_\psi)(h'_\phi - h'_\psi) = 2a, \tag{160}$$

giving

$$a = -\frac{B_\phi^{(2)}}{(2h'_{\mathcal{O}} - h'_\phi - h'_\psi)(h'_\phi - h'_\psi)} = \frac{\gamma'_{\mathcal{O}}}{2h'_{\mathcal{O}} - h'_\phi - h'_\psi}, \tag{161}$$

where in the second equation we have used (157). Note that again, this logarithmic coupling is completely fixed by the normalizations of the operators and their conformal dimensions as functions of the central charge. Note also that once the top field (154) is constructed, all the two-point constants are contained in its two-point function (159). The rest of the Jordan block (in this case the middle field, but in general any operator between the top and the bottom fields) can be reached by acting with dilatation $L_0, \bar{L}_0$. Special care needs to be taken when expanding the action in the parameter $c$ to guarantee conformal Ward identities are satisfied. See eqs. (C.4), (C.5) and the discussions there.

As a check, let us take explicitly the example of $\hat{\Phi}_{3,1}$ and $\hat{\Phi}_{1,5}$ in cluster and dilute loop models. In the case of cluster model, we have deduced the normalization of $\hat{\Phi}_{3,1}$ in (90) up to a positive function. In this case, we take $\phi \to \hat{\Phi}_{3,1}, \psi \to T\bar{T}$ with

$$\langle T\bar{T}(z,\bar{z})T\bar{T}(0,0)\rangle = \frac{c^2}{4}\mathbb{P}_2^c, \tag{162}$$

and

$$B_{\hat{\Phi}_{3,1}} \overset{c\to 0}{\simeq} -\frac{c^2}{4}, \tag{163}$$

so

$$B_\phi^{(2)} = -\frac{1}{4}. \tag{164}$$

Eq. (161) gives

$$a = -\frac{25}{48}, \tag{165}$$

as we have obtained before. The same works if we take $\phi \to \hat{\Phi}_{1,5}, \psi \to T\bar{T}$ as in the case of SAW.

The above constructions suggest a generic mechanism for building higher rank Jordan blocks. One starts with a number of (logarithmic and non-logarithmic) operators with vanishing norms and coincidental dimensions at a special value of central charge.[16] The number of operators involved appears to be related to the rank of the Jordan block: for example, if the pattern for the rank-2 and rank-3 Jordan blocks continues, then a rank-$k$ Jordan block could involve mixing $2^{k-1}$ non-logarithmic operators. On the other hand, instead of non-logarithmic operators, the mixing could also involve lower rank Jordan blocks with an equivalent number of non-logarithmic operators. For example, as we have seen, a rank-2 Jordan block is equivalent to two non-logarithmic operators when mixing to higher rank Jordan blocks. Although speculative at the moment, it is fascinating to note the possible drastic reduction of the number of operators when logarithmic mixing appears, particularly for higher rank Jordan blocks. Alternatively speaking, a (possibly exponentially) increasing number of operators are needed to build higher rank Jordan blocks. This is not obvious for the well-analyzed rank-2 Jordan blocks, since one starts with two non-logarithmic operators and ends up with two logarithmic operators. For rank-3 Jordan blocks that we analyzed explicitly, already four operators are used to build three logarithmic operators, and this pattern could continue more dramatically to higher rank. Additionally the increasing rank of the Jordan block appears to be related to the increasing vanishing norms of the unmixed operators. For a rank-$k$ Jordan block, a non-logarithmic operator might have a $(k-1)$-order zero norm at the mixing point. Intuitively, the higher order the zero in the two-point function, the more operators are needed to build the Jordan block since eventually, the logarithmic mixing allows a non-vanishing correlation between the top field and the bottom field characterized by the logarithmic coupling.

To be concrete, consider an operator $\phi$ whose normalization acquires an order $k-1$ zero at $c = c_*$

$$\langle \phi(z,\bar{z})\phi(0,0)\rangle = B_\phi(c)\mathbb{P}_2^c \simeq B_\phi^{(k-1)}(c-c_*)^{k-1}\mathbb{P}_2^* + o\big((c-c_*)^{k-1}\big), \tag{166}$$

and its dimension coincides with other operators, e.g. a rank-$l$ Jordan block $(\mathcal{O}^{(l;l)}, \dots, \mathcal{O}^{(l;1)})$. We can hen write the top of the rank-$k$ Jordan block with the top fields

$$\mathcal{O}^{(k;k)} = \gamma_k\left(\frac{\phi}{B_\phi(c)} + \dots + \frac{\mathcal{O}^{(l;l)}}{\gamma_l(c)} + \dots\right), \tag{167}$$

---

[16]In the case of the logarithmic operator, we can take the logarithmic coupling as a generalization of the norm. This is also the anti-diagonal element of the Gram matrix. This definition is consistent with the case of the non-logarithmic operator which could be seen as a rank-1 Jordan block.

where in each term, we have the field divided by its norm. The bottom field can be taken to be any of the operators:

$$\mathcal{O}^{(k;1)} = \phi, \mathcal{O}^{(l;1)}, \dots, \tag{168}$$

and this defines the logarithmic coupling of the rank-$k$ Jordan block:

$$\langle \mathcal{O}^{(k;k)}(z, \bar{z}) \mathcal{O}^{(k;1)}(0,0) \rangle = \gamma_k \mathbb{P}_2^*. \tag{169}$$

Once we have the top field $\mathcal{O}^{(k;k)}$, all the information about the Jordan block is encoded in its two-point function which can be computed by taking the $c \to c_*$ limit and extracting the finite part. In particular, the logarithmic coupling $\gamma_k$ can be computed using conformal Ward identities as we have seen in the previous examples. Furthermore, the remaining fields in the Jordan block can be reached by acting with dilatation.

## 5 Logarithmic OPE

We are now ready to go back to the "$c \to 0$ catastrophe" (12) and consider the resolution iii. When the operator $\mathcal{O}$ in (12) is a Kac operator, the resolution i does not apply. This is because the OPE of Kac operator is constrained by the Virasoro degeneracy. Consider the simplest Kac operator $\Phi_{2,1}$, i.e., the energy operator in the cluster model and its OPE at generic $c$:

$$
\begin{aligned}
\Phi_{2,1}(z, \bar{z}) \Phi_{2,1}(0,0) = (z\bar{z})^{-2h_{2,1}(c)} \Big( 1 &+ \frac{h_{2,1}(c)}{c/2}\big(z^2 T + \bar{z}^2 \bar{T}\big) + \frac{h_{2,1}^2(c)}{c^2/4}(z\bar{z})^2 T\bar{T} + \dots \\
&+ D_{\Phi_{2,1}\Phi_{2,1}\Phi_{3,1}}(c)(z\bar{z})^{h_{3,1}(c)} \Phi_{3,1} + \dots \Big).
\end{aligned}
\tag{170}
$$

Comparing with a non-Kac operator OPE such as the order parameter $\sigma$ (24), the key point here is that due to the degenerate fusion rule, one does not have the operators $X, \bar{X}$ at generic $c$ so the divergence of $1/c$ cannot be canceled in the same way as in resolution i. This scenario applies generically to Kac operators and as pointed out by Cardy [5], this leads to a third resolution to the "$c \to 0$ catastrophe", namely that all Kac operators have to have vanishing norm at $c = 0$ and sit at the bottom of Jordan blocks. We have seen such vanishing norms of Kac operators in section 3 from a different perspective. In this section, we go into the detail the resolution iii and analyze the fate of the energy operator OPE (170) at $c = 0$.

### 5.1 Energy logarithmic pair $(\tilde{\varepsilon}, \varepsilon)$

We start with another look at the normalization of the energy operator. Taking the example of percolation, recall our construction of the logarithmic pair $(t, T)$ at $c = 0$ with the top field

$$t = b\left(\frac{T}{c/2} + \frac{\hat{X}}{B_{\hat{X}}(c)}\right). \tag{171}$$

Due to the degenerate fusion rule, at generic $c$ we have the three-point function:

$$\langle \Phi_{2,1}(z_1, \bar{z}_1) \Phi_{2,1}(z_2, \bar{z}_2) X(z_3, \bar{z}_3) \rangle = 0. \tag{172}$$

So if we take the three-point function of the unit normalized energy operator $\Phi_{2,1}$ with the top field $t$ near $c = 0$, we find

$$\langle \Phi_{2,1}(z_1, \bar{z}_1) \Phi_{2,1}(z_2, \bar{z}_2) t(z_3, \bar{z}_3) \rangle \overset{c \to 0}{\simeq} \frac{b h_{2,1}(c)}{c/2} \mathbb{P}_3^c, \tag{173}$$

where we have used that at generic $c$

$$\langle \Phi_{2,1}(z_1,\bar{z}_1)\Phi_{2,1}(z_2,\bar{z}_2)T(z_3)\rangle = h_{2,1}(c)\mathbb{P}_3^c. \tag{174}$$

Note that earlier when computing the three-point function of the spin operator $\sigma$ with $t$ in (36), we did not encounter the $1/c$ divergence as in (173) because the divergence is exactly canceled between $\langle\sigma\sigma T\rangle$ and $\langle\sigma\sigma\hat{X}\rangle$ according to the definition of $t$ in (171). This cancellation does not happen for the three-point function of $\Phi_{2,1}$ and $t$ here in (173) due to the degenerate fusion rule (172) at generic $c$. The only resolution to keep the three-point function finite is to properly normalize the Kac operator $\Phi_{2,1}$ as $\varepsilon^{\text{perco}} \equiv \hat{\Phi}_{2,1}$ with $B_\varepsilon \sim c$. Taking $B_\varepsilon \simeq c/2$ as in (124), we find

$$\langle\varepsilon(z_1,\bar{z}_1)\,\varepsilon(z_2,\bar{z}_2)\,t(z_3,\bar{z}_3)\rangle^{\text{perco}} = C_{\varepsilon\varepsilon t}^{\text{perco}}\mathbb{P}_3^0, \qquad C_{\varepsilon\varepsilon t}^{\text{perco}} = bh_\varepsilon^{\text{perco}} = -\frac{25}{8}, \tag{175}$$

where we denote $h_\varepsilon^{\text{perco}} = h_{2,1}(c=0) = h_{0,2}(c=0)$. This normalization agrees with our previous analysis in section 3, and renders the three-point function

$$\langle\varepsilon(z_1,\bar{z}_1)\,\varepsilon(z_2,\bar{z}_2)T(z_3)\rangle^{\text{perco}} = 0, \tag{176}$$

consistent with that (175) does not contain a logarithmic dependence, according to conformal Ward identities. Such a vanishing three-point coupling $C_{\varepsilon\varepsilon T}$ appears to be a remnant effect of the Virasoro degeneracy of the energy operator. However, the coupling between the energy operator and the stress tensor does not really disappear at $c=0$, but instead gives rise to a coupling with the top field $t$, as we see in eq. (175).

We can continue further to the three-point function of $\varepsilon^{\text{perco}}$ with the top field $\Psi_2$ of the rank-3 Jordan block. Taking the definition as in (154):[17]

$$\Psi_2 = a\left(\frac{\hat{\Phi}_{3,1}}{B_{\hat{\Phi}_{3,1}}(c)} + \frac{T\bar{T}}{c^2/4} + \frac{\Psi}{\gamma_\Psi(c)}\right), \tag{177}$$

at generic $c$, we have

$$\langle\Phi_{2,1}(z_1,\bar{z}_1)\Phi_{2,1}(z_2,\bar{z}_2)\Psi(z_3,\bar{z}_3)\rangle = 0, \tag{178a}$$

$$\langle\Phi_{2,1}(z_1,\bar{z}_1)\Phi_{2,1}(z_2,\bar{z}_2)T\bar{T}(z_3,\bar{z}_3)\rangle = h_{2,1}^2(c)\mathbb{P}_3^c, \tag{178b}$$

$$\langle\Phi_{2,1}(z_1,\bar{z}_1)\Phi_{2,1}(z_2,\bar{z}_2)\Phi_{3,1}(z_3,\bar{z}_3)\rangle = C_{\Phi_{2,1}\Phi_{2,1}\Phi_{3,1}}(c)\mathbb{P}_3^c, \tag{178c}$$

with $C_{\Phi_{2,1}\Phi_{2,1}\Phi_{3,1}}(c)$ given in (A.22). We then find at $c=0$

$$\langle\varepsilon(z_1,\bar{z}_1)\,\varepsilon(z_2,\bar{z}_2)\,\Psi_2(z_3,\bar{z}_3)\rangle^{\text{perco}} = \left(C_{\varepsilon\varepsilon\Psi_2}^{\text{perco}} + C_{\varepsilon\varepsilon\Psi_1}^{\text{perco}}\tau_3\right)\mathbb{P}_3^0, \tag{179a}$$

$$\langle\varepsilon(z_1,\bar{z}_1)\,\varepsilon(z_2,\bar{z}_2)\,\Psi_0(z_3,\bar{z}_3)\rangle^{\text{perco}} = 0, \tag{179b}$$

where[18]

$$C_{\varepsilon\varepsilon\Psi_1}^{\text{perco}} = -2ah_{3,1}'\left(h_\varepsilon^{\text{perco}}\right)^2 = \frac{a\left(h_\varepsilon^{\text{perco}}\right)^2}{b_{1,2}^{\text{perco}}} = -\frac{125}{512}. \tag{180}$$

By conformal Ward identity, we have from (179)

$$\langle\varepsilon(z_1,\bar{z}_1)\,\varepsilon(z_2,\bar{z}_2)\,\Psi_1(z_3,\bar{z}_3)\rangle^{\text{perco}} = C_{\varepsilon\varepsilon\Psi_1}^{\text{perco}}\mathbb{P}_3^0. \tag{181}$$

---

[17]The definition of $\Psi_2$ in (177) and the one earlier in (47) are related by change of basis in the rank-3 Jordan block.

[18]Note that for the singularities to be canceled, one has $C_{\hat{\Phi}_{2,1}\hat{\Phi}_{2,1}\hat{\Phi}_{3,1}} \overset{c\to0}{\simeq} \frac{h_\varepsilon^2}{2}c + \dots$ which can be checked to be true.

The same result can be obtained by directly computing this three point function using (52). Similar to before, the three-point coupling here arises from the generic $c$ coupling between the energy operator $\varepsilon$ and the second energy operator $\varepsilon'$, despite that the three-point function $\langle \varepsilon\varepsilon\Psi_0 \rangle \sim \langle \varepsilon\varepsilon\varepsilon' \rangle$ itself vanishes at $c = 0$.

Using the three-point functions, we can construct the OPE of percolation energy operator $\varepsilon$ at $c = 0$:

$$\varepsilon^{\text{perco}}(z,\bar{z})\varepsilon^{\text{perco}}(0,0) = (z\bar{z})^{-2h_\varepsilon^{\text{perco}}}\left\{ z^2 \frac{C_{\varepsilon\varepsilon t}^{\text{perco}}}{b}T + c.c. + \ldots + (z\bar{z})^2 \right. \tag{182}$$

$$\times \left( \frac{C_{\varepsilon\varepsilon\Psi_1}^{\text{perco}}}{a}\Psi_1 + \frac{C_{\varepsilon\varepsilon\Psi_1}^{\text{perco}}}{a}\ln(z\bar{z})\Psi_0 + \frac{aC_{\varepsilon\varepsilon\Psi_2}^{\text{perco}} - a_1 C_{\varepsilon\varepsilon\Psi_1}^{\text{perco}}}{a^2}\Psi_0 \right)$$

$$\left. + \ldots \right\}.$$

It is worth comparing this OPE (of the bottom field in a rank-2 Jordan block) with that of the non-logarithmic spin operator (59) in percolation and SAW. We see the effect of the vanishing three-point structure constants of the bottom operators in the Jordan blocks. First, the terms $\bar{\partial}t, \partial\bar{t}, \bar{\partial}^2 t, \partial^2\bar{t}$ are lacking in the OPE since the three-point functions of $\varepsilon$ with them vanish. (In the case of spin operator, this does not vanish due to the logarithmic dependence in the three-point function of $\langle \sigma\sigma t \rangle$.) Moreover, the OPE here involves only a partial structure of the rank-2 and rank-3 Jordan blocks of the identity Virasoro module. Note that here, the identity Virasoro family still appears in the OPE of the energy operator $\varepsilon$, but not the identity operator itself.

We can proceed to consider further three-point functions involving the top field $\tilde{\varepsilon}^{\text{perco}}$. At generic $c$, in addition to (170), we have the OPE

$$\Phi_{2,1}(z,\bar{z})\Phi_{0,2}(0,0) = (z\bar{z})^{-h_{2,1}(c)-h_{0,2}(c)}\left( D_{\Phi_{2,1}\Phi_{0,2}X}(c)(z\bar{z})^{h_{1,2}(c)}\left( z^2 X + \bar{z}^2\bar{X} \right) + \ldots \right). \tag{183}$$

Take the definition of the top field from (126)

$$\tilde{\varepsilon}^{\text{perco}} = \gamma^{\text{perco}}\left( \frac{\hat{\Phi}_{2,1}}{B_{\hat{\Phi}_{2,1}}(c)} + \frac{\hat{\Phi}_{0,2}}{B_{\hat{\Phi}_{0,2}}(c)} \right), \tag{184}$$

and the generic $c$ three-point functions:

$$\langle \Phi_{0,2}(z_1,\bar{z}_1)\Phi_{2,1}(z_2,\bar{z}_2)T(z_3) \rangle = 0, \tag{185a}$$

$$\langle \Phi_{0,2}(z_1,\bar{z}_1)\Phi_{2,1}(z_2,\bar{z}_2)X(z_3,\bar{z}_3) \rangle = C_{\Phi_{0,2}\Phi_{2,1}X}(c)\mathbb{P}_3^c. \tag{185b}$$

We obtain the three-point functions:

$$\langle \tilde{\varepsilon}(z_1,\bar{z}_1)\varepsilon(z_2,\bar{z}_2)T(z_3) \rangle^{\text{perco}} = \gamma^{\text{perco}}h_\varepsilon^{\text{perco}}\mathbb{P}_3^0, \tag{186a}$$

$$\langle \tilde{\varepsilon}(z_1,\bar{z}_1)\varepsilon(z_2,\bar{z}_2)t(z_3,\bar{z}_3) \rangle^{\text{perco}} = \left( \gamma^{\text{perco}}h_\varepsilon^{\text{perco}}\tau_3 + bh_\varepsilon^{\text{perco}}\tau_1 + C_{\tilde{\varepsilon}\varepsilon t}^{\text{perco}} \right)\mathbb{P}_3^0. \tag{186b}$$

The singularity cancellation requires

$$C_{\hat{\Phi}_{0,2}\hat{\Phi}_{2,1}\hat{X}}(c) = -\frac{h_\varepsilon^{\text{perco}}}{2}c + o(c), \tag{187}$$

which can be checked to be true. See appendix B.2. We did not write out the expression of $C_{\tilde{\varepsilon}\varepsilon t}^{\text{perco}}$ since this depends on the basis of the Jordan blocks and is in this sense not an intrinsic

quantity. Furthermore, at generic $c$ we have

$$\langle \Phi_{0,2}(z_1,\bar{z}_1)\Phi_{2,1}(z_2,\bar{z}_2)T\bar{T}(z_3,\bar{z}_3)\rangle = 0\,, \tag{188a}$$

$$\langle \Phi_{0,2}(z_1,\bar{z}_1)\Phi_{2,1}(z_2,\bar{z}_2)\Phi_{3,1}(z_3,\bar{z}_3)\rangle = 0\,, \tag{188b}$$

$$\langle \Phi_{0,2}(z_1,\bar{z}_1)\Phi_{2,1}(z_2,\bar{z}_2)\Psi(z_3,\bar{z}_3)\rangle = \big(C_{\Phi_{0,2}\Phi_{2,1}\Psi}(c) + C_{\Phi_{0,2}\Phi_{2,1}A\bar{X}}(c)\tau_3\big)\mathbb{P}_3^c\,, \tag{188c}$$

where recall that at generic $c$ the fields $(\Psi, A\bar{X})$ form a rank-2 Jordan block. We find the 3-point functions with the rank-3 Jordan blocks:

$$\langle \tilde{\varepsilon}(z_1,\bar{z}_1)\varepsilon(z_2,\bar{z}_2)\Psi_0(z_3,\bar{z}_3)\rangle^{\mathrm{perco}} = \gamma^{\mathrm{perco}}\big(h_\varepsilon^{\mathrm{perco}}\big)^2\mathbb{P}_3^0\,,$$

$$\langle \tilde{\varepsilon}(z_1,\bar{z}_1)\varepsilon(z_2,\bar{z}_2)\Psi_1(z_3,\bar{z}_3)\rangle^{\mathrm{perco}} = \left(\frac{a\big(h_\varepsilon^{\mathrm{perco}}\big)^2}{b_{1,2}^{\mathrm{perco}}}\tau_1 + \gamma^{\mathrm{perco}}\big(h_\varepsilon^{\mathrm{perco}}\big)^2\tau_3 + C_{\tilde{\varepsilon}\varepsilon\Psi_1}\right)\mathbb{P}_3^0\,,$$

$$\langle \tilde{\varepsilon}(z_1,\bar{z}_1)\varepsilon(z_2,\bar{z}_2)\Psi_2(z_3,\bar{z}_3)\rangle^{\mathrm{perco}} = \left(C_{\tilde{\varepsilon}\varepsilon\Psi_2} + \frac{\gamma^{\mathrm{perco}}\big(h_\varepsilon^{\mathrm{perco}}\big)^2}{2}\tau_3^2 + \frac{a\big(h_\varepsilon^{\mathrm{perco}}\big)^2}{b_{1,2}^{\mathrm{perco}}}\tau_1\tau_3\right. \tag{189}$$

$$\left. + C_{\tilde{\varepsilon}\varepsilon\Psi_1}\tau_3 + C_{\varepsilon\varepsilon\Psi_2}\tau_1\right)\mathbb{P}_3\,,$$

where the singularity cancellation conditions are checked in appendix B.2. Note the agreement of $C_{\varepsilon\varepsilon\Psi_1}$ here with (180) provides an additional check of the computations. Up to this point, we have determined the following three-point constants at $c = 0$:

$$C_{\varepsilon\varepsilon t}^{\mathrm{perco}} = bh_\varepsilon^{\mathrm{perco}} = -\frac{25}{8}\,, \qquad\qquad C_{\varepsilon\varepsilon\Psi_1}^{\mathrm{perco}} = \frac{a\big(h_\varepsilon^{\mathrm{perco}}\big)^2}{b_{1,2}^{\mathrm{perco}}} = -\frac{125}{512}\,, \tag{190}$$

$$C_{\tilde{\varepsilon}\varepsilon T}^{\mathrm{perco}} = \gamma^{\mathrm{perco}}h_\varepsilon^{\mathrm{perco}} = -\frac{25}{32}\,, \qquad C_{\tilde{\varepsilon}\varepsilon\Psi_0}^{\mathrm{perco}} = \gamma^{\mathrm{perco}}\big(h_\varepsilon^{\mathrm{perco}}\big)^2 = -\frac{125}{256}\,,$$

which are intrinsic quantities that are invariant under change of basis in the rank-2 and rank-3 Jordan blocks (see appendix D.2).

The case of the energy operator $\Phi_{1,3}$ in SAW is analogous. At generic $c$, the Kac operator has the OPE

$$\Phi_{1,3}(z,\bar{z})\Phi_{1,3}(0,0) = (z\bar{z})^{-2h_{1,3}(c)}\left(1 + \frac{h_{1,3}(c)}{c/2}\big(z^2T + \bar{z}^2\bar{T}\big) + \frac{h_{1,3}^2(c)}{c^2/4}(z\bar{z})^2T\bar{T} + \ldots\right.$$

$$\left. + D_{\Phi_{1,3}\Phi_{1,3}\Phi_{1,3}}(c)(z\bar{z})^{h_{1,3}(c)}\Phi_{1,3} + \ldots \right. \tag{191}$$

$$\left. + D_{\Phi_{1,3}\Phi_{1,3}\Phi_{1,5}}(c)(z\bar{z})^{h_{1,5}(c)}\Phi_{1,5} + \ldots\right)\,,$$

and we also have

$$\Phi_{1,3}(z,\bar{z})\Phi_{1,0}(0,0) = (z\bar{z})^{-h_{1,3}(c)-h_{1,0}(c)}\left(D_{\Phi_{1,3}\Phi_{1,0}\Phi_{1,0}}(c)(z\bar{z})^{h_{1,0}(c)}\Phi_{1,0} + \ldots\right.$$

$$\left. + D_{\Phi_{1,3}\Phi_{1,0}X}(c)(z\bar{z})^{h_{1,2}(c)}\big(z^2X + \bar{z}^2\bar{X}\big) + \ldots\right)\,. \tag{192}$$

Using the same arguments as in the previous case, we can compute three-point functions involving the properly normalized energy operator $\varepsilon^{\mathrm{SAW}} \equiv \hat{\Phi}_{1,3}$ with $B_\varepsilon \simeq c/2$ and its logarithmic partner (from (126)):

$$\tilde{\varepsilon}^{\mathrm{SAW}} = \gamma^{\mathrm{SAW}}\left(\frac{\hat{\Phi}_{1,3}}{B_{\hat{\Phi}_{1,3}}(c)} + \frac{\hat{\Phi}_{1,0}}{B_{\hat{\Phi}_{1,0}}(c)}\right)\,. \tag{193}$$

The logarithmic three-point functions at $c = 0$ take the same form as in (186) and (189), with the three-point constants:

$$C_{\varepsilon\varepsilon t}^{\text{SAW}} = b h_{\varepsilon}^{\text{SAW}} = -\frac{5}{3}, \qquad C_{\varepsilon\varepsilon\Psi_1}^{\text{SAW}} = \frac{a\left(h_{\varepsilon}^{\text{SAW}}\right)^2}{b_{1,2}^{\text{SAW}}} = \frac{5}{54},$$
$$C_{\tilde{\varepsilon}\varepsilon T}^{\text{SAW}} = \gamma^{\text{SAW}} h_{\varepsilon}^{\text{SAW}} = \frac{5}{9}, \qquad C_{\tilde{\varepsilon}\varepsilon\Psi_0}^{\text{SAW}} = \gamma^{\text{SAW}}\left(h_{\varepsilon}^{\text{SAW}}\right)^2 = \frac{5}{27}. \tag{194}$$

An additional point in the case of SAW is that at generic $c$, one has a non-vanishing three-point function of the energy operator with itself

$$\langle \Phi_{1,3}(z_1,\bar{z}_1)\Phi_{1,3}(z_2,\bar{z}_2)\Phi_{1,3}(z_3,\bar{z}_3) \rangle = C_{\Phi_{1,3}\Phi_{1,3}\Phi_{1,3}}(c)\mathbb{P}_3^c, \tag{195}$$

with $C_{\Phi_{1,3}\Phi_{1,3}\Phi_{1,3}}(c)$ given in the appendix (eq. (A.28)). From here, we get the three-point constants

$$C_{\varepsilon\varepsilon\varepsilon}^{\text{SAW}} = 0, \qquad C_{\varepsilon\varepsilon\tilde{\varepsilon}}^{\text{SAW}} = \frac{40\sqrt{5}\pi^{5/4}\Gamma\left(\frac{5}{6}\right)}{\Gamma\left(\frac{1}{6}\right)^{7/2}}. \tag{196}$$

Again we find that the three-point constants of the bottom fields vanish but the generic $c$ coupling (195) gives rise to a non-vanishing coupling $C_{\varepsilon\varepsilon\tilde{\varepsilon}}^{\text{SAW}}$ in (196). The OPE of $\varepsilon^{\text{SAW}}$ takes a similar form to (182), with the additional contribution:

$$\varepsilon^{\text{SAW}}(z,\bar{z})\varepsilon^{\text{SAW}}(0,0) = \ldots + (z\bar{z})^{-h_{\varepsilon}^{\text{SAW}}}\frac{C_{\varepsilon\varepsilon\tilde{\varepsilon}}^{\text{SAW}}}{\gamma^{\text{SAW}}}\varepsilon^{\text{SAW}} + \ldots \tag{197}$$

## 5.2 Higher Kac operators

Although in this paper we focus on the simplest Kac operators – the energy operator $\varepsilon \sim \hat{\Phi}_{2,1}$, it is interesting to further comment on higher Kac operators, in particular, the second energy operator $\varepsilon' \sim \hat{\Phi}_{3,1}$ in percolation that sits at the bottom of a rank-3 Jordan block. The conclusion from the previous section that all Kac operators acquire zero norms clearly applies here. However, as we have seen in section 3, the zero in the norm of $\hat{\Phi}_{3,1}$ at $c = 0$ should be second order. This physically means that the two-point correlation of the second energy vanishes even faster as $c \to 0$ compared to the energy operator $\varepsilon$. To see this double-zero normalization in a slightly different way, consider now the generic $c$ OPE:

$$\Phi_{2,1}(z,\bar{z})\Phi_{3,1}(0,0) = (z\bar{z})^{-2h_{2,1}(c)}\Big((z\bar{z})^{h_{2,1}(c)}D_{\Phi_{2,1}\Phi_{3,1}\Phi_{2,1}}(c)\Phi_{2,1} + \ldots$$
$$+ (z\bar{z})^{h_{4,1}(c)}D_{\Phi_{2,1}\Phi_{3,1}\Phi_{4,1}}(c)\Phi_{4,1} + \ldots\Big). \tag{198}$$

Using the analytic bootstrap equation, one can find

$$\frac{C_{\hat{\Phi}_{2,1}\hat{\Phi}_{3,1}\hat{\Phi}_{2,1}}^2}{B_{\hat{\Phi}_{2,1}}^2 B_{\hat{\Phi}_{3,1}}} = \frac{\Gamma\left(\frac{1}{\beta^2}\right)^2\Gamma\left(1-\frac{2}{\beta^2}\right)^2}{\left(2\cos\left(\frac{2\pi}{\beta^2}\right)+1\right)\Gamma\left(2-\frac{3}{\beta^2}\right)^2\Gamma\left(\frac{2}{\beta^2}-1\right)^2}. \tag{199}$$

Since we have established the energy logarithmic pair $(\tilde{\varepsilon},\varepsilon)$ in (135), we can now consider the three-point function

$$\langle \tilde{\varepsilon}^{\text{perco}}(z_1,\bar{z}_1)\varepsilon^{\text{perco}}(z_2,\bar{z}_2)\Phi_{3,1}(z_3,\bar{z}_3) \rangle = \frac{\gamma^{\text{perco}}}{B_{\hat{\Phi}_{2,1}}(c)}\langle \hat{\Phi}_{2,1}(z_1,\bar{z}_1)\hat{\Phi}_{2,1}(z_2,\bar{z}_2)\Phi_{3,1}(z_3,\bar{z}_3) \rangle$$
$$\stackrel{c\to 0}{\sim} \left(\beta^2-\frac{2}{3}\right)^{-1}. \tag{200}$$

We see that the only way to have the three-point function finite is to have the normalization of $\varepsilon \sim \hat{\Phi}_{3,1}$ to go at least as

$$B_{\hat{\Phi}_{3,1}} \overset{c \to 0}{\sim} \left(\beta^2 - \frac{2}{3}\right)^2. \tag{201}$$

Recall that $\hat{\Phi}_{3,1}$ can be identified as the bottom field $\Psi_0$ just as $T\bar{T}$, we then recover the result from the previous section

$$\langle \varepsilon^{\text{perco}}(z_1, \bar{z}_1) \varepsilon^{\text{perco}}(z_2, \bar{z}_2) \Psi_0(z_3, \bar{z}_3) \rangle = 0, \tag{202a}$$

$$\langle \tilde{\varepsilon}^{\text{perco}}(z_1, \bar{z}_1) \varepsilon^{\text{perco}}(z_2, \bar{z}_2) \Psi_0(z_3, \bar{z}_3) \rangle = C^{\text{perco}}_{\varepsilon\tilde{\varepsilon}\Psi_0} \mathbb{P}^0_3, \tag{202b}$$

with

$$C^{\text{perco}}_{\varepsilon\tilde{\varepsilon}\Psi_0} = \gamma^{\text{perco}} \left(h_\varepsilon^{\text{perco}}\right)^2 = -\frac{125}{256}. \tag{203}$$

At $c = 0$, the leading terms in the OPE (198) is given by

$$\varepsilon^{\text{perco}}(z, \bar{z}) \Psi_0(0, 0) = (z\bar{z})^{-2} \frac{C^{\text{perco}}_{\varepsilon\tilde{\varepsilon}\Psi_0}}{\gamma^{\text{perco}}} \varepsilon^{\text{perco}} + \ldots \tag{204}$$

Consider further the generic $c$ degenerate OPE of the second energy operator:

$$\begin{aligned}
\Phi_{3,1}(z, \bar{z}) \Phi_{3,1}(0, 0) = (z\bar{z})^{-2h_{3,1}(c)} \bigg( &1 + \frac{h_{3,1}(c)}{c/2} \left(z^2 T + \bar{z}^2 \bar{T}\right) + \frac{h_{3,1}^2(c)}{c^2/4} (z\bar{z})^2 T\bar{T} + \ldots \\
&+ D_{\Phi_{3,1}\Phi_{3,1}\Phi_{3,1}} (z\bar{z})^{h_{3,1}(c)} \Phi_{3,1} + \ldots \\
&+ D_{\Phi_{3,1}\Phi_{3,1}\Phi_{5,1}} (z\bar{z})^{h_{5,1}(c)} \Phi_{5,1} + \ldots \bigg).
\end{aligned} \tag{205}$$

According to our analysis, the properly normalized operator $\varepsilon' \sim \hat{\Phi}_{3,1}$ becomes the bottom field $\Psi_0$ of the rank-3 Jordan block. We have fixed the normalization of $\hat{\Phi}_{3,1}$ as in (90). We can now compute some three-point functions of $\Psi_0$ using the same strategy as before and find

$$\langle \Psi_0(z_1, \bar{z}_1) \Psi_0(z_2, \bar{z}_2) \Psi_0(z_3, \bar{z}_3) \rangle = 0, \tag{206a}$$

$$\langle \Psi_0(z_1, \bar{z}_1) \Psi_0(z_2, \bar{z}_2) \Psi_2(z_3, \bar{z}_3) \rangle = C_{\Psi_0\Psi_0\Psi_2} \mathbb{P}^0_3, \tag{206b}$$

where

$$C_{\Psi_0\Psi_0\Psi_1} = 0, \quad C_{\Psi_0\Psi_0\Psi_2} = 4a. \tag{207}$$

This results in the $c = 0$ OPE of the second energy operator $\varepsilon' \sim \Psi_0$:

$$\Psi_0(z, \bar{z}) \Psi_0(0, 0) = (z\bar{z})^{-4} \left(4\Psi_0 + \ldots\right). \tag{208}$$

Above in (176), (179b), (196) and (206a), we have observed something curious: the three-point structure constants of the bottom fields of Jordan blocks in general vanish. It is also straightforward to see that the same applies to higher Kac operators: $\hat{\Phi}_{4,1}, \hat{\Phi}_{5,1}, \ldots$ which necessarily sit at the bottom of (possibly higher-rank) Jordan blocks. From analytic bootstrap equations, one can easily obtain for example:

$$\frac{C^2_{\hat{\Phi}_{2,1}\hat{\Phi}_{3,1}\hat{\Phi}_{4,1}}}{B_{\hat{\Phi}_{2,1}} B_{\hat{\Phi}_{3,1}} B_{\hat{\Phi}_{4,1}}} = \frac{2\Gamma\left(\frac{4}{\beta^2} - 1\right)^2 \Gamma\left(\frac{1}{\beta^2}\right)^2}{\left(\sec\left(\frac{2\pi}{\beta^2}\right) + 2\right) \Gamma\left(\frac{2}{\beta^2} - 1\right)^2 \Gamma\left(\frac{3}{\beta^2}\right)^2} \overset{c \to 0}{\sim} O(1), \tag{209a}$$

$$\frac{C^2_{\hat{\Phi}_{3,1}\hat{\Phi}_{3,1}\hat{\Phi}_{5,1}}}{B^2_{\hat{\Phi}_{3,1}} B_{\hat{\Phi}_{5,1}}} = \frac{(\beta^2 - 2)^2 \Gamma\left(2 - \frac{3}{\beta^2}\right) \Gamma\left(\frac{5}{\beta^2} - 1\right) \Gamma\left(\frac{1}{\beta^2}\right) \Gamma\left(\frac{\beta^2 - 3}{\beta^2}\right)}{(\beta^2 - 4)^2 \Gamma\left(2 - \frac{5}{\beta^2}\right) \Gamma\left(\frac{3}{\beta^2} - 1\right) \Gamma\left(\frac{3}{\beta^2}\right) \Gamma\left(\frac{\beta^2 - 1}{\beta^2}\right)} \overset{c \to 0}{\sim} O(1). \tag{209b}$$

Therefore, despite the unclear situation of the order of zeros in the norms of higher Kac operators at $c = 0$ (eq. (91),(92)), it is clear that the three-point constants with other bottom fields vanish:

$$C_{\varepsilon\Psi_0\hat{\Phi}_{4,1}} = C_{\Psi_0\Psi_0\hat{\Phi}_{5,1}} = 0\,. \tag{210}$$

The vanishing of the three-point couplings among the Kac operators which arise from energy-type lattice operators seems natural from lattice point of view. In the case of percolation, the bonds are independent and it makes sense for the energy-type operators describing local bond occupations to have no long-range correlations. This result is however mysterious from the CFT point of view. The vanishing three-point functions are clearly not required by conformal symmetry. In appendix D.2, we give some expressions of three-point functions involving rank-2 and rank-3 Jordan blocks as a result of conformal Ward identities. See eqs. (D.23)-(D.28). In the case of three-point functions involving two operators from a rank-2 Jordan block and another one from a rank-$k$ Jordan block with even spin, there should be $3k$ independent three-point constants after the reduction by Bose symmetry [51]. Note that the vanishing three-point couplings have interesting effects on the operator algebra and correlation functions. As we have seen in e.g. (182), the OPE of the bottom field only sees part of the full logarithmic structures of higher rank Jordan block, and as we will see below, their four-point function, while not vanishing, has less logarithmic dependence than the case of non-logarithmic operators. From the calculations, we see that the vanishing couplings appear to be a remnant effect of the Virasoro degeneracy of the Kac operators at generic $c$. It would be important in the future to understand this result from an algebraic perspective, in particular, whether the algebraic structure behind this "remnant Virasoro degeneracy" could give us a systematic analytic handle on the percolation and SAW bulk CFTs with $c = 0$. On the other hand, it would also be interesting to study this vanishing coupling from a probabilistic perspective to understand its geometrical origin.

## 5.3 Four-point function

Using the previously computed conformal data at $c = 0$, we can now consider the four-point function of the bottom field of a Jordan block and study its $s$-channel limit. As mentioned before, such bottom fields can be taken in general to be the properly normalized Kac operators. As Cardy pointed out in [5] and we have also seen in the previous subsection, it is necessary that all Kac operators acquire zero norm at $c = 0$ to resolve the "$c \to 0$ catastrophe". However, the zero of the norm is not always simply first order as previous thought [25,28]: for example, as we have seen in both section 3.2 and section 5.2 from different perspectives, the second energy operator $\varepsilon' \sim \Phi_{3,1}$ in the cluster model acquires a second order zero in its norm at $c = 0$, leading to a mix into a rank-3 Jordan block, and higher Kac operators might have even higher order zeros in the norm as argued at the end of section 3.4.

With the vanishing norm for Kac operators established, it was further concluded in [5] that higher point correlation functions of the bottom fields in Jordan blocks should all vanish at $c = 0$. This seems to be a reasonable statement from the lattice perspective. Take the energy operator for example. In percolation, the lattice energy operator is defined on two neighboring sites and represents bond occupation. Since percolation is a problem of random, uncorrelated bonds, the energy operator should have no long-distance correlations.[19] In the case of four-point functions, this also appears to arise straightforwardly if we consider a naive $c \to 0$ limit of the generic $c$ CFT four-point functions of Kac operator $\varepsilon \sim \Phi_{2,1}$ in cluster model. At generic $c$, its four-point function satisfies the BPZ equation and is given by

$$\langle\Phi_{2,1}\Phi_{2,1}\Phi_{2,1}\Phi_{2,1}\rangle = |\mathcal{F}_{\mathbb{I}}(z)|^2 + R(c)|\mathcal{F}_{3,1}(z)|^2\,, \tag{211}$$

---

[19]We thank Jesper Lykke Jacobsen for bringing up this point and for interesting discussions.

where $\mathcal{F}_\mathbb{I}, \mathcal{F}_{3,1}$ are the conformal blocks of identity operator $\Phi_{1,1}$ and second energy operator $\Phi_{3,1}$ given in terms of hypergeometric functions:

$$\mathcal{F}_\mathbb{I}(z) = (1-z)^{1-\frac{3}{2\beta^2}} z^{1-\frac{3}{2\beta^2}} F\left(2 - \frac{3}{\beta^2}, 1 - \frac{1}{\beta^2}, 2 - \frac{2}{\beta^2}, z\right), \tag{212}$$

$$\mathcal{F}_{3,1}(z) = (1-z)^{1-\frac{3}{2\beta^2}} z^{\frac{1}{2\beta^2}} F\left(1 - \frac{1}{\beta^2}, \frac{1}{\beta^2}, \frac{2}{\beta^2}, z\right). \tag{213}$$

To connect more closely to the OPE (170), we can write down the $s$ channel limit of the identity non-chiral block

$$|\mathcal{F}_\mathbb{I}(z)|^2 = 1 + \frac{h_{2,1}^2(c)}{c/2}(z^2 + \bar{z}^2) + \frac{h_{2,1}^4(c)}{c^2/4}(z\bar{z})^2 + \dots, \tag{214}$$

where the expansion coefficients come from the conformal data

$$\frac{C_{\Phi_{2,1}\Phi_{2,1}T}^2(c)}{B_T(c)} = \frac{h_{2,1}^2(c)}{c/2}, \qquad \frac{C_{\Phi_{2,1}\Phi_{2,1}T\bar{T}}^2(c)}{B_{T\bar{T}}(c)} = \frac{h_{2,1}^4(c)}{c^2/4}. \tag{215}$$

The $R$ in (211) is given by

$$R(c) = \frac{C_{\Phi_{2,1}\Phi_{2,1}\hat{\Phi}_{3,1}}^2(c)}{B_{\hat{\Phi}_{3,1}}(c)}. \tag{216}$$

See the expression in (A.22). Now further take into account the proper normalization $B_{\hat{\Phi}_{2,1}}$ (see eq. (136)), we get the generic $c$ four-point function in the $s$-channel limit:

$$\langle \hat{\Phi}_{2,1} \hat{\Phi}_{2,1} \hat{\Phi}_{2,1} \hat{\Phi}_{2,1} \rangle = (z\bar{z})^{-2h_{2,1}(c)} \left\{ B_{\hat{\Phi}_{2,1}}^2(c) + \frac{B_{\hat{\Phi}_{2,1}}^2(c) h_{2,1}^2(c)}{c/2}(z^2 + \bar{z}^2) + \frac{C_{\hat{\Phi}_{2,1}\hat{\Phi}_{2,1}T\bar{T}}^2(c)}{B_{T\bar{T}}(c)}(z\bar{z})^2 \right.$$
$$\left. + \frac{C_{\hat{\Phi}_{2,1}\hat{\Phi}_{2,1}\hat{\Phi}_{3,1}}^2(c)}{B_{\hat{\Phi}_{3,1}}(c)}(z\bar{z})^{h_{3,1}(c)} + \dots \right\}. \tag{217}$$

It is straightforward to see that the first two terms vanish at $c = 0$. Furthermore, as $c \to 0$, the third and fourth term cancel at $O(1)$ and therefore, up to $(z\bar{z})^2$ term, the four-point function indeed vanishes at $c = 0$. Note that at the order $O(c)$, there is a logarithmic dependence:[20]

$$\langle \hat{\Phi}_{2,1} \hat{\Phi}_{2,1} \hat{\Phi}_{2,1} \hat{\Phi}_{2,1} \rangle \overset{c \to 0}{=} (z\bar{z})^{-2h_\varepsilon + 2} \left( \frac{h_\varepsilon^4}{2b_{1,2}} \ln(z\bar{z}) + \alpha \right) c + \dots, \tag{218}$$

but this clearly does not survive at $c = 0$ to give any non-trivial four-point correlation. In [28], this led to the conclusion that non-zero physical correlations are in the derivatives with respect to $c$ at $c = 0$.

However, we can now build the four-point function in an alternative way using the exact conformal data at $c = 0$ we obtained previously. In doing this, we assume that cluster decomposition holds as in the case of spin operators we examined in previous sections.[21] This should be the case since all the operator dimensions are positive. The $s$-channel limit of the four-point functions can be assembled in the usual way: Place the operators $\varepsilon^{\mathrm{perco}}$ at the most convenient configuration $(\infty, 1, z, 0)$:

$$\langle \varepsilon(\infty, \infty) \varepsilon(1, 1) \varepsilon(z, \bar{z}) \varepsilon(0, 0) \rangle^{\mathrm{perco}} = \lim_{z_1, \bar{z}_1 \to \infty} (z_1 \bar{z}_1)^{2h_\varepsilon^{\mathrm{perco}}} \langle \varepsilon(z_1, \bar{z}_1) \varepsilon(1, 1) \varepsilon(z, \bar{z}) \varepsilon(0, 0) \rangle^{\mathrm{perco}}$$
$$= \sum_{\{\psi\}} \langle \varepsilon^{\mathrm{perco}} | \varepsilon^{\mathrm{perco}}(1, 1) | \psi \rangle G^{-1} \langle \psi | \varepsilon^{\mathrm{perco}}(z, \bar{z}) | \varepsilon^{\mathrm{perco}} \rangle, \tag{219}$$

---

[20]Here $\alpha = 2h_\varepsilon^2 \left( 2B_{\hat{\Phi}_{2,1}}^{(2)} h_\varepsilon^2 + 2h_\varepsilon h_\varepsilon' - 2h_\varepsilon^2 B_{\hat{\Phi}_{3,1}}^{(3)} - C_{\hat{\Phi}_{2,1}\hat{\Phi}_{2,1}\hat{\Phi}_{3,1}}'' \right)$.

[21]In [21], it was shown that the four-point function of the percolation spin operator as assembled from the logarithmic conformal data at $c = 0$ agrees with taking the $c \to 0$ limit of the generic $c$ four-spin correlator.

where $G^{-1}$ is the inverse of Gram matrix and $\{\psi\}$ contains the following states we have identified:

$$\{\psi\} : \mathbb{I}, \varepsilon, \tilde{\varepsilon}, T, \bar{T}, t, \bar{t}, \partial \bar{t}, \bar{\partial} t, \partial^2 \bar{t}, \bar{\partial}^2 t, \Psi_0, \Psi_1, \Psi_2, \dots \tag{220}$$

Recall that

$$\langle \varepsilon^{\text{perco}} | \varepsilon^{\text{perco}}(1,1) | \psi \rangle = \lim_{z_1, \bar{z}_1 \to \infty} \langle \varepsilon(z_1, \bar{z}_1) \varepsilon(1,1) \psi(0,0) \rangle^{\text{perco}}, \tag{221}$$

$$\langle \psi | \varepsilon^{\text{perco}}(z, \bar{z}) | \varepsilon^{\text{perco}} \rangle = \lim_{w, \bar{w} \to \infty} \langle \psi(w, \bar{w}) \varepsilon(z, \bar{z}) \varepsilon(0,0) \rangle^{\text{perco}}. \tag{222}$$

Using the three-point functions we obtained in the previous section, (219) can be computed straightforwardly. We give the detailed expression in (D.65). In this case, we see that there is a non-vanishing contribution from

$$\langle \varepsilon^{\text{perco}} | \varepsilon^{\text{perco}}(1,1) | \Psi_1 \rangle \frac{1}{a} \langle \Psi_1 | \varepsilon^{\text{perco}}(z, \bar{z}) | \varepsilon^{\text{perco}} \rangle, \tag{223}$$

giving rise to the four-point function in the $s-$channel limit:

$$\langle \varepsilon(\infty) \varepsilon(1) \varepsilon(z, \bar{z}) \varepsilon(0) \rangle^{\text{perco}} = (z\bar{z})^{-2h_\varepsilon^{\text{perco}}} \left\{ \frac{\left( C_{\varepsilon\varepsilon\Psi_1}^{\text{perco}} \right)^2}{a} (z\bar{z})^2 + \dots \right\}$$
$$= (z\bar{z})^{-2h_\varepsilon^{\text{perco}}} \left\{ \frac{a \left( h_\varepsilon^{\text{perco}} \right)^4}{\left( b_{1,2}^{\text{perco}} \right)^2} (z\bar{z})^2 + \dots \right\}, \tag{224}$$

and this does not vanish! Evidently the four-point correlation is rendered non-trivial by the non-vanishing three-point coupling $C_{\varepsilon\varepsilon\Psi_1}^{\text{perco}}$ between the energy operator $\varepsilon^{\text{perco}}$ and the middle field $\Psi_1$ of rank-3 Jordan block associated with the second energy operator $\varepsilon'$. It should be stressed that the existence of a higher rank ($> 2$) Jordan block in the intermediate channel is crucial in building this non-vanishing four-point correlation for $\varepsilon^{\text{perco}}$. As a comparison, let us consider instead the rank-2 Jordan block $(\Theta, \hat{\Phi}_{3,1})$ at $O(c)$ we encountered in section 2.2.2 when building the rank-3 Jordan block (see also appendix C). Using the definition (43), we can compute its three-point couplings with the energy operator $\varepsilon$ and find

$$C_{\varepsilon\varepsilon T\bar{T}} = \frac{h_\varepsilon^4}{2b_{1,2}} c + o(c), \qquad C_{\varepsilon\varepsilon\Theta} = -b_{1,2} \left( C''_{\hat{\Phi}_{2,1}\hat{\Phi}_{2,1}\hat{\Phi}_{3,1}} - 2h_\varepsilon^2 B^{(2)}_{\hat{\Phi}_{2,1}} - 2h_\varepsilon h'_\varepsilon \right) c + o(c). \tag{225}$$

Consider now the four-point function mediated by this rank-2 Jordan block $(\Theta, T\bar{T})$. Cluster decomposition gives

$$\langle \varepsilon(\infty) \varepsilon(1) \varepsilon(z, \bar{z}) \varepsilon(0) \rangle = (z\bar{z})^{-2h_\varepsilon + 2} \left( \frac{C_{\varepsilon\varepsilon T\bar{T}}(2\theta_0 C_{\varepsilon\varepsilon\Theta} - C_{\varepsilon\varepsilon T\bar{T}}\theta_1)}{\theta_0^2} + \frac{C_{\varepsilon\varepsilon T\bar{T}}^2}{\theta_0} \ln(z\bar{z}) \right) + \dots, \tag{226}$$

where $\theta_0, \theta_1$ denotes the 2-point constants from

$$\langle \Theta(z, \bar{z}) \Theta(0,0) \rangle = \left( -2\theta_0 \ln(z\bar{z}) + \theta_1 \right) \mathbb{P}_2^0$$
$$= \left( -b_{1,2} c \ln(z\bar{z}) + 4b_{1,2}^2 B^{(3)}_{3,1} c \right) \mathbb{P}_2^0. \tag{227}$$

Clearly, (226) vanishes at $c = 0$ and it is also straightforward to see that this recovers precisely the order $O(c)$ terms from (218). So the naive BPZ solution (218) agrees with the computation (226) based on cluster decomposition involving only the rank-2 Jordan block $(\Theta, \hat{\Phi}_{3,1})$. The interpretation is that the degenerate Kac operator $\hat{\Phi}_{2,1}$ at generic $c$ only contains the information of the operators $T\bar{T}, \hat{\Phi}_{3,1}$ and therefore can only "know about" the intermediate rank-2

Jordan block $(\Theta, \hat{\Phi}_{3,1})$, the coupling to which is not enough to build a non-trivial four-point function at $c = 0$.

The middle field $\Psi_1$ of rank-3 Jordan block however arises from further mixing $(\Theta, \hat{\Phi}_{3,1})$ with the generic $c$ rank-2 Jordan block $(\hat{\Psi}, \bar{A}\hat{X})$. Note that since

$$(L_0 - 2)\Theta = (\bar{L}_0 - 2)\Theta = \hat{\Phi}_{3,1}, \qquad (L_0 - 2)\Psi_1 = (\bar{L}_0 - 2)\Psi_1 = \Psi_0 = \hat{\Phi}_{3,1}, \qquad (228)$$

both fields $\Theta$ and $\Psi_1$ can be considered as "the top field" of a rank-2 Jordan block for $\hat{\Phi}_{3,1}$, although the pair $(\Psi_1, \hat{\Phi}_{3,1})$ is really part of a higher logarithmic structure through the mixing with $(\hat{\Psi}, \bar{A}\hat{X})$. This further mixing has an interesting effect that allows $\Psi_1$ a stronger coupling to the energy operator $\varepsilon$ compared to $\Theta$. To get some intuition on this, we can rewrite the expression (52) as

$$\Psi_1 = \frac{2a}{b_{1,2}c} T\bar{T} - \frac{a}{b_{1,2}^2}\Theta - 2ah_{3,1}'' T\bar{T} + \frac{ah_{3,1}'' c}{b_{1,2}}\Theta + \dots, \qquad (229)$$

where we have used the definition (43). Note the difference term $\frac{2a}{b_{1,2}c} T\bar{T}$ between $\Psi_1$ and $\Theta$ which arise from this mixing, and this is precisely the term that has a non-trivial coupling $C_{\varepsilon\varepsilon\Psi_1}^{\text{perco}}$ which we computed in (180). This implies that the energy operator $\varepsilon^{\text{perco}}$ in percolation, unlike its generic $c$ ancestor, "knows about" the other rank-2 Jordan block $(\hat{\Psi}, \bar{A}\hat{X})$. While this may seems puzzling at first, it makes sense considering that at $c = 0$, the energy operator $\varepsilon$ becomes degenerate with the hull operator $\hat{\Phi}_{0,2}$ by logarithmic mixing, and the operator algebra of the latter certainly contains information about the Jordan block $(\hat{\Psi}, \bar{A}\hat{X})$. From the physical point of view, while the energy operator itself describes bond occupations that have no long-range correlations, its degeneracy with the hull operator that describes cluster boundaries allows the possibility of non-trivial correlations at the geometrical critical point of percolation. With this interesting intuition, it would be important to clarify this point in more details, and this would require examining the four-point function of $\hat{\Phi}_{0,2}$ which is a more complicated problem. Algebraically, the question also involves an understanding of the fate of the Virasoro degeneracy of $\Phi_{2,1}$ (and other Kac operator) which requires a detail analysis of Virasoro algebra at $c = 0$. We leave these explorations for future work. Let us note that a completely analogous computation can be done for the $s$-channel four-point function of $\varepsilon^{\text{SAW}}$ with the same conclusion.

Before ending this section, let us remark that the possibility of zero-norm operators to have non-trivial higher-point correlations has been brought up in the Discussion section of [51] which in particular disagrees with the claim of [52]. Our results support the former although the situation seems more subtle than described in [51]: While the vanishing norm indicates the lack of the identity operator therein, the OPE of $\varepsilon$ (182) does contain fields $\Psi_0, \Psi_1$ from the identity Virasoro module, but only sees part of the full complicated structure [21].

# 6  Conclusions

In this work, we studied the resolutions of "$c \to 0$ catastrophe" in the context of percolation and SAW bulk CFTs at $c = 0$. We start with revisiting the appearance of rank-2 and rank-3 Jordan blocks associated with $T$ and $T\bar{T}$ by analyzing the OPE of spin operators at $c = 0$. While the rank-2 Jordan block $(t, T)$ was known for a long time, the rank-3 Jordan block in the bulk $c = 0$ CFTs was only recently uncovered [21] and as we see in this paper, this leads to interesting new insights on CFT correlations functions in critical percolation and SAW. A fresh point of view we have taken here in building the Jordan blocks is the proper normalization of operators: The appearance of zero-norm states at particular central charges such as $c = 0$

– when special conditions are met – naturally leads to logarithmic mixings. In particular, the rank-3 Jordan block arises from operators $T\bar{T}$ and $\varepsilon' \sim \hat{\Phi}_{3,1}$ acquiring a double zero in their norms at $c = 0$ and can be intuitively understood as mixing two rank-2 Jordan blocks at order $O(c)$. We then turned to our main focus of the paper – the fate of Kac operators at $c = 0$ that describe energy-type operators in percolation and SAW. This requires to first understand their proper normalizations for which we went back to the families of cluster and loop model CFTs at generic $c$. We resort to the physical principle that CFTs of geometrical models should be real as a result of the underlying microscopic description with real and positive measure. Using the analytic information on Kac operator that is completely under control, we deduced the normalizations of Kac operators as functions of the central charge $c$ by requesting that the three-point constants with the spin operator are finite and real. Interestingly, Kac operators were seen to acquire zero norms at $c = 0$ in general where the orders of the zeros can be higher for higher Kac operators. The zero norms indicate mixing into Jordan blocks at $c = 0$ and the orders of zeros suggest the ranks of the Jordan blocks. We compared the deduced normalizations with the closely-related $c < 1$ Liouville CFT and discussed the possible existence of arbitrarily higher rank Jordan blocks in $c = 0$ CFTs. With these insights, we then gave a generic construction of logarithmic multiplets where Kac operators sit at the bottom of the Jordan blocks. From here, conformal data like logarithmic couplings can be computed using the conformal Ward identities. We then focused on the simplest Kac operator – the energy density operator $\varepsilon$ and studied its two- and three-point functions. The results were then used to write down the operator algebra of energy operator involving the rank-2 and rank-3 Jordan blocks of $T, T\bar{T}$, thus resolving the "$c \to 0$ catastrophe". Based on cluster decomposition, we used these data to compute the $s$-channel limit of the energy operataor four-point function in percolation. Curiously, the four-point function does not vanish as previously believed or what one might expect from the lattice insights. A crucial role is played by the coupling to the rank-3 Jordan block which allows to build long-distance multi-point correlations of the zero-norm energy operator in percolation (and SAW).

There are several interesting points uncovered in this work which we think are worth further investigating.

**Operator normalizations**   The reality of the CFTs describing geometrical models means that there exist real operators whose correlation functions are real. Thus, despite the lack of unitarity/reflection positivity, the non-unitary CFTs we are dealing with here are not completely out of control. In this paper, we deduced the norms for Kac operators using the known analytic information on their amplitudes in the four-spin correlators. This may generalize to other operators and other correlation functions and deserves further studies. The examination of additional correlation functions beyond spin correlator could also shed light on some ambiguous situation involve Kac operator. In particular, we have seen in eqs. (91), (92) higher Kac operators might acquire higher order zero norm at $c = 0$ and this is supported by the comparison with the operator normalizations in $c < 1$ Liouville CFT. If this is indeed the case, there may exist Jordan blocks of arbitrarily high ranks [47] in the $c = 0$ CFTs whose effect on correlation functions would be fascinating to understand. Note that the ambiguous deduction of these norms from the four-spin correlator means that, even if these arbitrarily high rank Jordan block do exist, the higher logarithmic structure would be invisible from the spin operator algebra. The four-spin correlation can only probe the bottom rank-2 part of such high rank logarithmic structure, due to the vanishing couplings $C_{\sigma\sigma\hat{\Phi}_{4,1}}, C_{\sigma\sigma\hat{\Phi}_{4,1}}$ there. The question is then, where can we possibly see these higher structures? The answer is suggested by the fusion rule of Kac operators. Consider a potential rank-4 Jordan block whose bottom field is the third energy operatorr $\hat{\Phi}_{4,1} \sim \varepsilon''$ in percolation. Due to degenerate fusion

$$\hat{\Phi}_{2,1} \times \hat{\Phi}_{3,1} \sim \hat{\Phi}_{2,1} + \hat{\Phi}_{4,1}, \tag{230}$$

the Jordan block of $\hat{\Phi}_{4,1}(\varepsilon'')$ should couple non-trivially to those of $\hat{\Phi}_{2,1}(\varepsilon)$ and $\hat{\Phi}_{3,1}(\varepsilon')$, namely the first two energy operators. This of course does not mean that the coupling of the bottom fields $C_{\varepsilon\varepsilon'\varepsilon''}$ itself is non-trivial. In fact, we have seen in section 4.3 that this three-point coupling vanishes – a fact that makes sense physically although algebraically still remains to be understood (see the next part). The operator algebra of the top fields $\tilde{\varepsilon}$ and $\Psi_2$ would contain the full structure of the Jordan block structure of $\hat{\Phi}_{4,1}$. This also suggests that the proper normalization may be investigated by examining the four-point functions of the fields making up $\tilde{\varepsilon}$ and $\Psi_2$ from the generic $c$ point of view, that of the 2-hull operator $\hat{\Phi}_{0,2}$ for instance. Perhaps the recent extraction of analytic expressions in loop models [18] could help clarify some of these questions. Finally, it is of course desirable to understand if there is a way to compute these norms and understand the appearance of negative norm states from first principles. Such an understanding might provide a handle to study the non-unitary CFTs with the modern bootstrap techniques.

**Vanishing three-point constants**    As pointed out at the end of section 5.2, all the results we have found give vanishing three-point couplings for the bottom fields of Jordan blocks. This appears to be reasonable from the intuitions on the lattice. Since percolation and SAW are geometrical type phase transitions, the fluctuation of bond occupations described by energy density-type operators are random and have no long-distance correlations. On the other hand, from the CFT perspective, the vanishing three-point couplings are certainly not required by the conformal Ward identities (see appendix D.2) and calls for an algebraic explanation. From the calculations we have done, we see that this appears to be a consequence of the normalization of the bottom fields as a result of the Virasoro degeneracy. Recall that such degeneracy of the fields $\Phi_{2,1}$ or $\Phi_{1,2}$ are related to an interchiral symmetry in loop models [15]. It would be interesting to investigate the fate of the Virasoro degeneracy and thus the interchiral symmetry at $c = 0$. The main motivation here is to see what kind of analytic tool this could provide to further study the $c = 0$ bulk CFTs

Related to this, it is also curious to observe that such vanishing three-point couplings render the four-point functions of e.g. the energy density operator (at least in its leading $s$−channel expansion) non-logarithmic. In comparison, such logarithmic terms do appear in the spin four-point functions. Recently, [49] based on a probabilistic approach gave a very interesting geometrical interpretation the logarithm in correlation functions as a result of summing independent events with equal contributions at different scales. In the case of the energy operator four-point function, we can then reverse to ask for an interpretation of the lack of logarithm, and how this is related to the above mentioned algebraic explanation we are searching for.

**Non-trivial four-point function of energy-type operator**    The most intriguing result of this paper is the non-trivial four-point function of the energy operator $\varepsilon$ we saw in section 5.3. Let us remark that to get this, we have assumed two things: First, once the logarithmic operators are defined, the $c = 0$ conformal data – two- and three-point constants – can be obtained by taking the $c \to 0$ limit and keeping the finite terms. This seems to be reasonable, as checked by various cancellation conditions and the consistency with conformal Ward identities. Second, we assume that cluster decomposition holds at $c = 0$, also reasonable since all conformal dimensions are positive. With these two assumptions, we were able to assemble the energy four-point function from the logarithmic conformal data and this leads to a non-trivial result. This result is simultaneously puzzling and suggestive. The puzzle part is that at the geometrical type of transition like percolation and SAW, energy operator that characterizes bond occupations on the lattice are random and independent, so the non-trivial correlation contradicts this basic intuition from the lattice. In the meantime, the non-vanishing four-point function also disagrees with a direct $c \to 0$ limit of the generic $c$ four-point function of the Kac operator

$\Phi_{2,1}$ from BPZ equation. Nevertheless, if we believe the computations done there, the result then suggests some intriguing nature of higher-point correlations in percolation (and other $c = 0$ bulk CFTs) that we might be uncovering. The energy density operator is degenerate with the hull operators at $c = 0$ and the two logarithmically mix into a rank-2 Jordan block. The long-range correlations of the latter certainly do not vanish at $c = 0$ since they are closely associated with cluster boundaries or polymer lines. It is important to note that the non-trivial four-energy correlation is built by coupling with a rank-3 Jordan block, a newly discovered property [21] of $c = 0$ bulk CFTs that is still under lattice examination and might appear only in the continuum limit [53]. While the lattice effort is difficult, an alternative possibility is to directly analyze the geometrical configurations in the continuum using the probabilistic approach as in [48]. In either approach, a crucial point is to understand the precise definition of the energy and other Kac operators: While we have been referring to the Kac operator $\hat{\Phi}_{2,1}$ as the energy operator from usual CFT understanding, the correspondence between these CFT operators and operators one can construct from the lattice or identify in the probabilistic construction is a complicated issue. For the former, see e.g. [54] for such an identification from studying the theory on the torus. It would be interesting to first identify the rank-3 Jordan block associated with $\varepsilon' \sim \hat{\Phi}_{3,1}$ or $T\bar{T}$, and then analyze its role in the four-point correlation of $\varepsilon \sim \hat{\Phi}_{2,1}$ we studied in this paper to check our finding here. Such investigation will also provide a geometrical picture of the observable we are computing here. Furthermore, it seems reasonable to believe that similar situations may occur for correlation functions of higher rank Jordan blocks as well. For example, higher energy operators (i.e. higher Kac operators), despite having vanishing norms at $c = 0$, might also possess long-range high-point correlations, perhaps through coupling with even higher rank Jordan blocks. It would be fascinating to understand and fully appreciate the physical properties this is telling us about percolation and SAW CFTs, and how general this is for $c = 0$ bulk CFTs.

# Acknowledgments

I am grateful to H. Saleur for introducing to me the fascinating topic of $c = 0$ CFTs, for collaboration on previous work [21], and for insightful discussions. I would like to thank the participants of the 28th Rencontre Itzykson "Analytic results in Conformal Field Theory", in particular J. L. Jacobsen, S. Ribault, S. Rychkov and R. Santachiara, for the feedback that stimulated more thinking which I tried to reflect here. Finally, I am thankful to J. L. Jacobsen, S. Ribault and H. Saleur for comments on the manuscript.

# A  Conformal data at generic $c$

In this appendix, we briefly summarize the known analytic amplitudes $A_\Phi$ of Kac operators at generic $c$, which we have used in the main text for analyzing the normalizations of the operators, extracting the three-point couplings and further taking the $c \to 0$ limit. These amplitudes are obtained from analytic bootstrap equations that utilize the existence of Virasoro degeneracy of $\Phi_{2,1}$ or $\Phi_{1,2}$ in the cluster/dense loop or dilute loop CFTs. The analytic bootstrap equations were first studied for Liouville theory and generalized minimal models [22,23] that contain only diagonal fields. It was further generalized to the case of non-diagonal operators [14,24] and in particular for the cluster model [15]. It is worth stressing that in Liouville theory (including the case with $c < 1$), the analytic bootstrap equations use the degeneracy of both $\Phi_{2,1}$ and $\Phi_{1,2}$ and this has led to a full analytic solution of the CFT. In cluster or loop models, one has only one of these degeneracies which, for the moment, is not sufficient for ob-

taining analytic solutions, although recent numerical bootstrap work has shown such analytic expressions exist [18]. Since we are focused on the Kac operators in this paper, here we mainly summarize the analytic bootstrap results involving diagonal fields, plus an additional case of non-diagonal fields used in the next appendix B. For generic detailed derivations, see [15–17].

## A.1 Cluster/dense loop model

The key in the analytic bootstrap is the insertion of degenerate Kac operators $\Phi_{2,1}$ or $\Phi_{1,2}$ in correlation functions. In the cluster/dense loop models, the field $\Phi_{2,1}$ is degenerate and a four-point function $\langle \hat{\Phi}_{2,1} \hat{\Phi}_{r_2,s_2} \hat{\Phi}_{r_3,s_3} \hat{\Phi}_{r_4,s_4} \rangle$ with generic diagonal fields $\hat{\Phi}_{r_i,s_i}$ (the indices can be fractional, so they are not necessarily Kac operators) has its conformal block expansions truncated in the $s$- and $t$-channel:

$$\langle \hat{\Phi}_{2,1} \hat{\Phi}_{r_2,s_2} \hat{\Phi}_{r_3,s_3} \hat{\Phi}_{r_4,s_4} \rangle = A_+^{(s)} |\mathcal{F}_+^{(s)}|^2 + A_-^{(s)} |\mathcal{F}_-^{(s)}|^2 = A_+^{(t)} |\mathcal{F}_+^{(t)}|^2 + A_-^{(t)} |\mathcal{F}_-^{(t)}|^2 , \tag{A.1}$$

where $\pm$ represents $(r_i \pm 1, s_i)$ with $i = 2$ for $s$−channel and $i = 4$ for $t$−channel. Thanks to the Virasoro degeneracy, the $s$- and $t$-channel chiral conformal blocks are related through the fusing matrix:

$$\begin{bmatrix} \mathcal{F}_+ \\ \mathcal{F}_- \end{bmatrix}^{(s)} = \begin{bmatrix} F_{++} & F_{+-} \\ F_{-+} & F_{--} \end{bmatrix} \begin{bmatrix} \mathcal{F}_+ \\ \mathcal{F}_- \end{bmatrix}^{(t)} , \tag{A.2}$$

and similarly for the anti-chiral $\bar{\mathcal{F}}_\pm$ with $F_{\pm\pm} = \bar{F}_{\pm\pm}$ given by

$$F_{\mathsf{st}} = \frac{\Gamma\left(1 + \frac{2\mathsf{s}}{\beta} P_{r_2,s_2}\right) \Gamma\left(-\frac{2\mathsf{t}}{\beta} P_{r_4,s_4}\right)}{\prod_{+,-} \Gamma\left(\frac{1}{2} \pm \frac{1}{\beta} P_{r_3,s_3} + \frac{\mathsf{s}}{\beta} P_{r_2,s_2} - \frac{\mathsf{t}}{\beta} P_{r_4,s_4}\right)} , \quad \text{with} \quad \mathsf{s}, \mathsf{t} = \pm . \tag{A.3}$$

The $P$ here refers to the Liouville momentum:

$$P_{r_i,s_i} = \frac{r_i}{2\beta} - \frac{s_i \beta}{2} . \tag{A.4a}$$

The equation (A.1) then allows to relate the amplitudes by the relation (A.2). Plugging (A.2) into (A.1), one obtains the relation

$$\rho(P_{r_2,s_2}, P_{r_3,s_3}, P_{r_4,s_4}) = \frac{A_+^{(s)}}{A_-^{(s)}} = -\frac{F_{-+} \bar{F}_{--}}{F_{++} \bar{F}_{+-}} , \tag{A.5}$$

which corresponds to an analytic expression for the conformal data:

$$\rho(P_{r_2,s_2}, P_{r_3,s_3}, P_{r_4,s_4}) = \frac{A_+^{(s)}}{A_-^{(s)}} = \frac{C_{\hat{\Phi}_{2,1} \hat{\Phi}_{r_2,s_2} \hat{\Phi}_{r_2+1,s_2}} C_{\hat{\Phi}_{r_2+1,s_2} \hat{\Phi}_{r_3,s_3} \hat{\Phi}_{r_4,s_4}} B_{\hat{\Phi}_{r_2-1,s_2}}}{C_{\hat{\Phi}_{2,1} \hat{\Phi}_{r_2,s_2} \hat{\Phi}_{r_2-1,s_2}} C_{\hat{\Phi}_{r_2-1,s_2} \hat{\Phi}_{r_3,s_3} \hat{\Phi}_{r_4,s_4}} B_{\hat{\Phi}_{r_2+1,s_2}}} . \tag{A.6}$$

One can consider the case involving non-diagonal fields very similarly. Take the particular example of four-point function $\langle \hat{\Phi}_{2,1} \hat{\Phi}_{0,2} \hat{\Phi}_{0,2} \hat{\Phi}_{2,1} \rangle$. The spinless field $\Phi_{0,2}$ here is non-diagonal because by fusing with $\Phi_{2,1}$ it generates non-diagonal fields of $\hat{\Phi}_{-1,2}, \hat{\Phi}_{1,2}$ (which we call $\hat{X}, \hat{\bar{X}}$ in the main text) with dimensions $(h_{-1,2}, h_{1,2})$ and $(h_{1,2}, h_{-1,2})$. The four-point function has the conformal block expansion:

$$\langle \hat{\Phi}_{2,1} \hat{\Phi}_{0,2} \hat{\Phi}_{0,2} \hat{\Phi}_{2,1} \rangle = A_-^{(s)} \mathcal{F}_-^{(s)} \bar{\mathcal{F}}_+^{(s)} + A_+^{(s)} \mathcal{F}_+^{(s)} \bar{\mathcal{F}}_-^{(s)} = A_+^{(t)} |\mathcal{F}_+^{(t)}|^2 + A_-^{(t)} |\mathcal{F}_-^{(t)}|^2 , \tag{A.7}$$

where again, the conformal blocks are related through (A.2) and thus the amplitudes are related. In this case, we have the amplitudes given by the following conformal data:

$$A_+^{(s)} = \frac{C_{\hat{\Phi}_{2,1} \hat{\Phi}_{0,2} \hat{X}}^2}{B_{\hat{X}}} , \qquad A_-^{(t)} = C_{\hat{\Phi}_{0,2} \hat{\Phi}_{0,2} \mathbb{I}} C_{\hat{\Phi}_{2,1} \hat{\Phi}_{2,1} \mathbb{I}} = B_{\hat{\Phi}_{0,2}} B_{\hat{\Phi}_{2,1}} . \tag{A.8}$$

We can then define the ratio:

$$\chi = \frac{A_-^{(t)}}{A_+^{(s)}}, \tag{A.9}$$

with

$$\chi^{-1} = \frac{C^2_{\hat{\Phi}_{2,1}\hat{\Phi}_{0,2}\hat{X}}}{B_{\hat{X}} B_{\hat{\Phi}_{0,2}} B_{\hat{\Phi}_{2,1}}}. \tag{A.10}$$

The expression of $\chi$ through the fusing matrices can be found in section 4.1.1 of [15] (see eq. (151) there, in particular $\chi_{+-}^{ND}$). We use this expression below in (B.18).

Going back to the diagonal case, the expressions (A.5) and (A.6) allow to obtain a generic recursion relation of the amplitudes for $\hat{\Phi}_{r+1,s}$ and $\hat{\Phi}_{r-1,s}$ in generic four-point functions (here taken to be of identical diagonal fields). Consider the amplitudes for $\hat{\Phi}_{r+1,s}$ and $\hat{\Phi}_{r-1,s}$ in a four-point function of diagonal field $\mathcal{O}$:

$$R_{r,s}^{\mathcal{O},\text{cluster/dense}} = \frac{A^{\mathcal{O}}_{\hat{\Phi}_{r+1,s}}}{A^{\mathcal{O}}_{\hat{\Phi}_{r-1,s}}} = \frac{C^2_{\mathcal{O}\mathcal{O}\hat{\Phi}_{r+1,s}} B_{\hat{\Phi}_{r-1,s}}}{C^2_{\mathcal{O}\mathcal{O}\hat{\Phi}_{r-1,s}} B_{\hat{\Phi}_{r+1,s}}}. \tag{A.11}$$

To get such a recursion, one first takes the four-point function

$$\langle \Phi_{2,1} \Phi_{r,s} \mathcal{O}\mathcal{O} \rangle, \tag{A.12}$$

with $\rho$ given by

$$\rho(P_{r,s}, P_{\mathcal{O}}, P_{\mathcal{O}}) = \frac{C_{\Phi_{2,1}\Phi_{r,s}\hat{\Phi}_{r+1,s}} C_{\mathcal{O}\mathcal{O}\hat{\Phi}_{r+1,s}} B_{\hat{\Phi}_{r-1,s}}}{C_{\Phi_{2,1}\Phi_{r,s}\hat{\Phi}_{r-1,s}} C_{\Phi\mathcal{O}\hat{\Phi}_{r-1,s}} B_{\hat{\Phi}_{r+1,s}}}, \tag{A.13}$$

and further takes the four-point function

$$\langle \Phi_{2,1} \Phi_{r,s} \Phi_{r,s} \Phi_{2,1} \rangle, \tag{A.14}$$

with $\rho$ given by

$$\rho(P_{r,s}, P_{r,s}, P_{2,1}) = \frac{C^2_{\Phi_{2,1}\Phi_{r,s}\hat{\Phi}_{r+1,s}} B_{\hat{\Phi}_{r-1,s}}}{C^2_{\Phi_{2,1}\Phi_{r,s}\hat{\Phi}_{r-1,s}} B_{\hat{\Phi}_{r+1,s}}}. \tag{A.15}$$

It is then straightforward to see that $R_{r,s}^{\mathcal{O}}$ in (A.11) is given by

$$R_{r,s}^{\mathcal{O},\text{cluster/dense}} = \frac{\rho^2(P_{r,s}, P_{\mathcal{O}}, P_{\mathcal{O}})}{\rho(P_{r,s}, P_{r,s}, P_{2,1})}. \tag{A.16}$$

In section 3, for considering Kac operator in the dense loop model four-spin correlator, we have used the expression (A.16) with

$$\mathcal{O} \to \Phi_{0,\frac{1}{2}}, \qquad \hat{\Phi}_{r,s} \to \hat{\Phi}_{r,1}, \tag{A.17}$$

which gives rise to eq. (99). In cluster model, the amplitude recursion in the four-spin correlator is even stronger due to the special property that

$$\Phi_{2,1} \times \Phi_{\frac{1}{2},0} \to \Phi_{-\frac{1}{2},0} \sim \Phi_{\frac{1}{2},0}. \tag{A.18}$$

The detailed derivation follows the same idea as above but slightly more involved (see [15] section 4.1.1 for details) and one arrives at:

$$R_{r,s}^{\sigma,\text{cluster}} = \frac{A^{\sigma,\text{cluster}}_{\Phi_{r+1,s}}}{A^{\sigma,\text{cluster}}_{\Phi_{r,s}}} = \frac{2^{4s - \frac{4r+2}{\beta^2}} \Gamma\left(\frac{s}{2} - \frac{r}{2\beta^2}\right) \Gamma\left(\frac{r}{2\beta^2} + \frac{1-s}{2}\right) \Gamma\left(\frac{s+2}{2} - \frac{r+1}{2\beta^2}\right) \Gamma\left(\frac{r+1}{2\beta^2} + \frac{1-s}{2}\right)}{\Gamma\left(\frac{s+1}{2} - \frac{r}{2\beta^2}\right) \Gamma\left(\frac{r}{2\beta^2} + \frac{2-s}{2}\right) \Gamma\left(\frac{s+1}{2} - \frac{r+1}{2\beta^2}\right) \Gamma\left(\frac{r+1}{2\beta^2} - \frac{s}{2}\right)}, \tag{A.19}$$

which we use in section 3.2 eq. (85) for the case $(r,s) = (i,1)$.

The expression (A.16) can also be used for extracting three-point constants involving all Kac operators. In sections 3.2.2 and 5.1, we have considered the three-point constant

$$C_{\hat{\Phi}_{2,1}\hat{\Phi}_{2,1}\hat{\Phi}_{3,1}}\,, \tag{A.20}$$

to examine the decoupling of the $\hat{\Phi}_{3,1}$ at Ising point and the logarithmic mixing at $c = 0$. For this, we take

$$\mathcal{O} \to \hat{\Phi}_{2,1}\,, \qquad \hat{\Phi}_{r,s} \to \hat{\Phi}_{2,1}\,, \tag{A.21}$$

in (A.16) and obtain

$$R_{2,1}^{\hat{\Phi}_{2,1}} = \frac{A_{\hat{\Phi}_{3,1}}^{\hat{\Phi}_{2,1}}}{A_{\hat{\Phi}_{1,1}}^{\hat{\Phi}_{2,1}}} = \frac{C_{\hat{\Phi}_{2,1}\hat{\Phi}_{2,1}\hat{\Phi}_{3,1}}^2}{B_{\hat{\Phi}_{2,1}}^2 B_{\hat{\Phi}_{3,1}}} = -\frac{\Gamma\left(2 - \frac{2}{\beta^2}\right)^2 \Gamma\left(\frac{3}{\beta^2} - 1\right)\Gamma\left(\frac{1}{\beta^2}\right)}{\Gamma\left(2 - \frac{3}{\beta^2}\right)\Gamma\left(\frac{2}{\beta^2}\right)^2 \Gamma\left(\frac{\beta^2 - 1}{\beta^2}\right)}\,. \tag{A.22}$$

In section 5.2, we also used the three-point constant of the second energy operator $\varepsilon' \sim \hat{\Phi}_{3,1}$:

$$C_{\hat{\Phi}_{3,1}\hat{\Phi}_{3,1}\hat{\Phi}_{3,1}}\,. \tag{A.23}$$

This can be extracted by taking

$$\mathcal{O} \to \hat{\Phi}_{3,1}\,, \qquad \hat{\Phi}_{r,s} \to \hat{\Phi}_{2,1}\,, \tag{A.24}$$

to obtain

$$R_{2,1}^{\hat{\Phi}_{3,1}} = \frac{A^{\hat{\Phi}_{3,1}}(\hat{\Phi}_{3,1})}{A^{\hat{\Phi}_{3,1}}(\hat{\Phi}_{1,1})} = \frac{C_{\hat{\Phi}_{3,1}\hat{\Phi}_{3,1}\hat{\Phi}_{3,1}}^2}{B_{\hat{\Phi}_{3,1}}^3} = -\frac{\Gamma\left(2 - \frac{3}{\beta^2}\right)\Gamma\left(1 - \frac{2}{\beta^2}\right)^4 \Gamma\left(\frac{4}{\beta^2} - 1\right)^2 \Gamma\left(\frac{1}{\beta^2}\right)^3}{\Gamma\left(2 - \frac{4}{\beta^2}\right)^2 \Gamma\left(1 - \frac{1}{\beta^2}\right)^3 \Gamma\left(\frac{2}{\beta^2} - 1\right)^2 \Gamma\left(\frac{3}{\beta^2} - 1\right)\Gamma\left(\frac{2}{\beta^2}\right)^2}\,. \tag{A.25}$$

## A.2  Dilute loop model

The case of dilute loop model is analogous, where the degeneracy comes from $\Phi_{1,2}$, and one can derive the same type of recursion as above, but with the $s$ indices shifted:

$$R_{r,s}^{\mathcal{O},\text{dilute}} = \frac{A_{\hat{\Phi}_{r,s+1}}^{\mathcal{O}}}{A_{\hat{\Phi}_{r,s-1}}^{\mathcal{O}}}\,. \tag{A.26}$$

We summarize the results for $\mathcal{O}$ being the dilute loop spin operator:

$$R_{r,s}^{\sigma,\text{dilute}} = \frac{A^{\sigma,\text{dilute}}(\hat{\Phi}_{r,s+1})}{A^{\sigma,\text{dilute}}(\hat{\Phi}_{r,s-1})} = -\frac{2^{8(r-\beta^2 s)}\Gamma\left((1-s)\beta^2 + r\right)\Gamma\left(s\beta^2 - r\right)^2 \Gamma\left(-(s+1)\beta^2 + r + 1\right)}{\Gamma\left((s-1)\beta^2 - r + 1\right)\Gamma\left(r - s\beta^2\right)^2 \Gamma\left((s+1)\beta^2 - r\right)}\,, \tag{A.27}$$

which we have used the $r = 1$ expressions in (100).

In section 5.1, we also considered the three-point constants of energy operator $\hat{\Phi}_{1,3}$ in the dilute loop model. This can be extracted from:

$$R_{1,2}^{\hat{\Phi}_{1,3}} = \frac{A_{\hat{\Phi}_{1,3}}^{\hat{\Phi}_{1,3}}}{A_{\hat{\Phi}_{1,1}}^{\hat{\Phi}_{1,3}}} = -\frac{\Gamma\left(\beta^2\right)^3 \Gamma\left(2 - 3\beta^2\right)\Gamma\left(1 - 2\beta^2\right)^4 \Gamma\left(4\beta^2 - 1\right)^2}{\Gamma\left(2\beta^2\right)^2 \Gamma\left(2 - 4\beta^2\right)^2 \Gamma\left(1 - \beta^2\right)^3 \Gamma\left(2\beta^2 - 1\right)^2 \Gamma\left(3\beta^2 - 1\right)}\,, \tag{A.28}$$

and furthermore, the three-point constants $\hat{\Phi}_{1,3}$ with $\hat{\Phi}_{1,5}$ can be extracted from

$$\frac{C_{\hat{\Phi}_{1,3}\hat{\Phi}_{1,3}\hat{\Phi}_{1,5}}^2}{B_{\hat{\Phi}_{1,3}}^2 B_{\hat{\Phi}_{1,5}}} = R_{1,2}^{\hat{\Phi}_{1,3}} R_{1,4}^{\hat{\Phi}_{1,3}} = \frac{\left(1 - 2\beta^2\right)^2 \Gamma\left(\beta^2\right)\Gamma\left(1 - 3\beta^2\right)\Gamma\left(2 - 3\beta^2\right)\Gamma\left(5\beta^2 - 1\right)}{\left(1 - 4\beta^2\right)^2 \Gamma\left(3\beta^2\right)\Gamma\left(2 - 5\beta^2\right)\Gamma\left(1 - \beta^2\right)\Gamma\left(3\beta^2 - 1\right)}\,. \tag{A.29}$$

# B Singularity cancellation conditions

In a previous work [21], constructions of the rank-2 and rank-3 Jordan blocks were done by examining the four-point function of spin operator, in particular in the percolation context. There, various cancellation conditions that are required for the logarithmic two-point functions were checked to hold (see appendix B of [21]). In section 2.2, we revisited the rank-2 and rank-3 Jordan block constructions with a refreshed point of view: we use the properly normalized operators such as $\hat{X}, \hat{\Phi}_{3,1}$ etc. and take the $c \to 0$ limit to obtain the logarithmic two- and three-point functions at $c = 0$. These logarithmic conformal data at $c = 0$ are then further assembled into the four-point function according to cluster decomposition. In this context, given a definition of the top field in the Jordan blocks – for example, $t$ or $\Psi_2$, certain singularity cancellation conditions have to hold for the logarithms in the correlation functions to appear. These cancellation conditions represent special properties of the operators in order to build logarithmic type of correlations. In this appendix, we clarify and check some of these singularity cancellations using known conformal data at generic $c$.

## B.1 Section 2.2

Consider first the field $\hat{X}$ involved in the construction of $t$. Its generic $c$ two-point function is given in (30). The three-point function with the spin operator is given in (33):

$$\langle \sigma(z_1, \bar{z}_1) \sigma(z_2, \bar{z}_2) \hat{X}(z_3, \bar{z}_3) \rangle^{\text{cluster}} = C_{\sigma\sigma\hat{X}}^{\text{cluster}}(c) \mathbb{P}_3^c, \tag{B.1}$$

and we need the leading behavior of $C_{\sigma\sigma\hat{X}}^{\text{cluster}}(c)$ from (35):

$$C_{\sigma\sigma\hat{X}}^{\text{cluster}}(c) = h_\sigma^{\text{perco}} + O(c). \tag{B.2}$$

This behavior has been checked previously with numerical bootstrap results of [15]. It is worth noting that to check such relation, we have used the amplitude of $\hat{X}$ in the four-point function of the spin operator. To extract the 3-point constant $C_{\sigma\sigma\hat{X}}$, we still need to make a sign choice. For this, we use the identification of the field $\hat{X}$ and $T$ at $c = 0$ to select the sign. Namely, in the definition of the logarithmic operator $t$, the operator $\hat{X}$ is the one with the correct sign such that its three-point function with $\sigma$ near $c = 0$ has the same sign as $C_{\sigma\sigma T}$. The same idea applies for the following cases.

Proceeding to the construction of the field $\Psi_2$, we start with the generic $c$ logarithmic pair $(\Psi, \bar{A}X)$ whose two-point functions are given in (17):

$$\langle \hat{\Psi}(z, \bar{z}) \hat{\Psi}(0, 0) \rangle = \left( -2\gamma_{\hat{\Psi}}(c) \ln(z\bar{z}) + \omega(c) \right) \mathbb{P}_2^c, \tag{B.3a}$$

$$\langle \hat{\Psi}(z, \bar{z}) \bar{A}\hat{X}(0, 0) \rangle = \gamma_{\hat{\Psi}}(c) \mathbb{P}_2^c, \tag{B.3b}$$

$$\langle \bar{A}\hat{X}(z, \bar{z}) \bar{A}\hat{X}(0, 0) \rangle = 0, \tag{B.3c}$$

with

$$\omega(c) = B_{\hat{X}}(c) \lambda(c), \qquad \gamma_{\hat{\Psi}}(c) = B_{\hat{X}}(c) b_{1,2}(c). \tag{B.4}$$

In previous work [20], the two-point functions (B.3) were computed in a given operator basis and fixed $\lambda(c)$. In this basis, the three-point functions (53) are given by:

$$\langle \sigma(z_1, \bar{z}_1) \sigma(z_2, \bar{z}_2) \bar{A}\hat{X}(z_3, \bar{z}_3) \rangle = C_{\sigma\sigma\bar{A}\hat{X}}^{\text{cluster}}(c) \mathbb{P}_3^c, \tag{B.5a}$$

$$\langle \sigma(z_1, \bar{z}_1) \sigma(z_2, \bar{z}_2) \hat{\Psi}(z_3, \bar{z}_3) \rangle = \left( C_{\sigma\sigma\hat{\Psi}}^{\text{cluster}}(c) + C_{\sigma\sigma\bar{A}\hat{X}}^{\text{cluster}}(c) \tau_3 \right) \mathbb{P}_3^c, \tag{B.5b}$$

with

$$b_{1,2}(c) C_{\sigma\sigma\hat{\Psi}}(c) = \lambda(c) C_{\sigma\sigma\bar{A}\hat{X}}(c). \tag{B.6}$$

Note that the condition (B.6) arise from fixing the basis in the Jordan block $(\Psi, \bar{A}X)$, as explained below in (D.44), and this gives rise to the OPE term $\Psi + \ln(z\bar{z})\bar{A}X$ in (24). Correspondingly, their contribution in the four-point function of spin operator is given by

$$\langle\sigma\sigma\sigma\sigma\rangle^{\text{cluster}} = (z\bar{z})^{-2h_\sigma^{\text{cluster}}(c)+h_{-1,2}(c)}\left\{\frac{C^2_{\sigma\sigma\bar{A}\hat{X}}(c)}{\gamma^2_{\hat\Psi}(c)}\left(\omega(c)+\gamma_{\hat\Psi}(c)\ln(z\bar{z})\right)\right\}+\dots \tag{B.7}$$

The three-point constants in (B.5a) are given by

$$C_{\sigma\sigma\bar{A}\hat{X}}(c) = C_{\sigma\sigma\hat{X}}(c)f(c), \tag{B.8}$$

with

$$f(c) = \frac{1-\beta^4}{16\beta^2}, \tag{B.9}$$

as determined from the Virasoro symmetry. It is now straightforward to use (B.8) to check the singularity cancellation condition (55) to be true:

$$C^{\text{cluster}}_{\sigma\sigma\bar{A}\hat{X}}(0) = \left(h_\sigma^{\text{perco}}\right)^2. \tag{B.10}$$

Next, consider the operator $\hat{\Phi}_{3,1}$ in the rank-3 Jordan block construction. At generic $c$, using (A.19), one has the expression:

$$\frac{C^2_{\sigma\sigma\hat{\Phi}_{3,1}}(c)}{B_{\hat{\Phi}_{3,1}}(c)} = \frac{2^{\frac{7(\beta^2-2)}{\beta^2}}\Gamma\left(\frac{1}{2}-\frac{1}{\beta^2}\right)\Gamma\left(\frac{3}{2}-\frac{1}{\beta^2}\right)\Gamma\left(\frac{1}{2\beta^2}\right)^2\Gamma\left(\frac{1}{\beta^2}\right)\Gamma\left(\frac{3}{2\beta^2}\right)\Gamma\left(\frac{3(\beta^2-1)}{2\beta^2}\right)}{\Gamma\left(1-\frac{3}{2\beta^2}\right)\Gamma\left(1-\frac{1}{2\beta^2}\right)^2\Gamma\left(\frac{1}{\beta^2}-\frac{1}{2}\right)\Gamma\left(\frac{1}{2}+\frac{1}{\beta^2}\right)\Gamma\left(-\frac{\beta^2-3}{2\beta^2}\right)\Gamma\left(\frac{\beta^2-1}{\beta^2}\right)}. \tag{B.11}$$

Recall the $c \to 0$ expansion of the normalization of $\hat{\Phi}_{3,1}$ in (49). The leading term $-c^2/4$ guarantees the cancellation of singularity at $O(c^{-2})$ in the two-point function of $\Psi_2$. Using (B.11), one can check the second singularity cancellation condition in (55) to be true:

$$C^{\text{cluster}}_{\sigma\sigma\hat{\Phi}_{3,1}}(0) = \left(h_\sigma^{\text{perco}}\right)^2. \tag{B.12}$$

The cancellation of the singularity at $O(c^{-1})$ in the two-point function of $\Psi_2$ requires

$$4b^2_{1,2}B^{(3)}_{\hat{\Phi}_{3,1}} + \omega' = 0, \tag{B.13}$$

and this fixes further the expansion coefficient $B^{(3)}_{\hat{\Phi}_{3,1}}$ in the normalization of $\hat{\Phi}_{3,1}$ (49). With this, we can finally check the condition (56) to be true

$$C'_{\sigma\sigma\hat{\Phi}_{3,1}}(0) = C_{\sigma\sigma\hat\Psi}(0)h'_{3,1} + 2h_\sigma^{\text{perco}}\left(h_\sigma^{\text{perco}}\right)', \tag{B.14}$$

where we have used the value of $\lambda$ at $c = 0$ (see footnote 14 in [21]).

In section 4.3, we have considered a slightly different construction of the rank-3 Jordan block. Using the top field construction (154) on the field $\Psi_2$ leads to the definition (177) which, as commented in footnote 17, is related to the previous case by a change of basis in the rank-3 Jordan block. Computing its two-point leads to a slightly different singularity cancellation condition from (B.13):

$$4b^2_{1,2}B^{(3)}_{\hat{\Phi}_{3,1}} - \omega' = 0, \tag{B.15}$$

which chooses a different normalization expansion coefficient $B^{(3)}_{\hat{\Phi}_{3,1}}$. Proceeding to the three-point function with the spin operator $\sigma$, one finds

$$C^{\text{cluster}}_{\sigma\sigma\bar{A}\hat{X}}(0) = \left(h^{\text{perco}}_\sigma\right)^2, \qquad C^{\text{cluster}}_{\sigma\sigma\hat{\Phi}_{3,1}}(0) = \left(h^{\text{perco}}_\sigma\right)^2, \tag{B.16}$$

as well as

$$4B^{(3)}_{\hat{\Phi}_{3,1}}\left(h^{\text{perco}}_\sigma\right)^2 + C'_{\sigma\sigma\hat{\Phi}_{3,1}}(0) - C_{\sigma\sigma\hat{\Psi}}(0)h'_{3,1} - 2h^{\text{perco}}_\sigma\left(h^{\text{perco}}_\sigma\right)' = 0, \tag{B.17}$$

which can be checked to be true.

## B.2 Section 5.1

In section 5.1, we computed the three-point functions of the energy logarithmic pair $(\tilde{\varepsilon}, \varepsilon)$ with the rank-2 and rank-3 Jordan blocks. Here we check several cancellation conditions involved there.

First, for getting (186), we need (187):

$$C_{\hat{\Phi}_{0,2}\hat{\Phi}_{2,1}\hat{X}}(c) = -\frac{h_\varepsilon}{2}c + o(c). \tag{B.18}$$

As explained in appendix A.1, this conformal data can be extracted from (A.10) by taking the normalizations

$$B_{\hat{\Phi}_{0,2}} \simeq -\frac{c}{2}, \qquad B_{\hat{\Phi}_{2,1}} \simeq \frac{c}{2}, \qquad B_{\hat{X}} \simeq -\frac{c}{2}, \tag{B.19}$$

and it is straightforward to see that (B.18) holds. Moreover, for arriving at (189), we encounter the singularity cancellation conditions

$$C_{\hat{\Phi}_{0,2}\hat{\Phi}_{2,1}\hat{\Psi}}(c) = C'_{\hat{\Phi}_{0,2}\hat{\Phi}_{2,1}\hat{\Psi}}c + o(c), \tag{B.20a}$$

$$C_{\hat{\Phi}_{0,2}\hat{\Phi}_{2,1}A\hat{X}}(c) = -\frac{h_\varepsilon^2}{2}c + o(c), \tag{B.20b}$$

with

$$C'_{\hat{\Phi}_{0,2}\hat{\Phi}_{2,1}\hat{\Psi}} = 2\gamma'_\Psi\left(2h_\varepsilon^2 B''_{\hat{\Phi}_{2,1}} - 4h_\varepsilon^2 B^{(3)}_{\hat{\Phi}_{3,1}} + 2h_\varepsilon h'_{2,1} - C''_{\hat{\Phi}_{2,1}\hat{\Phi}_{2,1}\hat{\Phi}_{3,1}}\right). \tag{B.21}$$

At generic $c$, from Virasoro symmetry, we have the relation:

$$C_{\hat{\Phi}_{2,1}\hat{\Phi}_{0,2}\bar{A}\hat{X}}(c) = C_{\hat{\Phi}_{2,1}\hat{\Phi}_{0,2}\hat{X}}(c)\tilde{f}(c), \tag{B.22a}$$

with

$$\tilde{f}(c) = -\frac{1}{\beta^6} + \frac{2}{\beta^4} + \frac{1}{\beta^2} - 2. \tag{B.23}$$

This allows to confirm (B.20b). The condition (B.21) amounts to further fix the expansion coefficients in the normalization $B_{\hat{\Phi}_{2,1}}$.

## C Two rank-2 Jordan blocks at $O(c)$

In section 2.2.2, we presented a construction of the rank-3 Jordan blocks in an intuitive way. The observation is that a rank-3 Jordan block can be constructed by the mixing of two rank-2 Jordan blocks with opposite two-point functions. At generic $c$, we already have a rank-2 Jordan block $(\hat{\Psi}, \bar{A}\hat{X})$ whose two-point functions vanish at $c = 0$ as $O(c)$. The other rank-2 Jordan block comes from mixing the fields $T\bar{T}$ and $\varepsilon' \sim \hat{\Phi}_{3,1}$ into an "intermediate" top field $\Theta$, where either $T\bar{T}$ or $\hat{\Phi}_{3,1}$ can be taken as the bottom field. In this appendix, we explain in

more details about this intermediate rank-2 Jordan block at $O(c)$ and how this further gives rise to the middle field $\Psi_1$ in the rank-3 Jordan block $(\Psi_2, \Psi_1, \Psi_0)$.

Take the definition from (43)

$$\Theta = \frac{2b_{1,2}}{c}\big(T\bar{T} - \hat{\Phi}_{3,1}\big). \tag{C.1}$$

We can compute the two-point function

$$
\begin{aligned}
\langle\Theta(z,\bar{z})\Theta(0,0)\rangle &= \Big(2b_{1,2}^2 h_{1,3}' c \ln(z\bar{z}) + 4b_{1,2}^2 B_{3,1}^{(3)} c\Big)\mathbb{P}_2^0 \\
&= \Big(-b_{1,2} c \ln(z\bar{z}) + 4b_{1,2}^2 B_{3,1}^{(3)} c\Big)\mathbb{P}_2^0,
\end{aligned}
\tag{C.2}
$$

and we also have the two-point function (42). Note that through this mixing, the double zero $O(c^2)$ in the norm of $T\bar{T}$ (and $\hat{\Phi}_{3,1}$) is reduced to a single zero $O(c)$ in the logarithmic coupling of the rank-2 Jordan block. The two-point function (42) means that the logarithmic mixing allows a two-point correlation for $T\bar{T}$ or the second energy operator $\varepsilon' \sim \hat{\Phi}_{3,1}$ with the top field $\Theta$ that vanishes less fast than their original two-point functions as $c \to 0$ but still does not survive at the $c = 0$ point. On the other hand, this is also not enough the cure the "$c \to 0$ catastrophe", as the OPE of the spin operator is still divergent (although less than before). This can be seen below in (D.46) where a rank-2 Jordan block appear in the OPE of a non-logarithmic operator. To further cancel the $O(c)$ singularity in the OPE, the rank-2 Jordan block $(\Theta, \varepsilon')$ need to further mix with the other rank-2 Jordan block $(\hat{\Psi}, \bar{A}\hat{X})$ at $O(c)$. Note that this requires the two-point function of the two rank-2 Jordan blocks to have opposite constants. The logarithmic coupling $-b_{1,2}c$ is already so by definition. We further see that the non-logarithmic constant term in (C.2) should be

$$4b_{1,2}^2 B_{3,1}^{(3)} = -\omega', \tag{C.3}$$

where $\omega$ is given in (B.3). This is precisely the singularity cancellation condition (B.13) we encountered before. This mechanism of step-by-step cancellation gives some intuition about how a higher rank Jordan block $(\Psi_2, \Psi_1, \Psi_0)$ appears in the mixing.

With this, we can construct the top field $\Psi_2$ as done in the main text eq. (47). To arrive at the middle field $\Psi_1$, we apply $L_0, \bar{L}_0$ using

$$
\begin{aligned}
(L_0 - 2)\Theta &= \frac{2b_{1,2}}{c}(-h_{3,1}(c) - 2)\hat{\Phi}_{3,1} \\
&\overset{c\to 0}{=} \frac{2b_{1,2}}{c}\bigg(-h_{3,1}' c - \frac{h_{3,1}''}{2}c^2 + \ldots\bigg)\hat{\Phi}_{3,1} \\
&= \hat{\Phi}_{3,1} - b_{1,2} h_{3,1}'' c \hat{\Phi}_{3,1} + \ldots,
\end{aligned}
\tag{C.4}
$$

as well as

$$
\begin{aligned}
(L_0 - 2)\hat{\Psi} &= (h_{1,-2}(c) - 2)\hat{\Psi} + \bar{A}\hat{X} \\
&\overset{c\to 0}{=} h_{1,-2}' c \hat{\Psi} + \bar{A}\hat{X} + \ldots \\
&= -\frac{c}{2b}\hat{\Psi} + \bar{A}\hat{X} + \ldots,
\end{aligned}
\tag{C.5}
$$

and similarly for $\bar{L}_0$. Note that while naively the rank-2 Jordan blocks at $O(c)$ give $(L_0 - 2)\Theta = \hat{\Phi}_{3,1}$ and $(L_0 - 2)\hat{\Psi} = \bar{A}\hat{X}$, here we have to also keep the fields with coefficient $O(c)$ for writing the middle field $\Psi_1$. These terms are important to guarantee that the two-point functions involving $\Psi_2, \Psi_1$ satisfy conformal invariance. The result is the expression of $\Psi_1$ in (52).

# D  Logarithmic OPE and correlation functions

In this appendix, we briefly summarize the forms of correlation functions involving logarithmic operators as a result of the conformal Ward identities. There are various references discussing this in more details; see e.g. [51, 55, 56]. Here, our summary focuses on the quantities that we study in this paper.

The basic things are the conformal Ward identities

$$[L_n, \mathcal{O}_i(z, \bar{z})] = z^{n+1} \partial \mathcal{O}_i + (n+1) D_{ij} z^n \mathcal{O}_j \,, \tag{D.1a}$$

$$[\bar{L}_n, \mathcal{O}_i(z, \bar{z})] = \bar{z}^{n+1} \bar{\partial} \mathcal{O}_i + (n+1) \bar{D}_{ij} \bar{z}^n \mathcal{O}_j \,, \tag{D.1b}$$

where $D_{ij}, \bar{D}_{ij}$ take Jordan block forms. For example, for a rank-2 Jordan block $(\psi_1, \psi_0)$ with conformal dimension $(h, \bar{h})$, we have

$$D = \begin{bmatrix} h & 1 \\ 0 & h \end{bmatrix}, \qquad \bar{D} = \begin{bmatrix} \bar{h} & 1 \\ 0 & \bar{h} \end{bmatrix}, \qquad \mathcal{O} = \begin{bmatrix} \psi_1 \\ \psi_0 \end{bmatrix}, \tag{D.2}$$

and for a rank-3 Jordan block $(\Psi_2, \Psi_1, \Psi_0)$:

$$D = \begin{bmatrix} h & 1 & 0 \\ 0 & h & 1 \\ 0 & 0 & h \end{bmatrix}, \qquad \bar{D} = \begin{bmatrix} \bar{h} & 1 & 0 \\ 0 & \bar{h} & 1 \\ 0 & 0 & \bar{h} \end{bmatrix}, \qquad \mathcal{O} = \begin{bmatrix} \Psi_2 \\ \Psi_1 \\ \Psi_0 \end{bmatrix}. \tag{D.3}$$

Note that the Jordan blocks are subject to a change of basis:

$$\mathcal{O}_i \to \tilde{\mathcal{O}}_i = \mathcal{R}_{ij} \mathcal{O}_j \,, \tag{D.4}$$

which for rank-2 Jordan block, $\mathcal{R}$ takes the form

$$\mathcal{R}^{(2)} = \begin{bmatrix} 1 & r_0 \\ 0 & 1 \end{bmatrix}, \tag{D.5}$$

and for rank-3 Jordan block,

$$\mathcal{R}^{(3)} = \begin{bmatrix} 1 & r_1 & r_0 \\ 0 & 1 & r_1 \\ 0 & 0 & 1 \end{bmatrix}. \tag{D.6}$$

## D.1  Two-point functions

Consider

$$[L_n, \mathcal{O}_i \mathcal{O}_j] = \mathcal{O}_i [L_n, \mathcal{O}_j] + [L_n, \mathcal{O}_i] \mathcal{O}_j, \tag{D.7a}$$

$$[\bar{L}_n, \mathcal{O}_i \mathcal{O}_j] = \mathcal{O}_i [\bar{L}_n, \mathcal{O}_j] + [\bar{L}_n, \mathcal{O}_i] \mathcal{O}_j, \qquad n = -1, 0, 1 \,, \tag{D.7b}$$

and use

$$\langle [L_n, \mathcal{O}_i \mathcal{O}_j] \rangle = 0 \,, \qquad \langle [\bar{L}_n, \mathcal{O}_i \mathcal{O}_j] \rangle = 0 \,, \qquad n = -1, 0, 1 \,, \tag{D.8}$$

namely the vacuum is invariant under global conformal transformation. It is straightforward to obtain the two-point functions of rank-2 Jordan block $(\psi_1, \psi_0)$

$$\langle \psi_1(z, \bar{z}) \psi_1(0, 0) \rangle = \left( -2 b_0 \ln(z \bar{z}) + b_1 \right) \mathbb{P}_2 \,, \tag{D.9a}$$

$$\langle \psi_1(z, \bar{z}) \psi_0(0, 0) \rangle = b_0 \mathbb{P}_2 \,, \tag{D.9b}$$

$$\langle \psi_0(z, \bar{z}) \psi_0(0, 0) \rangle = 0 \,, \tag{D.9c}$$

and for rank-3 Jordan blocks $(\Psi_2, \Psi_1, \Psi_0)$:

$$\langle \Psi_2(z,\bar{z})\Psi_2(0,0)\rangle = \left(a_2 - 2a_1\ln(z\bar{z}) + 2a_0\ln^2(z\bar{z})\right)\mathbb{P}_2, \tag{D.10a}$$

$$\langle \Psi_2(z,\bar{z})\Psi_1(0,0)\rangle = \left(a_1 - 2a_0\ln(z\bar{z})\right)\mathbb{P}_2, \tag{D.10b}$$

$$\langle \Psi_1(z,\bar{z})\Psi_1(0,0)\rangle = a_0\mathbb{P}_2, \tag{D.10c}$$

$$\langle \Psi_2(z,\bar{z})\Psi_0(0,0)\rangle = a_0\mathbb{P}_2, \tag{D.10d}$$

$$\langle \Psi_1(z,\bar{z})\Psi_0(0,0)\rangle = 0, \tag{D.10e}$$

$$\langle \Psi_0(z,\bar{z})\Psi_0(0,0)\rangle = 0, \tag{D.10f}$$

where we have denoted:

$$\mathbb{P}_2 = \frac{1}{z^{2h}\bar{z}^{2\bar{h}}}. \tag{D.11}$$

As we have commented before, the logarithmic couplings $b_0, a_0$ are intrinsic quantities which do not depend on the choice of basis for the logarithmic operators. On the other hand, the parameters $b_1, a_1, a_2$ depend on such basis, or equivalently, the scale at which we look at the two-point functions. This can be understood as following. For the rank-2 Jordan block $(\psi_1, \psi_0)$, under dilatation, the bottom field $\psi_0$ transforms like an usual conformal operator

$$\psi_0(\mu z, \mu\bar{z}) = \mu^{-(h+\bar{h})}\psi_0(z,\bar{z}), \tag{D.12}$$

while the top field $\psi_1$ transforms as

$$\psi_1(\mu z, \mu\bar{z}) = \mu^{-(h+\bar{h})}\left(\psi_1(z,\bar{z}) - \ln\mu^2\psi_0(z,\bar{z})\right). \tag{D.13}$$

Namely, the change in the correlator by a change of scale can be compensated by a change of basis in the Jordan block and the factor $\ln\mu^2$ introduced by the scale change corresponds precisely to the $r_0$ factor in (D.5). In this paper we have set $\mu = 1$. Alternatively, in terms of the constants in the two-point functions (D.9) and (D.10), under a change of basis of the rank-2 and rank-3 Jordan block

$$\tilde{\psi} = \mathcal{R}^{(2)}\psi, \qquad \tilde{\Psi} = \mathcal{R}^{(3)}\Psi, \tag{D.14}$$

where $\mathcal{R}$ are given by (D.5) and (D.6), one finds

$$\tilde{b}_1 = b_1 + 2r_0 b_0, \qquad \tilde{b}_0 = b_0, \tag{D.15}$$

$$\tilde{a}_2 = a_2 + 2r_1 a_1 + (2r_0 + r_1^2)a_0, \qquad \tilde{a}_1 = a_1 + 2r_1 a_0, \qquad \tilde{a}_0 = a_0. \tag{D.16}$$

Here $\tilde{b}_i, \tilde{a}_i$ denote the constants appearing in the two-point functions of $\tilde{\psi}_i, \tilde{\Psi}_i$.

## D.2 Three-point functions

For three-point functions, we focus on two cases that we have studied in this paper:

- Non-logarithmic operator – the spin operator $\sigma$ with a rank-2 Jordan block $(t, T)$ or a rank-3 Jordan block $(\Psi_2, \Psi_1, \Psi_0)$.

- A rank-2 Jordan block $(\tilde{\varepsilon}, \varepsilon)$ with a rank-2 Jordan block $(t, T)$ or a rank-3 Jordan block $(\Psi_2, \Psi_1, \Psi_0)$.

One can then consider the action of $L_n, \bar{L}_n$ on three operators:

$$\left[L_n, \mathcal{O}_i\mathcal{O}_j\mathcal{O}_k\right] = \mathcal{O}_i\left[L_n, \mathcal{O}_j\right]\mathcal{O}_k + \left[L_n, \mathcal{O}_i\right]\mathcal{O}_j\mathcal{O}_k + \mathcal{O}_i\mathcal{O}_j\left[L_n, \mathcal{O}_k\right], \tag{D.17a}$$

$$\left[\bar{L}_n, \mathcal{O}_i\mathcal{O}_j\mathcal{O}_k\right] = \mathcal{O}_i\left[\bar{L}_n, \mathcal{O}_j\right]\mathcal{O}_k + \left[\bar{L}_n, \mathcal{O}_i\right]\mathcal{O}_j\mathcal{O}_k + \mathcal{O}_i\mathcal{O}_j\left[\bar{L}_n, \mathcal{O}_k\right], \quad n = -1, 0, 1, \tag{D.17b}$$

and use

$$\langle [L_n, \mathcal{O}_i \mathcal{O}_j \mathcal{O}_k] \rangle = 0, \qquad \langle [\bar{L}_n, \mathcal{O}_i \mathcal{O}_j \mathcal{O}_k] \rangle = 0, \tag{D.18}$$

namely the vacuum is invariant under global conformal transformation. We will denote

$$\mathbb{P}_3 = \frac{1}{z_{12}^{h_1+h_2-h_3} \bar{z}_{12}^{\bar{h}_1+\bar{h}_2-\bar{h}_3} z_{13}^{h_1+h_3-h_2} \bar{z}_{13}^{\bar{h}_1+\bar{h}_3-\bar{h}_2} z_{23}^{h_2+h_3-h_1} \bar{z}_{23}^{\bar{h}_2+\bar{h}_3-\bar{h}_1}}, \tag{D.19}$$

$$\tau_1 = \ln \frac{z_{23}\bar{z}_{23}}{z_{12}\bar{z}_{12}z_{13}\bar{z}_{13}}, \qquad \tau_2 = \ln \frac{z_{13}\bar{z}_{13}}{z_{12}\bar{z}_{12}z_{23}\bar{z}_{23}}, \qquad \tau_3 = \ln \frac{z_{12}\bar{z}_{12}}{z_{13}\bar{z}_{13}z_{23}\bar{z}_{23}}. \tag{D.20}$$

For three-point functions that involve two non-logarithmic operators and a rank-2 Jordan block $(t, T)$ we have

$$\langle \sigma(z_1,\bar{z}_1)\sigma(z_2,\bar{z}_2)T(z_3)\rangle = C_{\sigma\sigma T}\mathbb{P}_3, \tag{D.21a}$$

$$\langle \sigma(z_1,\bar{z}_1)\sigma(z_2,\bar{z}_2)t(z_3,\bar{z}_3)\rangle = \left(C_{\sigma\sigma t} + C_{\sigma\sigma T}\tau_3\right)\mathbb{P}_3, \tag{D.21b}$$

and with a rank-3 Jordan block $(\Psi_2, \Psi_1, \Psi_0)$, we have

$$\langle \sigma(z_1,\bar{z}_1)\sigma(z_2,\bar{z}_2)\Psi_0(z_3,\bar{z}_3)\rangle = C_{\sigma\sigma\Psi_0}\mathbb{P}_3, \tag{D.22a}$$

$$\langle \sigma(z_1,\bar{z}_1)\sigma(z_2,\bar{z}_2)\Psi_1(z_3,\bar{z}_3)\rangle = \left(C_{\sigma\sigma\Psi_1} + C_{\sigma\sigma\Psi_0}\tau_3\right)\mathbb{P}_3, \tag{D.22b}$$

$$\langle \sigma(z_1,\bar{z}_1)\sigma(z_2,\bar{z}_2)\Psi_2(z_3,\bar{z}_3)\rangle = \left(C_{\sigma\sigma\Psi_2} + C_{\sigma\sigma\Psi_1}\tau_3 + \frac{1}{2}C_{\sigma\sigma\Psi_0}\tau_3^2\right)\mathbb{P}_3. \tag{D.22c}$$

Further consider the three-point functions involving two operators in a rank-2 Jordan block $(\tilde{\varepsilon}, \varepsilon)$ and a rank-2 Jordan block $(t, T)$, we have the following three-point functions:

$$\langle \varepsilon(z_1,\bar{z}_1)\varepsilon(z_2,\bar{z}_2)T(z_3)\rangle = C_{\varepsilon\varepsilon T}\mathbb{P}_3, \tag{D.23a}$$

$$\langle \varepsilon(z_1,\bar{z}_1)\varepsilon(z_2,\bar{z}_2)t(z_3,\bar{z}_3)\rangle = \left(C_{\varepsilon\varepsilon t} + C_{\varepsilon\varepsilon T}\tau_3\right)\mathbb{P}_3, \tag{D.23b}$$

and

$$\langle \tilde{\varepsilon}(z_1,\bar{z}_1)\varepsilon(z_2,\bar{z}_2)T(z_3)\rangle = \left(C_{\tilde{\varepsilon}\varepsilon T} + C_{\varepsilon\varepsilon T}\tau_1\right)\mathbb{P}_3, \tag{D.24a}$$

$$\langle \tilde{\varepsilon}(z_1,\bar{z}_1)\varepsilon(z_2,\bar{z}_2)t(z_3,\bar{z}_3)\rangle = \left(C_{\tilde{\varepsilon}\varepsilon t} + C_{\tilde{\varepsilon}\varepsilon T}\tau_3 + C_{\varepsilon\varepsilon t}\tau_1 + C_{\varepsilon\varepsilon T}\tau_1\tau_3\right)\mathbb{P}_3, \tag{D.24b}$$

and

$$\langle \tilde{\varepsilon}(z_1,\bar{z}_1)\tilde{\varepsilon}(z_2,\bar{z}_2)T(z_3)\rangle = \left(C_{\tilde{\varepsilon}\tilde{\varepsilon}T} + C_{\varepsilon\tilde{\varepsilon}T}\tau_1 + C_{\tilde{\varepsilon}\varepsilon T}\tau_2 + C_{\varepsilon\varepsilon T}\tau_1\tau_2\right)\mathbb{P}_3, \tag{D.25a}$$

$$\begin{aligned}\langle \tilde{\varepsilon}(z_1,\bar{z}_1)\tilde{\varepsilon}(z_2,\bar{z}_2)t(z_3,\bar{z}_3)\rangle = \big(&C_{\tilde{\varepsilon}\tilde{\varepsilon}t} + C_{\varepsilon\tilde{\varepsilon}t}\tau_1 + C_{\tilde{\varepsilon}\tilde{\varepsilon}T}\tau_3 + C_{\tilde{\varepsilon}\varepsilon t}\tau_2 \\ &+ C_{\varepsilon\tilde{\varepsilon}T}\tau_1\tau_3 + C_{\tilde{\varepsilon}\varepsilon T}\tau_2\tau_3 + C_{\varepsilon\varepsilon t}\tau_1\tau_2 + C_{\varepsilon\varepsilon T}\tau_1\tau_2\tau_3\big)\mathbb{P}_3.\end{aligned} \tag{D.25b}$$

Note that Bose symmetry requires for example $C_{\varepsilon\tilde{\varepsilon}T} = C_{\tilde{\varepsilon}\varepsilon T}$, etc.

Finally consider the rank-3 Jordan block $(\Psi_2, \Psi_1, \Psi_0)$, solving the conformal Ward identities gives:

$$\langle \varepsilon(z_1,\bar{z}_1)\varepsilon(z_2,\bar{z}_2)\Psi_0(z_3,\bar{z}_3)\rangle = C_{\varepsilon\varepsilon\Psi_0}\mathbb{P}_3, \tag{D.26a}$$

$$\langle \varepsilon(z_1,\bar{z}_1)\varepsilon(z_2,\bar{z}_2)\Psi_1(z_3,\bar{z}_3)\rangle = \left(C_{\varepsilon\varepsilon\Psi_1} + C_{\varepsilon\varepsilon\Psi_0}\tau_3\right)\mathbb{P}_3, \tag{D.26b}$$

$$\langle \varepsilon(z_1,\bar{z}_1)\varepsilon(z_2,\bar{z}_2)\Psi_2(z_3,\bar{z}_3)\rangle = \left(C_{\varepsilon\varepsilon\Psi_2} + C_{\varepsilon\varepsilon\Psi_1}\tau_3 + \frac{1}{2}C_{\varepsilon\varepsilon\Psi_0}\tau_3^2\right)\mathbb{P}_3, \tag{D.26c}$$

and

$$\langle \tilde{\varepsilon}(z_1,\bar{z}_1)\varepsilon(z_2,\bar{z}_2)\Psi_0(z_3,\bar{z}_3)\rangle = \left(C_{\tilde{\varepsilon}\varepsilon\Psi_0} + C_{\varepsilon\varepsilon\Psi_0}\tau_1\right)\mathbb{P}_3\,, \tag{D.27a}$$

$$\langle \tilde{\varepsilon}(z_1,\bar{z}_1)\varepsilon(z_2,\bar{z}_2)\Psi_1(z_3,\bar{z}_3)\rangle = \left(C_{\tilde{\varepsilon}\varepsilon\Psi_1} + C_{\tilde{\varepsilon}\varepsilon\Psi_0}\tau_3 + C_{\varepsilon\varepsilon\Psi_1}\tau_1 + C_{\varepsilon\varepsilon\Psi_0}\tau_1\tau_3\right)\mathbb{P}_3\,, \tag{D.27b}$$

$$\langle \tilde{\varepsilon}(z_1,\bar{z}_1)\varepsilon(z_2,\bar{z}_2)\Psi_2(z_3,\bar{z}_3)\rangle = \left(C_{\tilde{\varepsilon}\varepsilon\Psi_2} + C_{\tilde{\varepsilon}\varepsilon\Psi_1}\tau_3 + C_{\varepsilon\varepsilon\Psi_2}\tau_1 \right. \tag{D.27c}$$
$$\left. + \frac{1}{2}C_{\tilde{\varepsilon}\varepsilon\Psi_0}\tau_3^2 + C_{\varepsilon\varepsilon\Psi_1}\tau_1\tau_3 + \frac{1}{2}C_{\varepsilon\varepsilon\Psi_0}\tau_1\tau_3^2\right)\mathbb{P}_3\,,$$

and

$$\langle \tilde{\varepsilon}(z_1,\bar{z}_1)\tilde{\varepsilon}(z_2,\bar{z}_2)\Psi_0(z_3,\bar{z}_3)\rangle = \left(C_{\tilde{\varepsilon}\tilde{\varepsilon}\Psi_0} + C_{\varepsilon\tilde{\varepsilon}\Psi_0}\tau_1 + C_{\tilde{\varepsilon}\varepsilon\Psi_0}\tau_2 + C_{\varepsilon\varepsilon\Psi_0}\tau_1\tau_2\right)\mathbb{P}_3\,, \tag{D.28a}$$

$$\langle \tilde{\varepsilon}(z_1,\bar{z}_1)\tilde{\varepsilon}(z_2,\bar{z}_2)\Psi_1(z_3,\bar{z}_3)\rangle = \left(C_{\tilde{\varepsilon}\tilde{\varepsilon}\Psi_1} + C_{\tilde{\varepsilon}\tilde{\varepsilon}\Psi_0}\tau_3 + C_{\varepsilon\tilde{\varepsilon}\Psi_1}\tau_1 + C_{\tilde{\varepsilon}\varepsilon\Psi_1}\tau_2 \right. \tag{D.28b}$$
$$\left. + C_{\varepsilon\tilde{\varepsilon}\Psi_0}\tau_1\tau_3 + C_{\tilde{\varepsilon}\varepsilon\Psi_0}\tau_2\tau_3 + C_{\varepsilon\varepsilon\Psi_1}\tau_1\tau_2 + C_{\varepsilon\varepsilon\Psi_0}\tau_1\tau_2\tau_3\right)\mathbb{P}_3\,,$$

$$\langle \tilde{\varepsilon}(z_1,\bar{z}_1)\tilde{\varepsilon}(z_2,\bar{z}_2)\Psi_2(z_3,\bar{z}_3)\rangle = \left(C_{\tilde{\varepsilon}\tilde{\varepsilon}\Psi_2} + C_{\varepsilon\tilde{\varepsilon}\Psi_2}\tau_1 + C_{\tilde{\varepsilon}\varepsilon\Psi_2}\tau_2 + C_{\tilde{\varepsilon}\tilde{\varepsilon}\Psi_1}\tau_3 + C_{\varepsilon\varepsilon\Psi_2}\tau_1\tau_2 \right. \tag{D.28c}$$
$$\left. + C_{\tilde{\varepsilon}\varepsilon\Psi_1}\tau_2\tau_3 + C_{\varepsilon\tilde{\varepsilon}\Psi_1}\tau_1\tau_3 + \frac{1}{2}C_{\tilde{\varepsilon}\tilde{\varepsilon}\Psi_0}\tau_3^2 + C_{\varepsilon\varepsilon\Psi_1}\tau_1\tau_2\tau_3 \right.$$
$$\left. + \frac{1}{2}C_{\tilde{\varepsilon}\varepsilon\Psi_0}\tau_2\tau_3^2 + \frac{1}{2}C_{\varepsilon\tilde{\varepsilon}\Psi_0}\tau_1\tau_3^2 + \frac{1}{2}C_{\varepsilon\varepsilon\Psi_0}\tau_1\tau_2\tau_3^2\right)\mathbb{P}_3\,.$$

Recall the possible change of basis (D.5) and (D.6) for the Jordan blocks. For the three-point functions (D.21) and (D.22), under (D.5) and (D.6) for the rank-2 and rank-3 Jordan blocks, the three-point constants change as

$$C_{\sigma\sigma T} \to C_{\sigma\sigma T}\,, \qquad C_{\sigma\sigma t} \to C_{\sigma\sigma t} + r_0 C_{\sigma\sigma T}\,, \tag{D.29}$$

and

$$C_{\sigma\sigma\Psi_0} \to C_{\sigma\sigma\Psi_0}\,, \quad C_{\sigma\sigma\Psi_1} \to C_{\sigma\sigma\Psi_1} + r_1 C_{\sigma\sigma\Psi_0}\,, \quad C_{\sigma\sigma\Psi_2} \to C_{\sigma\sigma\Psi_2} + r_1 C_{\sigma\sigma\Psi_1} + r_0 C_{\sigma\sigma\Psi_0}\,. \tag{D.30}$$

Therefore, the three-point constants $C_{\sigma\sigma T}$ and $C_{\sigma\sigma\Psi_0}$ are "intrinsic" similar to the logarithmic couplings $b_0, a_0$ in the sense that they are independent of the basis and we have determined them in the main text. The three-point constants in (D.23) – (D.28) are similar. For example, under the change of basis for $(t, T)$ and $(\Psi_2, \Psi_1, \Psi_0)$, one has

$$C_{\varepsilon\varepsilon T} \to C_{\varepsilon\varepsilon T}\,, \qquad C_{\varepsilon\varepsilon t} \to C_{\varepsilon\varepsilon t} + r_0 C_{\varepsilon\varepsilon T}\,, \qquad \dots \tag{D.31a}$$

$$C_{\varepsilon\varepsilon\Psi_0} \to C_{\varepsilon\varepsilon\Psi_0}\,, \qquad C_{\varepsilon\varepsilon\Psi_1} \to C_{\varepsilon\varepsilon\Psi_1} + r_1 C_{\varepsilon\varepsilon\Psi_0}\,, \qquad \dots \tag{D.31b}$$

However, for $C_{\varepsilon\varepsilon T} = C_{\varepsilon\varepsilon\Psi_0} = 0$ as we have obtained in the main text for the percolation and SAW energy operator, the three-point constants

$$C_{\varepsilon\varepsilon t} \to C_{\varepsilon\varepsilon t}\,, \qquad C_{\varepsilon\varepsilon\Psi_1} \to C_{\varepsilon\varepsilon\Psi_1}\,, \tag{D.32}$$

are invariant under the change of basis. Similarly, under a change of basis for $(\tilde{\varepsilon}, \varepsilon)$, we find

$$C_{\varepsilon\tilde{\varepsilon}T} \to C_{\varepsilon\tilde{\varepsilon}T}\,, \qquad C_{\varepsilon\tilde{\varepsilon}\Psi_0} \to C_{\varepsilon\tilde{\varepsilon}\Psi_0}\,. \tag{D.33}$$

In this sense, these three-point constants become "intrinsic" due to the vanishing $C_{\varepsilon\varepsilon T}, C_{\varepsilon\varepsilon\Psi_0}$ and these are what we have computed in the main text. It would be very interesting to understand what this means physically.

### D.3 Logarithmic OPEs

Given the two- and three-point functions, we can write down the OPE for logarithmic operators. Let us first consider the simple case of logarithmic operators appearing in the OPE of non-logarithmic operators. For this, what we have in mind is OPE of (59) which was previously considered in [21]. We use this case to explain the generic method described in [55] and apply it further below for the more complicated cases.

Consider the OPE of non-logarithmic operators $\Phi$. We want to know the OPE structure involving the rank-2 Jordan block $(\psi_1, \psi_0)$

$$\Phi(z_1, \bar{z}_1)\Phi(z_2, \bar{z}_2) \sim \psi_1(z_2, \bar{z}_2) + \psi_0(z_2, \bar{z}_2), \tag{D.34}$$

or the rank-3 Jordan block

$$\Phi(z_1, \bar{z}_1)\Phi(z_2, \bar{z}_2) \sim \Psi_2(z_2, \bar{z}_2) + \Psi_1(z_2, \bar{z}_2) + \Psi_0(z_2, \bar{z}_2). \tag{D.35}$$

To do this, write down the matrix of

$$\left(G_\Phi^{(3)}\right)_k = \lim_{z_1 \to z_2, \bar{z}_1 \to \bar{z}_2} \langle \Phi(z_1, \bar{z}_1)\Phi(z_2, \bar{z}_2)\mathcal{O}_k(z_3, \bar{z}_3)\rangle, \tag{D.36}$$

where $\mathcal{O}_k$ refers to the logarithmic multiplets. In the case that $\Phi$ is not logarithmic, we have only one row, and the number of columns correspond to the rank of the Jordan block $\mathcal{O}_k$. More concretely, for rank-2 $\psi_k$ we have

$$G_\Phi^{(3)} = \left[\lim_{z_1 \to z_2, \bar{z}_1 \to \bar{z}_2} \langle \Phi(z_1, \bar{z}_1)\Phi(z_2, \bar{z}_2)\psi_1(z_3, \bar{z}_3)\rangle \quad \lim_{z_1 \to z_2, \bar{z}_1 \to \bar{z}_2} \langle \Phi(z_1, \bar{z}_1)\Phi(z_2, \bar{z}_2)\psi_0(z_3, \bar{z}_3)\rangle\right]. \tag{D.37}$$

Then we define the two-point function matrix (in the case of rank-2 $\psi_k$)

$$G_\psi^{(2)} = \begin{bmatrix} \langle \psi_1(z_2, \bar{z}_2)\psi_1(z_3, \bar{z}_3)\rangle & \langle \psi_1(z_2, \bar{z}_2)\psi_0(z_3, \bar{z}_3)\rangle \\ \langle \psi_0(z_2, \bar{z}_2)\psi_1(z_3, \bar{z}_3)\rangle & \langle \psi_0(z_2, \bar{z}_2)\psi_0(z_3, \bar{z}_3)\rangle \end{bmatrix}. \tag{D.38}$$

The OPE structure is then given by the matrix

$$G_\Phi = G_\Phi^{(3)}\left(G_\psi^{(2)}\right)^{-1}. \tag{D.39}$$

In the case of rank-2 Jordan block with two-point functions

$$\langle \psi_1(z_2, \bar{z}_2)\psi_1(z_3, \bar{z}_3)\rangle = \frac{b_1 - 2b_0 \ln(z_{23}\bar{z}_{23})}{z_{23}^{2h}\bar{z}_{23}^{2\bar{h}}}, \tag{D.40a}$$

$$\langle \psi_1(z_2, \bar{z}_2)\psi_0(z_3, \bar{z}_3)\rangle = \frac{b_0}{z_{23}^{2h}\bar{z}_{23}^{2\bar{h}}}, \tag{D.40b}$$

$$\langle \psi_0(z_2, \bar{z}_2)\psi_0(z_3, \bar{z}_3)\rangle = 0, \tag{D.40c}$$

and three-point functions

$$\langle \Phi(z_1, \bar{z}_1)\Phi(z_2, \bar{z}_2)\psi_0(z_3, \bar{z}_3)\rangle = \frac{C_{\Phi\Phi\psi_0}}{z_{12}^{2\Delta-h}\bar{z}_{12}^{2\Delta-\bar{h}}z_{23}^h\bar{z}_{23}^{\bar{h}}z_{13}^h\bar{z}_{13}^{\bar{h}}}, \tag{D.41a}$$

$$\langle \Phi(z_1, \bar{z}_1)\Phi(z_2, \bar{z}_2)\psi_1(z_3, \bar{z}_3)\rangle = \frac{C_{\Phi\Phi\psi_1} + C_{\Phi\Phi\psi_0}\ln\frac{z_{12}\bar{z}_{12}}{z_{13}\bar{z}_{13}z_{23}\bar{z}_{23}}}{z_{12}^{2\Delta-h}\bar{z}_{12}^{2\Delta-\bar{h}}z_{23}^h\bar{z}_{23}^{\bar{h}}z_{13}^h\bar{z}_{13}^{\bar{h}}}, \tag{D.41b}$$

where $(\Delta, \Delta)$ indicate the dimension of $\Phi$ (taken to be diagonal here) and $(h, \bar{h})$ the dimension of $(\psi_1, \psi_0)$. Using (D.39) we can extract

$$\Phi(z_1, \bar{z}_1)\Phi(z_2, \bar{z}_2) = (z_{12}\bar{z}_{12})^{-2\Delta} z_{12}^h \bar{z}_{12}^{\bar{h}} \Bigg( \frac{C_{\Phi\Phi\psi_0}}{b_0}\psi_1 + \frac{C_{\Phi\Phi\psi_0}}{b_0}\ln(z_{12}\bar{z}_{12})\psi_0 \\ + \frac{b_0 C_{\Phi\Phi\psi_1} - b_1 C_{\Phi\Phi\psi_0}}{b_0^2}\psi_0 + \dots \Bigg).$$
(D.42)

Similarly in the case of rank-3 Jordan block we find

$$\Phi(z_1, \bar{z}_1)\Phi(z_2, \bar{z}_2) = (z_{12}\bar{z}_{12})^{-2\Delta+h}\Bigg( \frac{C_{\Phi\Phi\Psi_0}}{a_0}\Psi_2 + \frac{C_{\Phi\Phi\Psi_0}}{a_0}\ln(z_{12}\bar{z}_{12})\Psi_1 + \frac{a_0 C_{\Phi\Phi\Psi_1} - a_1 C_{\Phi\Phi\Psi_0}}{a_0^2}\Psi_1 \\ + \frac{C_{\Phi\Phi\Psi_0}}{2a_0}\ln^2(z_{12}\bar{z}_{12})\Psi_0 + \frac{a_0 C_{\Phi\Phi\Psi_1} - a_1 C_{\Phi\Phi\Psi_0}}{a_0^2}\ln(z_{12}\bar{z}_{12})\Psi_0 \\ + \frac{a_0^2 C_{\Phi\Phi\Psi_2} - a_2 a_0 C_{\Phi\Phi\Psi_0} - a_1 a_0 C_{\Phi\Phi\Psi_1} + a_1^2 C_{\Phi\Phi\Psi_0}}{a_0^3}\Psi_0 + \dots \Bigg).$$
(D.43)

Recall that the Jordan blocks are defined up to a change of basis (D.5) and (D.6) where for the rank-2 Jordan block there is one parameter $r_0$ and for the rank-3 Jordan block, there are two.[22] This freedom allows us to reduce the logarithmic OPE to a simple form, by setting

$$b_0 C_{\Phi\Phi\psi_1} - b_1 C_{\Phi\Phi\psi_0} = 0,$$
$$a_0 C_{\Phi\Phi\Psi_1} - a_1 C_{\Phi\Phi\Psi_0} = 0,$$
$$a_0^2 C_{\Phi\Phi\Psi_2} - a_2 a_0 C_{\Phi\Phi\Psi_0} - a_1 a_0 C_{\Phi\Phi\Psi_1} + a_1^2 C_{\Phi\Phi\Psi_0} = 0,$$
(D.44)

solved by

$$C_{\Phi\Phi\psi_1} = \frac{b_1 C_{\Phi\Phi\psi_0}}{b_0}, \qquad C_{\Phi\Phi\Psi_1} = \frac{a_1 C_{\Phi\Phi\Psi_0}}{a_0}, \qquad C_{\Phi\Phi\Psi_2} = \frac{a_2 C_{\Phi\Phi\Psi_0}}{a_0}.$$
(D.45)

This simplifies the OPEs (D.42) and (D.43) to

$$\Phi(z_1, \bar{z}_1)\Phi(z_2, \bar{z}_2) = (z_{12}\bar{z}_{12})^{-2\Delta} z_{12}^h \bar{z}_{12}^{\bar{h}} \frac{C_{\Phi\Phi\psi_0}}{b_0}\big(\psi_1 + \ln(z_{12}\bar{z}_{12})\psi_0\big) + \dots,$$
(D.46)

and

$$\Phi(z_1, \bar{z}_1)\Phi(z_2, \bar{z}_2) = (z_{12}\bar{z}_{12})^{-2\Delta+h}\frac{C_{\Phi\Phi\Psi_0}}{a_0}\Big(\Psi_2 + \ln(z_{12}\bar{z}_{12})\Psi_1 + \frac{1}{2}\ln^2(z_{12}\bar{z}_{12})\Psi_0\Big) + \dots,$$
(D.47)

as we have done in e.g. (59) in the main text.

The case of OPE of logarithmic operators is a direct generalization. Consider the logarithmic pair $(\tilde{\varepsilon}, \varepsilon)$. In this case, the matrix (D.37) has multiple rows:

$$G_{\tilde{\varepsilon},\varepsilon}^{(3)} = \begin{bmatrix} \lim_{z_1\to z_2,\bar{z}_1\to\bar{z}_2}\langle\tilde{\varepsilon}(z_1,\bar{z}_1)\tilde{\varepsilon}(z_2,\bar{z}_2)\psi_1(z_3,\bar{z}_3)\rangle & \lim_{z_1\to z_2,\bar{z}_1\to\bar{z}_2}\langle\tilde{\varepsilon}(z_1,\bar{z}_1)\tilde{\varepsilon}(z_2,\bar{z}_2)\psi_0(z_3,\bar{z}_3)\rangle \\ \lim_{z_1\to z_2,\bar{z}_1\to\bar{z}_2}\langle\tilde{\varepsilon}(z_1,\bar{z}_1)\varepsilon(z_2,\bar{z}_2)\psi_1(z_3,\bar{z}_3)\rangle & \lim_{z_1\to z_2,\bar{z}_1\to\bar{z}_2}\langle\tilde{\varepsilon}(z_1,\bar{z}_1)\varepsilon(z_2,\bar{z}_2)\psi_0(z_3,\bar{z}_3)\rangle \\ \lim_{z_1\to z_2,\bar{z}_1\to\bar{z}_2}\langle\varepsilon(z_1,\bar{z}_1)\varepsilon(z_2,\bar{z}_2)\psi_1(z_3,\bar{z}_3)\rangle & \lim_{z_1\to z_2,\bar{z}_1\to\bar{z}_2}\langle\varepsilon(z_1,\bar{z}_1)\varepsilon(z_2,\bar{z}_2)\psi_0(z_3,\bar{z}_3)\rangle \end{bmatrix},$$
(D.48)

and similarly for the rank-3 Jordan block where there are three columns corresponding to $(\Psi_2, \Psi_1, \Psi_0)$. The OPE structure can be obtained similar to (D.39) through:

$$G_{\tilde{\varepsilon},\varepsilon} = G_{\tilde{\varepsilon},\varepsilon}^{(3)}\big(G_\Psi^{(2)}\big)^{-1}.$$
(D.49)

---

[22]In [51], this was used to bring the two-point functions of the Jordan blocks into a canonical form. See appendix A in [51].

The full expression in the generic case is quite long and tedious. However, for the energy logarithmic pair $(\tilde{\varepsilon}, \varepsilon)$, we find in the main text the special condition

$$C_{\varepsilon\varepsilon T} = 0, \qquad C_{\varepsilon\varepsilon\Psi_0} = 0, \tag{D.50}$$

which greatly simplifies the form of the OPE. Taking $(\psi_1, \psi_0)$ to be $(t, T)$ and $(\Psi_2, \Psi_1, \Psi_0)$ the rank-3 Jordan block, we can write down directly (e.g. for $\varepsilon^{\text{perco}}$):

$$\varepsilon(z_1, \bar{z}_1)\varepsilon(z_2, \bar{z}_2) = (z_{12}\bar{z}_{12})^{-2h_\varepsilon} \left\{ z_{12}^2 \frac{C_{\varepsilon\varepsilon t}}{b_0} T + c.c. + \ldots + (z_{12}\bar{z}_{12})^2 \right. \tag{D.51}$$

$$\left. \times \left( \frac{C_{\varepsilon\varepsilon\Psi_1}}{a_0} \Psi_1 + \frac{C_{\varepsilon\varepsilon\Psi_1}}{a_0} \ln(z_{12}\bar{z}_{12})\Psi_0 + \frac{a_0 C_{\varepsilon\varepsilon\Psi_2} - a_1 C_{\varepsilon\varepsilon\Psi_1}}{a_0^2} \Psi_0 \right) + \ldots \right\},$$

$$\tilde{\varepsilon}(z_1, \bar{z}_1)\varepsilon(z_2, \bar{z}_2) = (z_{12}\bar{z}_{12})^{-2h_\varepsilon} \left\{ z_{12}^2 \left( \frac{C_{\tilde{\varepsilon}\varepsilon T}}{b_0} t + \frac{b_0 C_{\tilde{\varepsilon}\varepsilon t} - b_1 C_{\tilde{\varepsilon}\varepsilon T}}{b_0^2} T + \frac{C_{\tilde{\varepsilon}\varepsilon T} - C_{\varepsilon\varepsilon t}}{b_0} \ln(z_{12}\bar{z}_{12})T \right) + c.c. + \ldots \right.$$

$$+ (z_{12}\bar{z}_{12})^2 \left( \frac{C_{\tilde{\varepsilon}\varepsilon\Psi_0}}{a_0} \Psi_2 + \frac{a_0 C_{\tilde{\varepsilon}\varepsilon\Psi_1} - a_1 C_{\tilde{\varepsilon}\varepsilon\Psi_0}}{a_0^2} \Psi_1 \right.$$

$$+ \frac{C_{\tilde{\varepsilon}\varepsilon\Psi_0} - C_{\varepsilon\varepsilon\Psi_1}}{a_0} \ln(z_{12}\bar{z}_{12})\Psi_1$$

$$+ \frac{a_0^2 C_{\tilde{\varepsilon}\varepsilon\Psi_2} - a_2 a_0 C_{\tilde{\varepsilon}\varepsilon\Psi_0} + a_1^2 C_{\tilde{\varepsilon}\varepsilon\Psi_0} - a_1 a_0 C_{\tilde{\varepsilon}\varepsilon\Psi_1}}{a_0^3} \Psi_0$$

$$+ \frac{a_1(C_{\varepsilon\varepsilon\Psi_1} - C_{\tilde{\varepsilon}\varepsilon\Psi_0}) - a_0(C_{\varepsilon\varepsilon\Psi_2} + C_{\varepsilon\varepsilon\Psi_1} - 2C_{\tilde{\varepsilon}\varepsilon\Psi_1})}{a_0^2} \tag{D.52}$$

$$\left. \times \ln(z_{12}\bar{z}_{12})\Psi_0 + \frac{C_{\tilde{\varepsilon}\varepsilon\Psi_0} - 2C_{\varepsilon\varepsilon\Psi_1}}{2a_0} \ln^2(z_{12}\bar{z}_{12})\Psi_0 \right) + \ldots \right\},$$

$$\tilde{\varepsilon}(z_1, \bar{z}_1)\tilde{\varepsilon}(z_2, \bar{z}_2)$$

$$= (z_{12}\bar{z}_{12})^{-2h_\varepsilon} \left\{ z_{12}^2 \left( \frac{C_{\tilde{\varepsilon}\tilde{\varepsilon}T}}{b_0} t - \frac{2C_{\tilde{\varepsilon}\varepsilon T}}{b_0} \ln(z_{12}\bar{z}_{12})t + \frac{b_0 C_{\tilde{\varepsilon}\tilde{\varepsilon}t} - b_1 C_{\tilde{\varepsilon}\tilde{\varepsilon}T}}{b_0^2} T \right. \right.$$

$$+ \frac{b_0 C_{\tilde{\varepsilon}\tilde{\varepsilon}T} + 2b_1 C_{\tilde{\varepsilon}\varepsilon T} - 2b_0 C_{\tilde{\varepsilon}\varepsilon t}}{b_0^2} \ln(z_{12}\bar{z}_{12})T$$

$$\left. + \frac{C_{\varepsilon\varepsilon t} - 2C_{\tilde{\varepsilon}\varepsilon T}}{b_0} \ln^2(z_{12}\bar{z}_{12})T \right) + c.c. + \ldots$$

$$+ (z_{12}\bar{z}_{12})^2 \left( \frac{C_{\tilde{\varepsilon}\tilde{\varepsilon}\Psi_0}}{a_0} \Psi_2 - \frac{2C_{\tilde{\varepsilon}\varepsilon\Psi_0}}{a_0} \ln(z_{12}\bar{z}_{12})\Psi_2 + \frac{a_0 C_{\tilde{\varepsilon}\tilde{\varepsilon}\Psi_1} - a_1 C_{\tilde{\varepsilon}\tilde{\varepsilon}\Psi_0}}{a_0^2} \Psi_1 \right.$$

$$+ \frac{a_0(C_{\tilde{\varepsilon}\tilde{\varepsilon}\Psi_0} - 2C_{\tilde{\varepsilon}\varepsilon\Psi_1}) + 2a_1 C_{\tilde{\varepsilon}\varepsilon\Psi_0}}{a_0^2} \ln(z_{12}\bar{z}_{12})\Psi_1 \tag{D.53}$$

$$\left. + \frac{C_{\varepsilon\varepsilon\Psi_1} - 2C_{\tilde{\varepsilon}\varepsilon\Psi_0}}{a_0} \ln^2(z_{12}\bar{z}_{12})\Psi_1 \right)$$

$$+ \frac{a_0^2 C_{\tilde{\varepsilon}\tilde{\varepsilon}\Psi_2} - a_2 a_0 C_{\tilde{\varepsilon}\tilde{\varepsilon}\Psi_0} + a_1^2 C_{\tilde{\varepsilon}\tilde{\varepsilon}\Psi_0} - a_1 a_0 C_{\tilde{\varepsilon}\tilde{\varepsilon}\Psi_1}}{a_0^3} \Psi_0$$

$$+ \frac{a_0^2(2C_{\tilde{\varepsilon}\tilde{\varepsilon}\Psi_1} - 4C_{\tilde{\varepsilon}\varepsilon\Psi_2}) - 2a_0\left(a_1(C_{\tilde{\varepsilon}\tilde{\varepsilon}\Psi_0} - 2C_{\tilde{\varepsilon}\varepsilon\Psi_1}) - 2a_2 C_{\tilde{\varepsilon}\varepsilon\Psi_0}\right) - 4a_1^2 C_{\tilde{\varepsilon}\varepsilon\Psi_0}}{2a_0^3} \ln(z_{12}\bar{z}_{12})\Psi_0$$

$$+ \frac{a_0^2(2C_{\varepsilon\varepsilon\Psi_2} - 4C_{\tilde{\varepsilon}\varepsilon\Psi_1} + C_{\tilde{\varepsilon}\tilde{\varepsilon}\Psi_0}) - 2a_0 a_1(C_{\varepsilon\varepsilon\Psi_1} - 2C_{\tilde{\varepsilon}\varepsilon\Psi_0})}{2a_0^3} \ln^2(z_{12}\bar{z}_{12})\Psi_0$$

$$\left. + \frac{C_{\varepsilon\varepsilon\Psi_1} - C_{\tilde{\varepsilon}\varepsilon\Psi_0}}{a_0} \ln^3(z_{12}\bar{z}_{12})\Psi_0 \right) + \ldots \right\}.$$

### D.4 Four-point functions

From the logarithmic conformal data, using the cluster decomposition properties, we can now assemble four-point functions involving logarithmic operators.

Consider first the $s$-channel limit of four-point function of a non-logarithmic operator $\Phi$ with dimension $(\Delta, \Delta)$. As usual, we place the operators $\Phi$ at the most convenient configurations $(\infty, 1, z, 0)$

$$
\begin{aligned}
\langle \Phi(\infty, \infty) \Phi(1,1) \Phi(z, \bar{z}) \Phi(0,0) \rangle &= \lim_{z_1, \bar{z}_1 \to \infty} (z_1 \bar{z}_1)^{2\Delta} \langle \Phi(z_1, \bar{z}_1) \Phi(1,1) \Phi(z, \bar{z}) \Phi(0,0) \rangle \\
&= \sum_{\{\psi\}} \langle \Phi | \Phi(1,1) | \psi \rangle G^{-1} \langle \psi | \Phi(z, \bar{z}) | \Phi \rangle,
\end{aligned}
\tag{D.54}
$$

where $G^{-1}$ is the inverse of Gram matrix, and

$$
\langle \Phi | \Phi(1,1) | \psi \rangle = \lim_{z_1, \bar{z}_1 \to \infty} \langle \Phi(z_1, \bar{z}_1) \Phi(1,1) \psi(0,0) \rangle,
\tag{D.55}
$$

$$
\langle \psi | \Phi(z, \bar{z}) | \Phi \rangle = \lim_{w, \bar{w} \to 0} \langle \tilde{\psi}(w, \bar{w}) \Phi(z, \bar{z}) \Phi(0,0) \rangle,
\tag{D.56}
$$

where $\tilde{\psi}(w, \bar{w})$ is the transformation of the field $\psi(z, \bar{z})$ under inversion

$$
w = \frac{1}{z}, \qquad \bar{w} = \frac{1}{\bar{z}}.
\tag{D.57}
$$

Up to dimensions $(2,2)$, we have identified the following set of operators in the $c = 0$ percolation and SAW bulk CFTs:[23]

$$
\{\psi\} : \mathbb{I}, \varepsilon, \tilde{\varepsilon}, T, \bar{T}, t, \bar{t}, \partial \bar{t}, \bar{\partial} t, \partial^2 \bar{t}, \bar{\partial}^2 t, \Psi_0, \Psi_1, \Psi_2, \ldots,
\tag{D.58}
$$

where $\varepsilon, \tilde{\varepsilon}$ refer to the energy operator logarithmic pair (see section 4.2). The Gram matrix for these states are given by:

$$
\begin{pmatrix} \langle \tilde{\varepsilon} | \tilde{\varepsilon} \rangle & \langle \tilde{\varepsilon} | \varepsilon \rangle \\ \langle \varepsilon | \tilde{\varepsilon} \rangle & \langle \varepsilon | \varepsilon \rangle \end{pmatrix} = \begin{pmatrix} \theta_1 & \gamma \\ \gamma & 0 \end{pmatrix}, \qquad \begin{pmatrix} \langle t | t \rangle & \langle t | T \rangle \\ \langle T | t \rangle & \langle T | T \rangle \end{pmatrix} = \begin{pmatrix} \theta & b \\ b & 0 \end{pmatrix}, \qquad \begin{pmatrix} \langle \bar{\partial} t | \bar{\partial} t \rangle & \langle \bar{\partial} t | \partial \bar{t} \rangle \\ \langle \partial \bar{t} | \bar{\partial} t \rangle & \langle \partial \bar{t} | \partial \bar{t} \rangle \end{pmatrix} = \begin{pmatrix} 2b & 0 \\ 0 & 2b \end{pmatrix},
\tag{D.59}
$$

and for the states with dimension $(2,2)$:

$$
\begin{pmatrix} \langle \Psi_2 | \Psi_2 \rangle & \langle \Psi_2 | \Psi_1 \rangle & \langle \Psi_2 | \Psi_0 \rangle & \langle \Psi_2 | \bar{\partial}^2 t \rangle & \langle \Psi_2 | \partial^2 \bar{t} \rangle \\ \langle \Psi_1 | \Psi_2 \rangle & \langle \Psi_1 | \Psi_1 \rangle & \langle \Psi_1 | \Psi_0 \rangle & \langle \Psi_1 | \bar{\partial}^2 t \rangle & \langle \Psi_1 | \partial^2 \bar{t} \rangle \\ \langle \Psi_0 | \Psi_2 \rangle & \langle \Psi_0 | \Psi_1 \rangle & \langle \Psi_0 | \Psi_0 \rangle & \langle \Psi_0 | \bar{\partial}^2 t \rangle & \langle \Psi_0 | \partial^2 \bar{t} \rangle \\ \langle \bar{\partial}^2 t | \Psi_2 \rangle & \langle \bar{\partial}^2 t | \Psi_1 \rangle & \langle \bar{\partial}^2 t | \Psi_0 \rangle & \langle \bar{\partial}^2 t | \bar{\partial}^2 t \rangle & \langle \bar{\partial}^2 t | \partial^2 \bar{t} \rangle \\ \langle \partial^2 \bar{t} | \Psi_2 \rangle & \langle \partial^2 \bar{t} | \Psi_1 \rangle & \langle \partial^2 \bar{t} | \Psi_0 \rangle & \langle \partial^2 \bar{t} | \bar{\partial}^2 t \rangle & \langle \partial^2 \bar{t} | \partial^2 \bar{t} \rangle \end{pmatrix} = \begin{pmatrix} a_2 & a_1 & a & 0 & 0 \\ a_1 & a & 0 & 0 & 0 \\ a & 0 & 0 & 0 & 0 \\ 0 & 0 & 0 & 4b & 0 \\ 0 & 0 & 0 & 0 & 4b \end{pmatrix}.
\tag{D.60}
$$

In writing the third equation of (D.59), we have used

$$
\langle t | \bar{L}_1 \bar{L}_{-1} | t \rangle = \langle t | [\bar{L}_1, \bar{L}_{-1}] | t \rangle = 2 \langle t | \bar{L}_0 | t \rangle = 2 \langle t | T \rangle = 2b,
\tag{D.61}
$$

with $\bar{L}_1 | t \rangle = 0$. Moreover, for (D.60), we have used

$$
\langle t | \bar{L}_{-1}^2 \bar{L}_{-1}^2 | t \rangle = \langle t | 4 \bar{L}_0 + 8 \bar{L}_0^2 + 8 \bar{L}_0 \bar{L}_{-1} \bar{L}_1 | t \rangle = 4 \langle t | T \rangle = 4b.
\tag{D.62}
$$

The $s$-channel limit of the four-point function of the operator $\Phi$ is then given by: (the normalization

---

[23]We are not concerned with the descendants of $\varepsilon, \tilde{\varepsilon}$ here.

of $\Phi$ taken to be 1)

$$\langle\Phi(\infty)\Phi(1)\Phi(z,\bar{z})\Phi(0)\rangle$$

$$= (z\bar{z})^{-2\Delta}\left\{1 + (z^2+\bar{z}^2)\left[\frac{C_{\Phi\Phi T}(2bC_{\Phi\Phi t}-C_{\Phi\Phi T}\theta)}{b^2} + \frac{C_{\Phi\Phi T}^2}{b}\ln(z\bar{z})\right]\right.$$

$$+ (z\bar{z}^2+z^2\bar{z})\frac{C_{\Phi\Phi T}^2}{2b} + (z\bar{z})^2\frac{C_{\Phi\Phi T}^2}{2b}$$

$$+ (z\bar{z})^{h_\varepsilon}\left[\frac{C_{\Phi\Phi\varepsilon}(2\gamma C_{\Phi\Phi\bar{\varepsilon}}-C_{\Phi\Phi\varepsilon}\theta_1)}{\gamma^2} + \frac{C_{\Phi\Phi\varepsilon}^2}{\gamma}\ln(z\bar{z})\right]$$

$$+ (z\bar{z})^2\left[\frac{a_1^2C_{\Phi\Phi\Psi_0}^2 - aa_2C_{\Phi\Phi\Psi_0}^2 - 2aa_1C_{\Phi\Phi\Psi_0}C_{\Phi\Phi\Psi_1} + a^2C_{\Phi\Phi\Psi_1}^2 + 2a^2C_{\Phi\Phi\Psi_0}C_{\Phi\Phi\Psi_2}}{a^3}\right.$$

$$\left.\left. + \frac{C_{\Phi\Phi\Psi_0}(2aC_{\Phi\Phi\Psi_1}-a_1C_{\Phi\Phi\Psi_0})}{a^2}\ln(z\bar{z}) + \frac{C_{\Phi\Phi\Psi_0}^2}{2a}\ln^2(z\bar{z})\right]+\dots\right\}. \tag{D.63}$$

Clearly the expression of (D.63) is invariant upon changing the basis of the Jordan blocks (see (D.15), (D.16), (D.29) and (D.30)). The expression can be simplified using the fixed basis (D.45) and becomes

$$\langle\Phi(\infty)\Phi(1)\Phi(z,\bar{z})\Phi(0)\rangle = (z\bar{z})^{-2\Delta}\left\{1 + (z^2+\bar{z}^2)\frac{C_{\Phi\Phi T}^2}{b^2}\big(\theta+b\ln(z\bar{z})\big) + (z\bar{z}^2+z^2\bar{z})\frac{C_{\Phi\Phi T}^2}{2b}\right.$$

$$+ (z\bar{z})^2\frac{C_{\Phi\Phi T}^2}{2b} + (z\bar{z})^{h_\varepsilon}\frac{C_{\Phi\Phi\varepsilon}^2}{\gamma^2}\big(\theta_1+\gamma\ln(z\bar{z})\big) \tag{D.64}$$

$$\left. + (z\bar{z})^2\frac{C_{\Phi\Phi\Psi_0}^2}{a_0^2}\left(\frac{a}{2}\ln^2(z\bar{z}) + a_1\ln(z\bar{z}) + a_2\right)+\dots\right\}.$$

Applying to the spin operator in percolation $\Phi\sim\sigma^{\text{perco}}$, we get the $s$-channel expansion of four-point function $\langle\sigma\sigma\sigma\sigma\rangle^{\text{perco}}$ in (143). The same expression (D.64) applied to the spin operator $\Phi\sim\sigma^{\text{hull}}$ for percolation hull in section 2.3 results in the correlation (72).

Let us now consider the bottom field $\varepsilon$ of the energy rank-2 Jordan block in percolation and SAW. The previous expression (D.63) essentially applies but since the bottom field $\varepsilon$ has zero norm, the contribution from identity operator drops out. We find

$$\langle\varepsilon(\infty)\varepsilon(1)\varepsilon(z,\bar{z})\varepsilon(0)\rangle$$

$$= (z\bar{z})^{-2h_\varepsilon}\left\{(z^2+\bar{z}^2)\left[\frac{C_{\varepsilon\varepsilon T}(2bC_{\varepsilon\varepsilon t}-C_{\varepsilon\varepsilon T}\theta)}{b^2} + \frac{C_{\varepsilon\varepsilon T}^2}{b}\ln(z\bar{z})\right]\right.$$

$$+ (z\bar{z}^2+z^2\bar{z})\frac{C_{\varepsilon\varepsilon T}^2}{2b} + (z\bar{z})^2\frac{C_{\varepsilon\varepsilon T}^2}{2b}$$

$$+ (z\bar{z})^{h_\varepsilon}\left[\frac{C_{\varepsilon\varepsilon\varepsilon}(2\gamma C_{\varepsilon\varepsilon\bar{\varepsilon}}-C_{\varepsilon\varepsilon\varepsilon}\theta_1)}{\gamma^2} + \frac{C_{\varepsilon\varepsilon\varepsilon}^2}{\gamma}\ln(z\bar{z})\right] \tag{D.65}$$

$$+ (z\bar{z})^2\left[\frac{a_1^2C_{\varepsilon\varepsilon\Psi_0}^2 - aa_2C_{\varepsilon\varepsilon\Psi_0}^2 - 2aa_1C_{\varepsilon\varepsilon\Psi_0}C_{\varepsilon\varepsilon\Psi_1} + a^2C_{\varepsilon\varepsilon\Psi_1}^2 + 2a^2C_{\varepsilon\varepsilon\Psi_0}C_{\varepsilon\varepsilon\Psi_2}}{a^3}\right.$$

$$\left.\left. + \frac{C_{\varepsilon\varepsilon\Psi_0}(2aC_{\varepsilon\varepsilon\Psi_1}-a_1C_{\varepsilon\varepsilon\Psi_0})}{a^2}\ln(z\bar{z}) + \frac{C_{\varepsilon\varepsilon\Psi_0}^2}{2a}\ln^2(z\bar{z})\right]+\dots\right\}.$$

For both $\varepsilon^{\text{perco}}$ and $\varepsilon^{\text{SAW}}$, since $C_{\varepsilon\varepsilon\varepsilon} = C_{\varepsilon\varepsilon T} = C_{\varepsilon\varepsilon\Psi_0} = 0$, there turns out to be no logarithmic terms, but the four-point functions also do not vanish. One has

$$\langle\varepsilon(\infty)\varepsilon(1)\varepsilon(z,\bar{z})\varepsilon(0)\rangle = (z\bar{z})^{-2h_\varepsilon}\frac{C_{\varepsilon\varepsilon\Psi_1}^2}{a}(z\bar{z})^2 + \dots, \tag{D.66}$$

as we have seen in the main text eqs. (224) and note that this non-trivial four-point function is due to the non-vanishing coupling between the field $\varepsilon$ and the middle field $\Psi_1$ in the rank-3 Jordan block.

Although we do not analyze this in this paper, it is also straightforward to write down the $s$-channel limit of the four-point function involving one top field $\tilde{\varepsilon}$:

$$\langle \varepsilon(\infty)\tilde{\varepsilon}(1)\varepsilon(x)\varepsilon(0)\rangle = (z\bar{z})^{-2h_\varepsilon}\bigg\{(z^2+\bar{z}^2)\bigg[\frac{bC_{\varepsilon\varepsilon T}C_{\tilde{\varepsilon}\varepsilon t} + bC_{\varepsilon\varepsilon t}C_{\tilde{\varepsilon}\varepsilon T} - C_{\varepsilon\varepsilon T}C_{\tilde{\varepsilon}\varepsilon T}\theta}{b^2} + \frac{C_{\varepsilon\varepsilon T}C_{\tilde{\varepsilon}\varepsilon T}}{b}\ln(z\bar{z})\bigg] + \dots$$
$$+ (z\bar{z})^2\big(A + B\ln(z\bar{z}) + C\ln^2(z\bar{z})\big) + \dots\bigg\}, \tag{D.67}$$

where

$$A = \frac{1}{a^3}\Big(2a_1^2 C_{\varepsilon\varepsilon\Psi_0}C_{\varepsilon\tilde{\varepsilon}\Psi_0} - 2aa_1(C_{\varepsilon\tilde{\varepsilon}\Psi_0}C_{\varepsilon\varepsilon\Psi_1} + C_{\varepsilon\varepsilon\Psi_0}C_{\varepsilon\tilde{\varepsilon}\Psi_1})$$
$$+ a\big(-2a_2 C_{\varepsilon\varepsilon\Psi_0}C_{\varepsilon\tilde{\varepsilon}\Psi_0} + 2a(C_{\varepsilon\varepsilon\Psi_1}C_{\varepsilon\tilde{\varepsilon}\Psi_1} + C_{\varepsilon\tilde{\varepsilon}\Psi_0}C_{\varepsilon\varepsilon\Psi_2} + C_{\varepsilon\varepsilon\Psi_0}C_{\varepsilon\tilde{\varepsilon}\Psi_2})\big)\Big), \tag{D.68a}$$
$$B = \frac{aC_{\varepsilon\varepsilon\Psi_0}C_{\varepsilon\tilde{\varepsilon}\Psi_1} + aC_{\varepsilon\tilde{\varepsilon}\Psi_0}C_{\varepsilon\varepsilon\Psi_1} - a_1 C_{\varepsilon\varepsilon\Psi_0}C_{\varepsilon\tilde{\varepsilon}\Psi_0}}{a^2}, \tag{D.68b}$$
$$C = \frac{C_{\varepsilon\varepsilon\Psi_0}C_{\varepsilon\tilde{\varepsilon}\Psi_0}}{2a}. \tag{D.68c}$$

Again, the expressions are invariant under a change of basis in the Jordan blocks. The expression (D.67) again simplifies for $C_{\varepsilon\varepsilon T} = C_{\varepsilon\varepsilon\Psi_0} = 0$:

$$\langle \varepsilon(\infty)\tilde{\varepsilon}(1)\varepsilon(x)\varepsilon(0)\rangle = (z\bar{z})^{-2h_\varepsilon}\bigg((z^2+\bar{z}^2)\frac{C_{\varepsilon\varepsilon t}C_{\tilde{\varepsilon}\varepsilon T}}{b} + (z\bar{z})^2\bigg(\frac{aC_{\varepsilon\tilde{\varepsilon}\Psi_0}C_{\varepsilon\varepsilon\Psi_2} + aC_{\varepsilon\varepsilon\Psi_1}C_{\varepsilon\tilde{\varepsilon}\Psi_1} - a_1 C_{\varepsilon\tilde{\varepsilon}\Psi_0}C_{\varepsilon\varepsilon\Psi_1}}{a^2}$$
$$+ \frac{C_{\varepsilon\tilde{\varepsilon}\Psi_0}C_{\varepsilon\varepsilon\Psi_1}}{a}\ln(z\bar{z})\bigg) + \dots\bigg). \tag{D.69}$$

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
