# Peer review of "Logarithmic operators in $c=0$ bulk CFTs"

_SciPost Physics, doi:SciPost Phys. 19, 008 (2025)_

## Round 1 · Referee Report · Anonymous (Referee 1) · 2025-4-4

Report
Requested changes
-
Below (2.2), diagonal is a property of the theory rather than the operator: a non-diagonal theory can still contain scalar operators. Their OPE contains opertors with non zero spin.
-
Below (2.14), it is unclear why <TT>=0 should mean the operator T being at risk of being removed, given that nothing is said about other correlation functions of T with other operators of the theory. Indeed the operator is not removed.
-
Typo in (2.39), bb instead of b
-
I cannot understand the sentence below (5.5)
-
What is exactly the definition of $\hat{\Phi}_{2,1}$ appearing in (5.48)?
-
The notation of $w \to 0$ in(5.53) is very confusing, given that we already have an operator living at $z=0$. I reccommend sending $w \to \infty$ with the proper rescaling (what is done in (5.50)) rather than defining $w = \frac 1z$ and sending $w \to 0$.
Recommendation
Ask for minor revision

---

## Round 1 · Referee Report · Anonymous (Referee 2) · 2025-4-29

Strengths
2-Well-structured and articulate presentation of ideas
Weaknesses
Report
iIn this paper, the authors highlight the emergence of a logarithmic structure at the c=0c=0 dense and dilute loop critical point. This is achieved primarily by interpreting new bootstrap solutions of c≤1c≤1 theories, both diagonal and non-diagonal. Particular attention is given to the energy operator and its logarithmic properties.
On one hand, this approach is not entirely new. In arXiv:1311.2055, where the occurrence of logarithmic structures for arbitrary c was first identified, similar results were obtained through the analysis of Coulomb gas correlation functions. These earlier results relied on the interplay between the so-called imaginary DOZZ formula and conformal blocks, which are ultimately central to the bootstrap solutions analyzed in the present work.
On the other hand, the authors here apply conformal field theories that have been explicitly connected to statistical models, which adds further significance to their findings. I therefore recommend this paper for publication.
Requested changes
The paper is very well written and clear. The results arXiv:1311.2055 should be cited.
Recommendation
Publish (easily meets expectations and criteria for this Journal; among top 50%)

---

## Round 1 · Referee Report · Anonymous (Referee 3) · 2025-5-9

Report
One of the most intriguing results is the non-trivial four-point function of an energy-type operator discussed in Section 5.3.
While the paper is quite technical and not always easy to follow, it already covers a large amount of material in considerable depth. It would be difficult to improve the clarity significantly without substantially increasing the length of the paper.
Requested changes
-
It would be helpful to precisely define what is meant by 'Kac operators.' In particular, the term 'Kac operator' already appears in the mathematical physics literature, referring to an operator introduced by M. Kac in 1966, which corresponds to the transfer operator for a lattice model in statistical mechanics. As I understand it, in the present paper, the author intends to define 'Kac operators' as all operators with integer Kac indices.
-
I found it somewhat confusing that the author initially motivates the study using the bulk CFTs of percolation and self-avoiding walks, linking these models to cluster and loop model CFTs. However, by the end of the introduction, the terminology shifts to 'the cluster, dilute, and dense loop models,' corresponding respectively to 'percolation, SAW, and percolation hulls.' This change in vocabulary may confuse readers unfamiliar with these distinctions. I suggest that the introduction include a more detailed explanation of the differences between these three cases—particularly between the dilute and dense loop models—and clarify their connections to the physical models mentioned.
-
Reference [21] appears to be incomplete; I believe it corresponds to arXiv:cond-mat/0111031.
Recommendation
Publish (meets expectations and criteria for this Journal)

---

## Round 2 · Referee Report · Anonymous (Referee 1) · 2025-6-16

Report

I am happy with the changes made by the author and I recommend publication

Recommendation

Publish (easily meets expectations and criteria for this Journal; among top 50%)

---

## Round 2 · Author Response

Dear Editor,

We would like to thank you for considering the publication of the paper, and thank all the referees for careful reading of the manuscript, for writing the reports and for the comments that help improving the paper. Below we list the changes made as suggested by the referees and answer some of the questions.

---

## Round 2 · List of Changes

Referee 1: 1, We reworded the sentence below eq.(2.2). 2, From <TT>=0, by state-operator correspondence, the stress-energy tensor corresponds to a zero-norm state, and if it is orthogonal to all other states in the CFT, it will decouple from the CFT state space. In this case, it also does not have non-trivial three point functions with other operators, for example <O_1|O_2|T> would vanish by the OPE of O_1 and O_2. We added a footnote 4 to clarify this argument. 3, There is no typo in eq.(2.39). The b here is given in eq.(2.21) and the b_{1,2} here is given in eq.(2.34). To avoid confusion, we reordered the letters in the expression. 4, We have rewritten the sentence below eq.(5.5) which hopefully clarifies things. 5, The $\hat{\Phi}_{2,1}$ in eq. (5.48) is the Kac operator $\Phi_{2,1}$ (which satisfies the BPZ equation at generic c) but normalized as in eq.(4.25). The s-channel expansion in eq.(5.48) is the usual BPZ solution but taking into account this operator normalization. We included a reference to eq.(4.25) above eq.(5.48). 6, We have modified the expression following the referee's suggestion.

Referee 2: 1, We included the reference arXiv:1311.2055 in the second paragraph of the introduction.

Referee 3: 1, We specified the meaning of 'Kac operators' in the first sentence of the third paragraph in the introduction. 2, We made some clarifications and explanations of the cluster, dense and dilute loop models in the second paragraph of the introduction, we also included additional references on these models. 3, We fixed the citation error of [22] (previously [21]) to include the arXiv number.

---

## Editorial Decision

published